# Combinatorial Ski Rental Problem: Robust and Learning-Augmented Algorithms

**Ziwei Li**
Nanjing University
ziwei.li@smail.nju.edu.cn

**Bo Sun**
University of Ottawa, Vector Institute
bo.sun@uwaterloo.ca

**Zhiqiu Zhang**
Nanjing University
221900399@smail.nju.edu.cn

**Mohammad Hajiesmaili**
University of Massachusetts, Amherst
hajiesmaili@cs.umass.edu

**Binghan Wu**
AsiaInfo Technologies Limited
wubh3@asiainfo.com

**Lin Yang**\*
Nanjing University
linyang@nju.edu.cn

**Yang Gao**
Nanjing University
gaoy@nju.edu.cn

## Abstract

We introduce and study the *Combinatorial Ski Rental* (CSR) problem, which involves multiple items that can be rented or purchased, either individually or in combination. At each time step, a decision-maker must make an irrevocable buy-or-rent decision for items that have not yet been purchased, without knowing the end of the time horizon. We propose a randomized online algorithm, *Sorted Optimal Amortized Cost* (SOAC), that achieves the optimal competitive ratio. Moreover, SOAC can be extended to address various well-known ski rental variants, including the multi-slope, multi-shop, multi-commodity ski rental and CSR with upgrading problems. Building on the proposed SOAC algorithm, we further develop a learning-augmented algorithm that leverages machine-learned predictions to improve the performance of CSR. This algorithm is capable of recovering or improving upon existing results of learning-augmented algorithms in both the classic ski rental and multi-shop ski rental problems. Experimental results validate our theoretical analysis and demonstrate the advantages of our algorithms over baseline methods for ski rental problems.

## 1 Introduction

Sequential decision-making under uncertainty is ubiquitous yet challenging in the digital transformation of society. We study the ski rental problem [17, 8], a classic online decision-making problem that addresses the rent-or-buy dilemma. In its basic form [1, 19, 16, 20], the problem involves repeatedly deciding whether to rent or buy an item without knowing its future usage duration. This paper introduces and studies a more general variant, the Combinatorial Ski Rental (CSR) problem, which extends the rent-or-buy decisions from one item to multiple items.

In the CSR problem, a player participates in a skiing season of unknown duration. Skiing requires multiple essential items, such as skis, boots, helmets, and goggles. Each day, the player must decide whether to buy or rent items that have not yet been purchased, with the goal of minimizing the total rent and purchase cost over the entire season. The combinatorics of CSR stems from the

---

\*Corresponding Author

39th Conference on Neural Information Processing Systems (NeurIPS 2025).

purchase cost, which is cheaper when purchasing multiple items as a combo compared to buying each item separately. The `CSR` model is inspired by real-world scenarios involving decisions over multiple combinations [26, 15, 24], such as selecting data plan bundles in telecommunications [18], configuring flexible service packages in cloud computing [4], and subscribing to software suites (e.g., Microsoft Office) [32], where users must choose among various combinations of applications.

Traditionally, online algorithms for ski rental problems are designed for worst-case scenarios, with performance measured by the classic competitive ratios [7]. While this ensures robustness, it often comes at the cost of poor performance on favorable real-world inputs. To address the pessimism of classic competitive algorithms, *learning-augmented algorithms* [21, 27, 14, 23], as a rapidly emerging field, aim to enhance traditional online algorithms by leveraging predictions from machine learning models. These algorithms are designed to make use of potentially imperfect predictions to achieve two goals: performing near-optimally when the predictions are accurate (i.e., *consistency*) and retaining worst-case guarantees when predictions are misleading (i.e., *robustness*) [33, 34]. In this paper, we aim to design both robust and learning-augmented algorithms for the `CSR` problem.

**Prior work.** Many variants of the ski rental problem have been studied to address increasingly complex application scenarios. In single-item ski rental problems, two main variants have been explored: (i) *multi-shop ski rental*[1, 25], where a skier must first choose one from multiple shops, and then decide whether to rent or buy one item from the chosen shop; (ii) *multi-slope ski rental* [19, 6], where the skier has access to multiple rental and purchase options, and can adaptively switch among them over time. In multi-item settings, in addition to the rent-or-buy decisions, the skier must also determine the order in which to purchase different items, re-

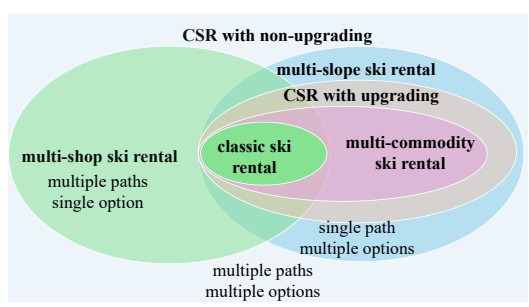

Figure 1: Relationship between `CSR` and other ski rental variants

ferred to as the *purchase path* in this paper. Two special cases of the multi-item ski rental problem have been studied. [29] considers the multi-commodity ski rental problem, where the rental and purchase costs are additive across items. [32] studies a more general model in which both rental and purchase costs are submodular and also frames the model as combinatorial ski rental. However, the problem in [32] assumes that *purchases can be upgraded*, i.e., the player only needs to pay the price difference to sequentially acquire new items. We can show that under the simplified models in [29] and [32], the optimal order of purchasing items is fixed, i.e., there is only one purchase path. As a result, their algorithms focus solely on determining the timing of purchases along this single path, thereby avoiding the challenge of selecting from multiple purchase paths. Algorithmically, both models can be viewed as special cases of the multi-slope ski rental problem for a single item. The relationships among the different variants are illustrated in Figure 1.

The `CSR` problem studied in this work generalizes all the aforementioned models. We consider multiple items with a submodular purchase cost and consider that *purchases cannot be upgraded*, i.e., the full price must be paid for each new item. In this setting, the selection of the purchase path becomes crucial and must be determined jointly with the timing of item purchases, making the problem significantly more challenging. The most relevant work [32] has derived an optimal $e/(e-1)$-competitive algorithm for the `CSR` with upgrading. However, the non-upgrading variant remains open, requiring new algorithmic techniques and analysis. We focus on the variant with non-upgrading, and unless otherwise specified, use `CSR` to refer to the non-upgrading variant.

**Contributions.** In this work, we first propose an optimal randomized algorithm called *Sorted Optimal Amortized Cost* (`SOAC`) for the `CSR` problem. To the best of our knowledge, *this is the first optimal online algorithm for `CSR`*. The `SOAC` algorithm represents a general framework that can obtain optimal solutions across a wide spectrum of ski rental variants, including the multi-slope [19], multi-shop [1], multi-commodity ski rental [29] and `CSR` with upgrading problems [32].

Subsequently, we propose a learning-augmented algorithm for the `CSR` problem, termed `LA-SOAC`, which provides consistency and robustness guarantees. `LA-SOAC` can recover and even improve existing consistency-robustness trade-offs of learning-augmented algorithms for ski rental variants. In particular, we recover the consistency and robustness bounds for the classic ski rental problem

studied in Purohit *et al.* [21]. For the multi-shop variant, we prove that our algorithm achieves both tighter consistency and robustness bounds than those established in Wang *et al.* [25]. Numerical experiments validate our theoretical results and demonstrate the performance advantage of `LA-SOAC`.

**Algorithmic ideas and technical novelty.** Technically, the proposed algorithms `SOAC` and `LA-SOAC` adopt a novel strategy, `OAC`, which makes decisions by tracking the cost incurred by the offline optimal solution. At each time step, it allocates a daily cost budget proportional to the offline benchmark. This design eliminates the need for intricate mathematical case analysis and can be broadly applied to the family of ski rental problems. Furthermore, to address the challenges from multiple purchase paths, we reformulate the problem as computing an augmented path that encompasses all possible purchase paths. We then prove that the competitive ratio is a convex function with respect to the probability distribution over these paths, enabling us to efficiently compute the optimal strategy.

## 2 Problem Statement

### 2.1 The combinatorial ski rental problem

Consider a player who aims to participate in a skiing season with an unknown horizon $T \in \mathbb{N}^+$. To ski on each day, the player must have a set $\mathcal{M} := \{1, 2, \ldots, m\}$ of skiing equipment (referred to as items) that are either rented or purchased. At the beginning of each day, the player must choose to either buy or rent items that have not yet been purchased without knowing the horizon $T$. The goal is to minimize the overall cost of renting and purchasing items during the skiing season.

`CSR` is a generalization of the classic ski rental problem, considering rent-or-buy decisions for multiple items. For convenience, we refer to each individual item as a *base item* and any subset of these base items as a *super item*. $2^{\mathcal{M}}$ denotes the set of super items. Let $g(S) : 2^{\mathcal{M}} \to \mathbb{R}^+$ and $f(S) : 2^{\mathcal{M}} \to \mathbb{R}^+$ denote the rental rate function and the purchase cost function for a super item $S \in 2^{\mathcal{M}}$, respectively, with $f(\emptyset) = g(\emptyset) = 0$. In this paper, we focus on a setup where the rental rate is additive and the purchase cost is submodular.

**Assumption 1** (Additive rental rate). *The rental rate of* `CSR` *$g(S)$ is additive, i.e., $g(S) = \sum_{i \in S} g(\{i\})$, where $g(\{i\})$ is the rental rate of base item $i$.*

**Assumption 2** (Submodular purchase cost). *The purchase cost of* `CSR` *is submodular, i.e., for any $S \in 2^{\mathcal{M}}$ and $Z \subseteq Z' \in 2^{\mathcal{M}}$, it holds that $f(S \mid Z') \leq f(S \mid Z)$, where $f(S \mid Z) := f(S \cup Z) - f(Z)$.*

We assume that the purchase price is submodular because it reflects the fact that buying multiple items together is cheaper than purchasing them individually. The combinatorics of `CSR` stems from this submodularity. Furthermore, we consider an additive rental rate to simplify our analysis, as our results can be readily extended to scenarios with submodular rental rates, since our algorithm relies only on the buy-to-rent ratio ordering, which remains well-defined under submodular rental rates. Both purchase cost and rental rate functions are known to the decision maker upfront.

**Definition 1** (Information Setup $\mathcal{I}$). *We define all information known prior to the online decision maker of* `CSR` *as information setup $\mathcal{I} := \{\mathcal{M}, f(\cdot), g(\cdot)\}$.*

### 2.2 Online formulation

An online deterministic strategy for `CSR` consists of two layers of decisions: the *purchase path* $\sigma := (S_1, S_2, \ldots, S_l)$, which is an ordered sequence of super items such that $\cup_{i=1}^l S_i = \mathcal{M}$; and the associated *purchase time* $\boldsymbol{t}(\sigma) := \{t(S \mid \sigma)\}_{S \in \sigma}$, where $t(S \mid \sigma)$ is the time of purchasing the super item $S$, given that purchase path $\sigma$ is chosen. For randomized strategies, without loss of generality, we focus on the mixed strategy that is a distribution over deterministic strategies. Thus, we denote a general randomized algorithm for `CSR` by $A := A(\boldsymbol{q}, \boldsymbol{p})$, where $\boldsymbol{q} := \{q(\sigma)\}_{\sigma \in \Sigma}$ is the probability of choosing purchase path $\sigma$ over all possible purchase paths in $\Sigma$ and $\boldsymbol{p} := \{p_n(S \mid \sigma)\}_{n \in \mathbb{N}^+, S \in \sigma, \sigma \in \Sigma}$ is the purchase probability, i.e., $p_n(S \mid \sigma)$ represents the probability of purchasing super item $S$ on day $n$, given purchase path $\sigma$. For notational convenience, we omit the index $\sigma$ of super item $S_i$ and path length $l$ in a purchase path $\sigma$. The expected cost of an online algorithm $A$ is

$$\texttt{ALG}(T; A) = \sum_{n \in [T]} \sum_{\sigma \in \Sigma} \sum_{i \in [l]} \Big( q(\sigma) \cdot f(S_i) \cdot p_n(S_i \mid \sigma) + q(\sigma) \cdot (1 - F_n(S_i \mid \sigma)) \cdot g(S_i) \Big), \quad (1)$$

where $F_n(S \mid \sigma) = \sum_{i=1}^n p_i(S \mid \sigma)$ is the cumulative distribution function of buying super item $S$ by day $n$ when purchase path $\sigma$ is chosen. The first summation term inside the parentheses represents the expected purchase cost. And the second term represents the expected rental cost, where $1 - F_n(S_i \mid \sigma)$ denotes the probability that $S_i$ has not yet been purchased by day $n$. Additionally, under the purchase path $\sigma$, $p_n(S_{i+1} \mid \sigma) = 0$ when $F_n(S_i \mid \sigma) < 1$ to enforce the order of purchase in $\sigma$.

## 2.3 Offline optimal structure

The goal of this work is to design an online algorithm for the CSR such that its expected cost is comparable to the optimal offline cost, which is obtained with prior knowledge of the time horizon $T$. Let $\texttt{OPT}(T)$ represent the optimal offline cost of the CSR for a given $T$. Since the purchase cost is submodular, purchasing items as a combo is always cheaper than buying them separately. The offline optimal algorithm is to purchase one super item at the beginning and rent remaining items until $T$. Therefore, $\texttt{OPT}(T)$ can be determined by solving the following equation

$$\texttt{OPT}(T) = \min_{S \in 2^{\mathcal{M}}} \{f(S) + T \cdot g(\mathcal{M} \setminus S)\}. \tag{2}$$

The super item purchased in the beginning varies under different time horizons. Let $\sigma^* := (S_1^*, S_2^*, \ldots, S_K^*)$ denote the sequence of optimal purchase sets as $T$ increases. Note that $\sigma^*$ does not represent multiple purchases over time, but rather a structural summary of how the purchase decision of offline algorithm changes with $T$. However, in special case when the purchase cost can be upgraded [31], $\sigma^*$ can be shown to be the optimal purchase path in an online algorithm, which can be pre-determined in the beginning and thus simplifies the algorithm design. Thus, $\texttt{OPT}(T)$ is a piece-wise linear function of $T$ and we illustrate it using the red curve in Figure 2. Each blue line illustrates the total cost of purchasing and renting, given by $f(S_i^*) + T \cdot g(\mathcal{M} \setminus S_i^*)$, which corresponds to the scenario where one super item $S_i^*$

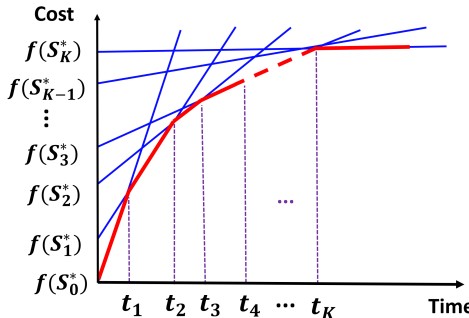

Figure 2: Illustrating the optimal offline cost for the CSR problem.

is purchased at the beginning, while the remaining items are continuously rented until time $T$. The $y$-intercept of the line represents the purchase cost of the super item, and its slope indicates the total rental rate of the remaining items.

Given the information setup $\mathcal{I}$, the performance of an online algorithm $A$ is evaluated by a competitive ratio $\texttt{CR}(A) := \sup_T \frac{\texttt{ALG}(T;A)}{\texttt{OPT}(T)}$, which is the worst-case ratio over all possible skiing horizon $T$. In the following, we design optimal robust algorithms that minimize the competitive ratio among all online algorithms in Section 3, and learning-augmented algorithms by leveraging machine learned predictions in Section 4.

# 3 Competitive Algorithms

In this section, we propose SOAC, an optimal randomized algorithm to solve the CSR. Recall that a randomized algorithm involves two layers of challenging decisions: the purchase path selection, followed by purchase time decisions for each super item in the path. We first consider a purchase path $\sigma$ is fixed, and focus on determining the purchase probability. To address this, we introduce an optimal amortized cost strategy, denoted as OAC. Then we design the optimal algorithm SOAC by constructing an augmented purchase path and employing OAC as subroutines for CSR.

## 3.1 Optimal amortized cost strategy

In the CSR problem, the competitive performance of the online algorithm depends on how to balance the immediate purchase cost against the potential rental costs. A straightforward idea is to amortize the daily costs in order to balance the two sources of costs. Consequently, we define a class of amortized cost strategies, which set the quota of the amortized cost on day $n$ as $\alpha \cdot \Delta\texttt{OPT}(n)$, where

$\Delta \mathtt{OPT}(n) = \mathtt{OPT}(n) - \mathtt{OPT}(n-1)$ is the increment of offline optimal cost in Eq. (2) and $\alpha$ is the target competitive ratio.

Let $\Delta \mathtt{ALG}(n; A) := \mathtt{ALG}(n; A) - \mathtt{ALG}(n-1; A)$ and $\Delta \mathtt{OPT}(n) = \mathtt{OPT}(n) - \mathtt{OPT}(n-1)$ denote the incremental costs of online algorithm $A$ and offline algorithm if the time horizon progresses from $n-1$ to $n$, where $\mathtt{ALG}(n; A)$ is given in Eq. (1) for $\mathtt{CSR}$, and $\mathtt{OPT}(n)$ is defined in Eq. (2).

**Definition 2** (Amortized Cost Strategy, $\mathtt{AC}(\sigma; \alpha)$)**.** *Given a purchase path $\sigma$ (equivalently the path selection probability $\boldsymbol{q}$) and a parameter $\alpha > 1$, $\mathtt{AC}(\sigma; \alpha)$ determines the purchase probability $\boldsymbol{p}$ of an online algorithm $A = (\boldsymbol{q}, \boldsymbol{p})$ by solving the following system of equations*

$$
\begin{aligned}
\Delta \mathit{ALG}(n;\, A) &= \alpha \cdot \Delta \mathit{OPT}(n), \quad \text{when } n < T_A, \\
\Delta \mathit{ALG}(n;\, A) &\leq \alpha \cdot \Delta \mathit{OPT}(n), \quad \text{when } n = T_A, \\
\Delta \mathit{ALG}(n;\, A) &= 0, \quad \text{when } n > T_A.
\end{aligned}
\tag{3}
$$

$\Delta \mathit{ALG}(n;\, A) := \mathit{ALG}(n;\, A) - \mathit{ALG}(n-1;\, A)$ *is the incremental cost of the online algorithm $A$, where $\mathit{ALG}(n;\, A)$ is given in Eq. (1). $T_A$ is the "completion time" when algorithm $A$ purchases all items and the cost stops growing. It is a variable to be determined by solving the system of equations.*

Note that the amortized cost strategy $\mathtt{AC}(\sigma; \alpha)$ and the completion time $T_A$ depend on the parameter $\alpha$. A larger $\alpha$ encourages earlier purchases of items to avoid incurring high rental costs, thereby completing the acquisition of all base items at an earlier stage, i.e., a smaller $T_A$. We define an optimal amortized cost strategy ($\mathtt{OAC}$) as the $\mathtt{AC}(\sigma; \alpha)$ that sets $\alpha$ such that $T_A = T_{\mathtt{OPT}} = \min\{\arg\max_{T \in \mathbb{N}^+} \mathtt{OPT}(T)\}$. $T_{\mathtt{OPT}} \in \mathbb{N}^+$ is the "critical time" when the optimal offline cost stops growing, i.e., the minimum length of time horizon in which the offline algorithm purchases all items in the beginning.

Given a purchase path $\sigma$, Algorithm 1 illustrates the algorithm for the optimal amortized cost strategy ($\mathtt{OAC}(\sigma)$) and its competitive ratio is denoted as $\alpha(\sigma) := \mathtt{CR}(\mathtt{OAC}(\sigma))$. Since $T_A$ is a non-increasing function of $\alpha$, Algorithm 1 employs a dichotomous search to find the $\alpha$ such that $T_A = T_{\mathtt{OPT}}$.

**Example 1.** We show how to apply $\mathtt{OAC}$ for the classic single-item ski rental problem. In this case, the set of items $\mathcal{M}$ contains only one item, i.e., $|\mathcal{M}| = 1$, and the rental rate and purchase price are given by $g(\{1\}) = 1$ and $f(\{1\}) = b$, respectively. The costs of optimal offline algorithm and the online algorithm can be derived as follows: $\mathtt{OPT}(n) = \min\{n, b\}$, $\mathtt{ALG}(n; A) = \sum_{i \in [n]} (i - 1 + b) \cdot p_i(\{1\}) + n \cdot (1 - \sum_{i \in [n]} p_i(\{1\}))$. Based on Eq. (3), $\mathtt{OAC}$ is the solution to the equations:

$$
\Delta \mathtt{ALG}(n; A) = \alpha, \quad n \leq b; \quad \Delta \mathtt{ALG}(n; A) = 0, \quad n > b.
$$

Solving the above equations yields the optimal competitive ratio $\alpha = 1 + \frac{1}{((1-\frac{1}{b})^{-b} - 1)}$ and purchasing probability $p_n(\{1\}) = \left(\frac{b-1}{b}\right)^{b-n} \frac{1}{b(1-(1-(1/b))^b)}, n \leq b$, which match the classic results for the single-item ski rental problem in [12].

### 3.2 The $\mathtt{SOAC}$ algorithm

In $\mathtt{CSR}$, an online algorithm that follows a determinstic purchase path achieves suboptimal performance due to the submodular purchase cost. We next design a *Sorted Optimal Amortized Cost Algorithm* ($\mathtt{SOAC}$, Algorithm 2) that determines the purchase path $\Sigma$ with probability $\boldsymbol{q} = \{q(\sigma)\}_{\sigma \in \Sigma}$ and the corresponding purchase probability $\boldsymbol{p} := \{p_n(S \mid \sigma)\}_{n \in \mathbb{N}^+, S \in \sigma, \sigma \in \Sigma}$ simultaneously. In the high level, $\mathtt{SOAC}$ uses the purchase path probability $\boldsymbol{q}$ to construct an ancillary augmentation path $\gamma^{\mathtt{BR}}(\boldsymbol{q})$. This augmentation path encompasses all possible optimal outcomes for $\mathtt{CSR}$ problems. We show that the competitive ratio of $\mathtt{OAC}(\gamma^{\mathtt{BR}}(\boldsymbol{q}))$ is convex in $\boldsymbol{q}$ (see Lemma 5), and thus we derive the optimal purchasing path probability using projected gradient descent method. In the following, we show the details of $\mathtt{SOAC}$.

**Algorithm 1** OAC: Optimal Amortized Cost

**Input:** $\mathcal{I} = (\mathcal{M}, f, g), \sigma, \varepsilon$
**Initialize**: $\alpha = 1, \alpha_{\max} = f(\mathcal{M}), \alpha_{\min} = 1$
Construct algorithm $A = \text{AC}(\sigma; \alpha)$
Calculate $\text{OPT}(T)$ and $T_{\text{OPT}}$ based on Eq. (2)
**while** $\alpha_{\max} - \alpha_{\min} > \varepsilon$ **do**
    Calculate $T_A$ using $\alpha$ according to
Eq. (1) and Eq. (3)
    **if** $T_A < T_{\text{OPT}}$ **then** $\alpha_{\max} \leftarrow \alpha$
    **else** $\alpha_{\min} \leftarrow \alpha$
    **end if**
    Update $\alpha \leftarrow (\alpha_{\max} + \alpha_{\min})/2$
**end while**

**Algorithm 2** SOAC: Sorted Optimal Amortized Cost

**Input**: $\mathcal{I} = (\mathcal{M}, f, g), \varepsilon, \eta$
All disjoint partitions: $\Gamma = \{\gamma_1, \gamma_2, \ldots, \gamma_{|\Gamma|}\}$
Sort divisions $\Gamma$ by BR: $\{\sigma_1, \sigma_2, \ldots, \sigma_{|\Gamma|}\}$
Construct $\gamma^{\text{BR}}(\boldsymbol{q})$ based on Eq. (4)
**Initialize**: $\alpha \leftarrow f(\mathcal{M}), \alpha_{\text{new}} \leftarrow 1, \Delta\alpha \leftarrow 1$
$\boldsymbol{q} = (q(\sigma_i))_{i=1}^{|\Gamma|} \leftarrow (\frac{1}{|\Gamma|}, \ldots, \frac{1}{|\Gamma|})$
**while** $\Delta\alpha > \varepsilon$ **do**
    $\boldsymbol{q} \leftarrow \boldsymbol{q} - \eta \cdot \nabla_{\boldsymbol{q}}\alpha(\gamma^{\text{BR}}(\boldsymbol{q}))$
    $\boldsymbol{q} \leftarrow \text{ProjectSimplex}(\boldsymbol{q})$
    $\alpha_{\text{new}} \leftarrow \alpha(\gamma^{\text{BR}}(\boldsymbol{q})), \Delta\alpha \leftarrow |\alpha - \alpha_{\text{new}}|$
    $\alpha \leftarrow \alpha_{\text{new}}$
**end while**

Let $\text{BR}(S) := f(S)/g(S)$ denote the buy-to-rent ratio (BR) of a super item $S$. A key observation is that the smaller the BR, the earlier the super item should be bought (See Lemma 3), which can greatly reduce the number of possible purchase paths. Define $\Gamma = \{\gamma_1, \gamma_2, \ldots, \gamma_{|\Gamma|}\}$ as the set of all possible disjoint partitions of $\mathcal{M}$ and $\Sigma = \{\sigma_1, \sigma_2, \ldots, \sigma_{|\Gamma|}\}$ as the corresponding set of purchase paths, where $\sigma_i$ is obtained by sorting the super items in $\gamma_i$ in ascending order of BR values. Let $D$ denote the total number of possible super items[2], including all individual base items as well as combinations with bundle discounts. Denote these super items by $S_{[1]}, S_{[2]}, \ldots, S_{[D]}$, sorted in ascending order of BR, i.e., $\text{BR}(S_{[1]}) \leq \text{BR}(S_{[2]}) \leq \ldots \leq \text{BR}(S_{[D]})$.

Given set $\Sigma$ of purchase paths and the path selection probability $\boldsymbol{q}$, we define an augmented path as

$$\gamma^{\text{BR}}(\boldsymbol{q}) := (I(\hat{q}_{[1]}, S_{[1]}), I(\hat{q}_{[2]}, S_{[2]}), \ldots, I(\hat{q}_{[D]}, S_{[D]})), \tag{4}$$

where $I(\hat{q}_{[i]}, S_{[i]})$ denotes a new super item that comprises the same base items as $S_{[i]}$ but has a purchase price of $\hat{q}_{[i]} \cdot f(S_{[i]})$ and a rental rate of $\hat{q}_{[i]} \cdot g(S_{[i]})$, and $\hat{q}_{[i]} = \sum_{\sigma \in \Sigma: S_{[i]} \in \sigma} q(\sigma)$ is the sum of selection probabilities from paths that contain super item $S_{[i]}$. The construction of $\gamma^{\text{BR}}(\boldsymbol{q})$ equivalently transforms the path-level probability structure in the CSR problem into item-level decision information, enabling efficient decision-making under a fixed $\boldsymbol{q}$ (See Lemma 4).

Recall that competitive ratio of SOAC($\gamma^{\text{BR}}(\boldsymbol{q})$) is $\alpha(\gamma^{\text{BR}}(\boldsymbol{q}))$. Our problem reduces to find the path selection probability to minimize the competitive ratio, i.e., $\boldsymbol{q}^* = \arg\min_{\boldsymbol{q}} \alpha(\gamma^{\text{BR}}(\boldsymbol{q}))$. To achieve this, we employ the *Projected Gradient Descent* (PGD) method. At each iteration, we compute the gradient of the competitive ratio with respect to $\boldsymbol{q}$ and update $\boldsymbol{q}$ along the direction of the negative gradient. To ensure that $\boldsymbol{q}$ remains a valid probability distribution (i.e., $q(\sigma_i) \geq 0$ and $\sum_{i=1}^{|\Gamma|} q(\sigma_i) = 1$), we project the updated $\boldsymbol{q}$ onto the probability simplex.

**Example 2.** We demonstrate the SOAC algorithm with three items, rented daily at \$0.30, \$0.80, and \$0.50, and individually purchasable at \$149.99 each. Bundled purchases offer discounts: \$229.99 for any pair and \$329.99 for all three. The buy-to-rent ratios of all super items can be sorted as follows

$$\text{BR}(\{2,3\}) \leq \text{BR}(\{2\}) \leq \text{BR}(\{1,2,3\}) \leq \text{BR}(\{1,2\}) \leq \text{BR}(\{1,3\}) \leq \text{BR}(\{3\}) \leq \text{BR}(\{1\}).$$

We can find a total of 5 sorted purchase paths of the set $\{1, 2, 3\}$, where the super items in each path are arranged in ascending order of their buy-to-rent ratios.

Path 1-5: $(\{1,2,3\}), (\{1,2\}, \{3\}), (\{2\}, \{1,3\}), (\{2,3\}, \{1\}), (\{2\}, \{3\}, \{1\}).$

Let $q_i$ for $i \in \{1, 2, \ldots, 5\}$ denote the probability of selecting the $i$-th path. These probabilities are used to construct an augmented purchase path as follows:

$$\big(I(q_4, \{2,3\}), I(q_3{+}q_5, \{2\}), I(q_1, \{1,2,3\}), I(q_2, \{1,2\}), I(q_3, \{1,3\}), I(q_2{+}q_5, \{3\}), I(q_4{+}q_5, \{1\})\big).$$

---

[2]In practical applications, $D$ can be much smaller than $|2^{\mathcal{M}}|$ as not all combinations are eligible for discounts.

To determine the optimal probabilities $q_i$, we adopt projected gradient descent to minimize the competitive ratio of the augmented purchase path. The results show that only two paths have non-zero probabilities: Path 1 ($\{1, 2, 3\}$) with 70.4% and Path 4 ($\{2, 3\}, \{1\}$) with 29.6%. Using $q_i$, we compute the optimal randomized pur-

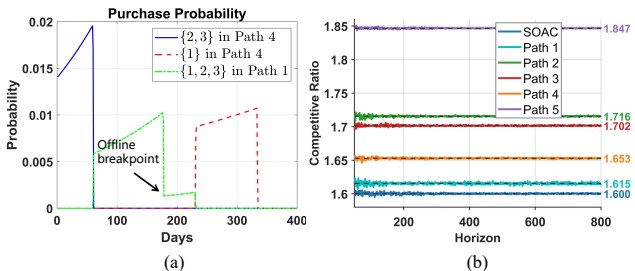

Figure 3: (Example 2) (a) Purchase probabilities along different paths. (b) Theoretical vs. empirical competitive ratios across varying horizons.

chase probabilities for each item per path per day, as illustrated in Figure 3(a). The offline breakpoint in Figure 3(a) arises because this point lies exactly at a transition point of the piecewise linear offline optimal function, where the slope of the function changes. Figure 3(b) compares the competitive ratio of SOAC with a baseline that fixes a single path. The results indicate that fixing the path leads to sub-optimal performance. Moreover, the average competitive ratio over varying horizons aligns well with the theoretical expected competitive ratio, demonstrating the accuracy of SOAC.

## 3.3 Theoretical Result for Competitive Algorithms

Our first main result is to show the optimality of SOAC for the CSR problem.

**Theorem 1.** *Given setup information $\mathcal{I}$, the SOAC algorithm attains the optimal competitive ratio among all online algorithms for the CSR problem.*

The competitive ratio of the CSR problem depends on the setup information, and exhibits no closed form expressions. Notably, the competitive ratio can exceed the well known ratio $e/(e-1)$ for the classic ski rental problem as shown in Figure 3. This is inherently due to risk of a fault path selection in the online algorithms, and there is a cost to switch across different paths when the purchases cannot be upgraded. This is also the key difference of CSR from other existing variants of ski rental problems. Furthermore, we can show that the CSR problem generalizes many well-established variants of the ski rental problems, including the *multi-shop ski rental* [1], *multi-slope ski rental* [19], *multi-commodity ski rental* [29] and CSR *with upgrading* [32] problems. Therefore, our proposed SOAC algorithm can also achieve optimal ratios for those variants.

**Lemma 1.** *The SOAC algorithm achieves the optimal competitive ratios for multi-shop, multi-slope, multi-commodity and CSR with upgrading ski rental problems.*

In fact, SOAC not only provides an alternative algorithm to previous variants, but also resolves some open problems. For example, in multi-commodity ski rental problem [29], prior work only achieves optimal solutions when all items have the same buy-to-rent ratio. The SOAC algorithm provides the first optimal randomized solution for arbitrary price configurations, filling this gap in the literature. A formal discussion related to generality of our model and algorithm is provided in Appendix E. We end this section by providing a proof sketch for Theorem 1, and its full proof is given in Appendix C.

**Proof sketch of Theorem 1.** To prove the optimality of SOAC, we first show that for a given purchase path $\sigma$, the optimal amortized cost strategy OAC($\sigma$) is optimal.

**Lemma 2** (Optimality of OAC). *For the CSR problem with setup information $\mathcal{I}$, given purchase path $\sigma$, OAC($\sigma$) achieves the optimal competitive ratio among all online algorithms.*

Next, we show that we can only focus on the purchase paths whose super items are arranged in increasing order of their buy-to-rent ratios. This greatly reduces the possible purchase paths.

**Lemma 3** (Optimal purchase order). *Given a disjoint partition $\gamma = \{S_1, S_2, \ldots, S_l\}$ of $\mathcal{M}$, OAC($\gamma^{\text{BR}}$) achieves the minimum competitive ratio among all purchase paths with the same super items in $\gamma$, where $\gamma^{BR} = (S_{[1]}, S_{[2]}, \ldots, S_{[l]})$ such that $\text{BR}(S_{[1]}) \leq \text{BR}(S_{[2]}) \leq \ldots \leq \text{BR}(S_{[l]})$.*

Building on Lemma 3, we can further prove that the optimal strategy for a CSR, with a given set of possible purchase paths and the associated path selection probability $\boldsymbol{q}$, is equivalent to a CSR with one augmented purchase path $\gamma^{\text{BR}}(\boldsymbol{q})$, which is defined in Eq. (4).

**Lemma 4** (Equivalent problem transformation). *Given the setup information $\mathcal{I}$ for* CSR*, given a purchase path probability $\boldsymbol{q}$, the* CSR *problem reduces to determining the optimal purchase probability $\boldsymbol{p}$ for the augmented purchase path $\gamma^{BR}(\boldsymbol{q})$.*

Given the path selection probability $\boldsymbol{q}$, SOAC uses $\mathtt{OAC}(\gamma^{\mathrm{BR}}(\boldsymbol{q}))$ to determine the purchase probability and obtain the corresponding competitive ratio $\alpha(\gamma^{\mathrm{BR}}(\boldsymbol{q}))$. Finally, SOAC obtains the optimal $\boldsymbol{q}^*$ using a PGD method. The optimality of $\boldsymbol{q}^*$ is ensured by proving the convexity of $\alpha(\gamma^{\mathrm{BR}}(\boldsymbol{q}))$ in $\boldsymbol{q}$.

**Lemma 5** (Convexity property). *The competitive ratio $\alpha(\gamma^{BR}(\boldsymbol{q}))$ of the optimal amortized cost strategy under the augmented path $\gamma^{BR}(\boldsymbol{q})$ is a convex function of the path selection probability $\boldsymbol{q}$.*

Combining all above results gives the proof Theorem 1. □

**Remark 1.** *The proposed algorithm* SOAC *establishes the first unified theoretical framework for various ski-rental variants, proving both the existence and computability of the optimal solution. Although the worst-case computational complexity of* SOAC *is exponential due to the exponential number of possible combinations, in practical scenarios only a limited number of combinations are relevant, which reduces the runtime to polynomial time. We provide a detailed runtime analysis in Appendix G.5. Moreover, for existing variants such as the multi-shop ski rental, multi-slope ski rental, and multi-commodity ski rental problems, the computational complexity of* SOAC *remains polynomial.*

## 4 Learning-Augmented Algorithms

Based on the optimal robust algorithm SOAC, this section continues to explore how machine-learned predictions can be leveraged to break the pessimistic worst-case bounds. Consider that in the beginning of a ski season, we obtain a prediction $y$ of the time horizon $T$. Our goal is to design a learning-augmented SOAC (LA-SOAC) that can provide the best possible consistency and robustness guarantees. In particular, let $A_{y;\lambda} := (\boldsymbol{q}_{y;\lambda}, \boldsymbol{p}_{y;\lambda})$ denote the algorithm for a given prediction $y$ and confidence parameter $\lambda$, where $\lambda \in (0,1)$ is a hyperparameter that indicates the confidence in the prediction (the smaller $\lambda$, the more confidence in the prediction). An algorithm $A_{y;\lambda}$ is $\mu$-consistent if $\mathtt{ALG}(T; A_{y;\lambda}) \le \mu \cdot \mathtt{OPT}(T)$ when the prediction is accurate (i.e., $y = T$), and $\beta$-robust if $\mathtt{ALG}(T; A_{y;\lambda}) \le \beta \cdot \mathtt{OPT}(T)$ for any actual end times $T$.

Recall that a key design in SOAC is to enforce the completion time $T_A$, at which the algorithm's cost stops to increase, to the critical time of offline optimal $T_{\mathtt{OPT}}$, at which the offline algorithm stops to increase. Our core idea of LA-SOAC is to dynamically adjust the completion time $T_\alpha$ based on the predicted value $y$. Intuitively, when the prediction $y$ is large, a smaller completion time $T_\alpha$ is preferred. However, the original SOAC fails if the completion time is set larger than $T_{\mathtt{OPT}}$, since the incremental cost $\Delta\mathtt{OPT}(n) = 0$ for $n > T_{\mathtt{OPT}}$, making it infeasible to track $\mathtt{OPT}(n)$ beyond this point. To address this problem, we introduce an augmented cost function $\overline{\mathtt{OPT}}(n)$. Let $t^* = \max\{n \in \mathbb{N}^+ \mid \Delta\mathtt{OPT}(n) \ge \mathtt{OPT}(T_{\mathtt{OPT}})/T_{\mathtt{OPT}}\}$. The increment of $\overline{\mathtt{OPT}}(n)$ is defined as $\Delta\overline{\mathtt{OPT}}(n) = \Delta\mathtt{OPT}(n)$ for $n \le t^*$, and $\Delta\overline{\mathtt{OPT}}(n) = \mathtt{OPT}(T_{\mathtt{OPT}})/T_{\mathtt{OPT}}$ for $n > t^*$. This modified cost ensures that the incremental cost of $\overline{\mathtt{OPT}}(n)$ is at least $\mathtt{OPT}(T_{\mathtt{OPT}})/T_{\mathtt{OPT}}$.

In LA-SOAC, we extend SOAC to $\overline{\mathtt{SOAC}}$, which replaces $\mathtt{OPT}(n)$ with $\overline{\mathtt{OPT}}(n)$ and changes the critical time from $T_{\mathtt{OPT}}$ to $T_{\mathtt{ML}}$. Let $\overline{\mathtt{SOAC}}(T_{\mathtt{ML}})$ denote the modified strategy (See Appendix F for full description). The LA-SOAC algorithm adjusts $T_{\mathtt{ML}}$ based on the prediction $y$: If $y \ge T_{\mathtt{OPT}}$, it prioritizes early purchases by setting $T_{\mathtt{ML}} = \lfloor \lambda T_{\mathtt{OPT}}^{(1)} \rfloor$, where the parameter $T_{\mathtt{OPT}}^{(1)} \le T_{\mathtt{OPT}}$. If $y < T_{\mathtt{OPT}}$, it delays purchases by setting $T_{\mathtt{ML}} = \lceil T_{\mathtt{OPT}}^{(2)}/\lambda \rceil$, where $T_{\mathtt{OPT}}^{(2)} \ge T_{\mathtt{OPT}}$. Using adjusted critical times, LA-SOAC computes purchase probabilities $A_{y;\lambda}$. The procedure is outlined in Algorithm 3.

---

**Algorithm 3** LA-SOAC: Learning-Augmented SOAC

---

1: **Input:** Setup information $\mathcal{I} = (\mathcal{M}, f, g)$; prediction $y$; parameters $T_{\mathtt{OPT}}^{(1)}, T_{\mathtt{OPT}}^{(2)}, \lambda$;
2: **if** $y \ge T_{\mathtt{OPT}}$ **then**
3:     Set $T_{\mathtt{ML}} \leftarrow \lfloor \lambda T_{\mathtt{OPT}}^{(1)} \rfloor$ and obtain $A_{y;\lambda}^{(1)} = (\boldsymbol{q}_{y;\lambda}^{(1)}, \boldsymbol{p}_{y;\lambda}^{(1)}) = \overline{\mathtt{SOAC}}(T_{\mathtt{ML}})$;
4: **else**
5:     Set $T_{\mathtt{ML}} \leftarrow \lceil T_{\mathtt{OPT}}^{(2)}/\lambda \rceil$ and obtain $A_{y;\lambda}^{(2)} = (\boldsymbol{q}_{y;\lambda}^{(2)}, \boldsymbol{p}_{y;\lambda}^{(2)}) = \overline{\mathtt{SOAC}}(T_{\mathtt{ML}})$.
6: **end if**

---

**Theorem 2.** *Given parameters $T_{OPT}^{(1)} \leq T_{OPT}$, $T_{OPT}^{(2)} \geq T_{OPT}$, and confidence factor $\lambda \in (0,1)$, let $A_{y;\lambda}^{(1)}$ and $A_{y;\lambda}^{(2)}$ denote purchase probabilities determined in Algorithm 3. Then the consistency of Algorithm 3 is* $\max\left\{\frac{\text{ALG}(T_{OPT}; A_{y;\lambda}^{(1)})}{\text{OPT}(T_{OPT})}, \frac{\text{ALG}(T_{OPT}; A_{y;\lambda}^{(2)})}{\text{OPT}(T_{OPT})}\right\}$ *and its robustness is* $\max\left\{\frac{\text{ALG}(\lfloor \lambda T_{OPT}^{(1)}\rfloor; A_{y;\lambda}^{(1)})}{\text{OPT}(\lfloor \lambda T_{OPT}^{(1)}\rfloor)}, \frac{\text{ALG}(\lceil T_{OPT}^{(2)}/\lambda\rceil; A_{y;\lambda}^{(2)})}{\text{OPT}(T_{OPT})}\right\}$.

Due to the complexity of the expected costs in the CSR problem, there are no closed-form expressions for consistency and robustness in general. Theorem 2 states that both consistency and robustness of LA-SOAC are dominated by two extreme ratios. Suppose the prediction is accurate, i.e., $y = T$. If $y \geq T_{OPT}$, the cost ratio $\frac{\text{ALG}(T; A_{T;\lambda}^{(1)})}{\text{OPT}(T)}$ is maximized at $T = T_{OPT}$; if $y < T_{OPT}$, the ratio $\frac{\text{ALG}(T; A_{T;\lambda}^{(2)})}{\text{OPT}(T)}$ is also maximized at $T = T_{OPT}$. Thus, consistency is dominated by the maximum of these two extreme ratios. In contrast, when the prediction $y$ may be arbitrarily inaccurate, the cost ratio peaks at $T = \lfloor \lambda T_{OPT}^{(1)}\rfloor$ if $y \geq T_{OPT}$, and at $T = \lceil T_{OPT}^{(2)}/\lambda\rceil$ if $y < T_{OPT}$. Note that $\text{OPT}(\lceil T_{OPT}^{(2)}/\lambda\rceil) = \text{OPT}(T_{OPT})$, so in the denominator we use $T = T_{OPT}$ for computing $\text{OPT}(T)$. Thus, robustness is governed by the maximum of these two extreme-case ratios.

Parameters $T_{OPT}^{(1)}$ and $T_{OPT}^{(2)}$ are set to $T_{OPT}$ in general, and can be specifically designed to optimize consistency and robustness. Specifically, in the classic ski rental problem, we can set $T_{OPT}^{(1)} = T_{OPT}^{(2)} = T_{OPT}$, and LA-SOAC recovers the explicit bounds [21], as shown in Corollary 1. For the multi-shop ski rental problem, by properly setting $T_{OPT}^{(1)}$ and $T_{OPT}^{(2)}$, LA-SOAC attains improved consistency and robustness bounds compared to previous results [25], as shown in Corollary 2.

**Corollary 1.** *For the classic ski rental problem with purchase price $b$ and rental price 1, when $T_{OPT}^{(1)} = T_{OPT}^{(2)} = T_{OPT}$ and $\lambda \in (\frac{1}{b}, 1]$, LA-SOAC is $\frac{\lambda}{1-e^{-\lambda}}$-consistent and $\frac{1+1/b}{1-e^{-(\lambda-1/b)}}$-robust.*

**Corollary 2.** *For the multi-shop ski rental problem with $s$ shops, where the purchase prices satisfy $b_1 > \cdots > b_s$ and the rental prices satisfy $1 = r_1 < \cdots < r_s$, when $T_{OPT}^{(1)} = T_{OPT}$, $T_{OPT}^{(2)} = b_1$, and $\lambda \in \left(\frac{1}{b_s}, 1\right]$, LA-SOAC guarantees a consistency ratio no worse than $\frac{r_s \lambda}{1-e^{-r_s \lambda}}$ and a robustness ratio no worse than $\frac{b_1}{b_s}\max\left\{\frac{r_s}{1-e^{-r_s(\lambda-1/b_s)}}, \frac{1/\lambda+1/b_1}{1-e^{-1/\lambda}}\right\}$, which are the bounds established in [25].*

## 5   Numerical Results

We empirically evaluate the performances of LA-SOAC. Consider three items, with item prices $b_1 = 80$, $b_2 = 110$, and $b_3 = 130$, and the same rental price of 1. The discount factor for purchasing any two items together is set to 0.95, and 0.9 for three items. We let the actual number of days, $T$, be uniformly distributed within the region $[1, 4T_{OPT}]$. The predicted number of days, $y$, is set to $y = T + \epsilon$, where the simulated error $\epsilon$ follows a normal distribution with mean $\delta$ and standard deviation $\eta$, i.e., $\epsilon \sim \mathcal{N}(\delta, \eta^2)$. Here, $\delta$ controls the bias of the prediction, while $\eta$ (referred to as the error parameter) characterizes the variability or uncertainty in the prediction. For each standard deviation, 10,000 samples are randomly sampled in this study, and their average

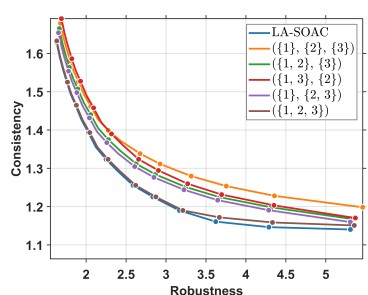

Figure 4: Consistency vs. robustness.

competitive ratios are calculated. Figure 4 compares the trade-off performance of the LA-SOAC algorithm against strategies that follow only a single path. The results demonstrate that LA-SOAC consistently outperforms the others.

Figures 5(a) and 5(b) further illustrate the average competitive ratio under varying levels of prediction error $\eta$. Specifically, Figure 5(a) presents results with $\delta = 0$ for different values of $\lambda \in \{0.25, 0.5, 0.75, 1\}$. When $\lambda$ is smaller, the average competitive ratio is lower under accurate predictions but higher under inaccurate predictions. Figure 5(b) evaluates performance across varying error levels $\delta \in \{50, 100, 150, 200\}$. For a fixed $\lambda$, the results indicate that the optimal choice of bias $\delta$ depends on the error parameter $\eta$. When $\eta$ is small, a smaller bias $\delta$ tends to yield better competitive ratios, as it allows the algorithm to more effectively exploit accurate predictions.

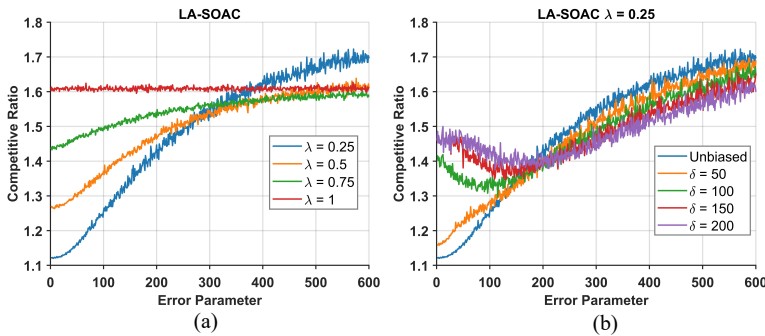

Figure 5: (a) Average numerical competitive ratio over various error parameters $\eta$ and $\lambda$. (b) Average numerical competitive ratio over varying error parameters $\eta$ and $\delta$.

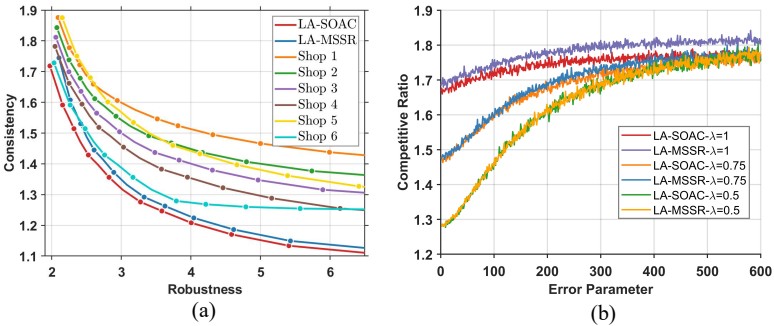

Figure 6: (Multi-shop ski rental) (a) Consistency vs. robustness. (b) Average competitive ratios.

In contrast, when $\eta$ is large, a larger bias $\delta$ becomes beneficial, as it helps hedge against extreme inaccuracies, thereby improving overall performance.

We compare our `LA-SOAC` algorithm with the state-of-the-art learning-augmented algorithm for multi-shop ski rental (Algorithm 3 in [25] referred to as `LA-MSSR`). The evaluation utilizes the dataset from [25], comprising 6 shops with purchase prices defined as $b_1 = 100, b_2 = 95, b_3 = 90, b_4 = 85, b_5 = 80, b_6 = 75$ and rental prices given by $r_1 = 1.00, r_2 = 1.05, r_3 = 1.10, r_4 = 1.15, r_5 = 1.20, r_6 = 1.25$. Consistent with [25], the actual number of days is modeled as uniformly distributed within the interval $[1, 3b_1]$. Figure 6(a) illustrates the performance trade-off of `LA-SOAC` compared to `LA-MSSR` [25] and a baseline strategy that follows a single path. The results demonstrate that `LA-SOAC` achieves a superior trade-off compared to [25]. Furthermore, Figure 6(b) compares the average competitive ratio, demonstrating that `LA-SOAC` offers a better trade-off and surpasses the performance of [25] in certain cases. For more numerical experiments about `SOAC` and `LA-SOAC`, see Appendix G. The source code for reproducing all experiments is available at `https://github.com/guodongsanjianke/Combinatorial_Ski_Rental_Problems`.

## 6 Conclusion

In this paper, we study robust and learning-augmented algorithms for the `CSR` problem. We first propose the `SOAC` algorithm, an optimal randomized algorithm for `CSR`, which can be extended to address many other variants of the ski rental problem. Building on this, we introduce a learning-augmented algorithm that provides both consistency and robustness guarantees, and can recover or improve existing results of learning-augmented algorithms in classic ski rental and multi-shop ski rental problems. Although the computational complexity of the `SOAC` algorithm can grow exponentially with the number of super items, this is an algorithmic limitation stemming from the combinatorial nature of the problem. In practical applications, however, the number of combinations is often limited, i.e., not all combinations are eligible for discounts. This can significantly reduce the computational complexity of the `SOAC` algorithm. In future work, we plan to apply our algorithms to real-world scenarios to investigate their practical impact.

## Acknowledgments

Lin Yang would like to thank the support from NSFC (No. 62306138), JiangsuNSF (No. BK20230784), and the Innovation Program of State Key Laboratory for Novel Software Technology at Nanjing University (Nos. ZZKT2024B15, ZZKT2025B25). Mohammad Hajiesmaili would like to thank the support from NSF CAREER (No. 2045641).

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

# Appendix

# A Related work

The classic ski rental problem was first introduced by [12], where optimal deterministic and randomized algorithms were developed, achieving competitive ratio upper bounds of 2 and $e/(e-1)$, respectively. The rent-or-buy dilemma in ski rental problem [17, 8] has a wide range of applications, including on/off scheduling of small cell base stations for energy saving with intermittent demands [15], request-response caching for cloud services [13], and bandwidth cost minimization in content delivery networks [20]. Additionally, it applies to purchasing multi-kind Bahncards to reduce ticket prices for future journeys, thereby lowering repeated travel costs [24].

Various variants [29, 28, 9] of the ski rental problem have since been proposed to address more complex application scenarios. The most relevant work [32] proposes a randomized algorithm for the CSR that allows for upgrades, achieving a competitive ratio of $e/(e-1)$ using a primal-dual framework. Their work addresses CSR with upgrading as the primal problem and combinatorial online bipartite matching as the dual problem. The *multi-slope ski rental* problem [19, 6, 10] is a variant that introduces a one-time setup cost and a usage-duration-dependent cost for each option, enabling customers to switch between options. This problem is further divided into two versions: additive and non-additive. The additive version allows buying costs to accumulate, while the non-additive version has arbitrary switching costs for each switch. In [19], an optimal randomized algorithm for the additive version and an $e$-competitive ratio for the non-additive version are proposed. [1] introduces the *multi-shop ski rental* problem, in which multiple shops offer the required item. Upon arrival at the ski field, customers must select a shop and decide whether to rent or purchase the item from that shop [25]. The authors derive closed-form solutions for this problem and develop a linear-time algorithm to compute these solutions efficiently.

Another variant is the *two-level ski rental* problem [28], which involves making decisions on individual or combined purchases of all items. They develop an optimal deterministic online algorithm that uses two fixed thresholds for decision-making when rental costs exceed predefined levels. Further, the authors [29] investigate the *multi-commodity ski rental* problem, which requires considering multiple purchase combinations without any discounts. They propose an online algorithm and demonstrate its optimality when items have the same rent-to-buy ratio. [31] study the *multi-discount ski rental* problem, which involves a single item with multiple rental options [22, 30], each with a rental duration and increasing discounts as the rental duration increases. They propose a 4-competitive ratio algorithm and observe that the competitive ratio increases with the number of available rental options.

In recent years, the integration of learning-based predictions into online algorithms has introduced a new paradigm known as learning-augmented online algorithms. These methods aim to balance two key metrics: consistency and robustness [27, 14, 23]. Consistency measures the algorithm's performance with perfect predictions, while robustness focuses on the algorithm's performance when the prediction is of low quality. Researchers have proposed several learning-augmented algorithms to tackle different variants of the ski rental problem [5, 25, 2]. For instance, [21] has demonstrated the high robustness of their machine-learning algorithm for the classic ski rental problem. Similarly, [25] has used machine learning algorithms to significantly improve the performance of their algorithm for the multi-shop ski rental problem. [2] have studied strategies for customizing machine learning algorithms by incorporating optimization objectives into the loss function. [11] have developed an algorithm that balances consistency with robustness for the ski rental problem with expert advice and established strict upper and lower bounds for the consistency ratio based on the number of experts.

# B Proof for Lemma 2

The Lemma 2 is established by Proposition 1, Proposition 2, and Proposition 3. Given a purchase path $\sigma$, we define a family of amortized cost strategies $\{\text{AC}(\sigma; \alpha)\}_{\alpha>1}$, parameterized by the target competitive ratio $\alpha$, and show that any optimal online algorithm belongs to this family (Proposition 1). We then establish that the completion time $T_A$ of the algorithm $A = \text{AC}(\sigma; \alpha)$, defined as the time at which all purchases are completed, is monotonically non-increasing with respect to $\alpha$ (Proposition 2). Furthermore, we prove that no $\text{AC}(\sigma; \alpha)$ algorithm can have a completion time $T_A$ exceeding $T_{\text{OPT}}$, the completion time of the optimal offline algorithm (Proposition 3). Combining these results, we conclude that the minimal competitive ratio among $\text{AC}(\sigma; \alpha)$ strategies is attained when the completion time equals $T_{\text{OPT}}$.

**Proposition 1.** *Given the information setup $\mathcal{I}$ of the CSR problem, among all online algorithms that follow the purchase path $\sigma$, the amortized cost strategies $\{AC(\sigma; \alpha)\}_{\alpha > 1}$ include the optimal online algorithm.*

**Proposition 2.** *Given information setup $\mathcal{I}$ of the CSR problem and a purchase path $\sigma$, if there exist algorithms $A_1 = AC(\sigma; \alpha_1)$ and $A_2 = AC(\sigma; \alpha_2)$ with $\alpha_1 < \alpha_2$, then $T_{A_2} \leq T_{A_1}$.*

**Proposition 3.** *Given a setup $\mathcal{I}$ for the CSR problem, a purchase path $\sigma$, and the algorithm $A = AC(\sigma; \alpha)$; there does not exist an $\alpha$ such that $T_A > T_{OPT}$.*

## B.1  Proof of Proposition 1

We begin by establishing Proposition 4, which demonstrates that for the CSR problem with $|\mathcal{M}| = 1$, purchasing the item earlier results in a lower cost when the game duration is long.

**Proposition 4.** *For a information setup $\mathcal{I} = (\mathcal{M}, f, g)$ of CSR, where $|\mathcal{M}| = 1$, let $k, u \in \mathbb{N}^+$ and $k < u$. Suppose $A_1$ and $A_2$ are two online algorithms for the problem satisfy $\Delta ALG(n;\ A_2) = \Delta ALG(n;\ A_1)$ for $n \neq k, u$ and $\Delta ALG(n;\ A_2) > \Delta ALG(n;\ A_1)$ for $n = k$. Then, we have $\Delta ALG(u;\ A_1) - \Delta ALG(u;\ A_2) > \Delta ALG(k;\ A_2) - \Delta ALG(k;\ A_1)$.*

*Proof.* Consider the set $S = \{1\}$, which is an element of $\mathcal{M}$. The cost incurred by the algorithm on day $n$ can be expressed as

$$\Delta ALG(T) = (1 - F_{n-1}(S)) \cdot g(S) + (f(S) - g(S)) \cdot p_n(S),$$

where the first term on the right-hand side represents the expected rental cost, and the second term represents the expected purchase cost. From the above expression, we can derive the purchase probability on day $n$ as $p_n(S) = \big(\Delta ALG(T) - (1 - F_{n-1}(S)) \cdot g(S)\big)/\big(f(S) - g(S)\big)$. Applying this formula to algorithms $A_1$ and $A_2$, we obtain the following observations

(1) When $i < k$ or $i > u$, we have $p_i(S; A_1) = p_i(S; A_2)$ and $F_i(S; A_1) = F_i(S; A_2)$.

(2) When $i = k$, it holds that $p_i(S; A_1) < p_i(S; A_2)$ and $F_i(S; A_1) < F_i(S; A_2)$.

(3) When $k < i < u$, similarly, $p_i(S; A_1) < p_i(S; A_2)$ and $F_i(S; A_1) < F_i(S; A_2)$.

As a result, we obtain

$$p_u(S; A_1) - p_u(S; A_2) > p_k(S; A_2) - p_k(S; A_1),$$

and therefore, the following inequality holds

$$
\begin{aligned}
&\Delta ALG(u; A_1) - \Delta ALG(u; A_2) \\
=\ & (F_{u-1}(S; A_2) - F_{u-1}(S; A_1)) \cdot g(S) + (f(S) - g(S)) \cdot (p_u(S; A_1) - p_u(S; A_2)) \\
>\ & (F_{k-1}(S; A_1) - F_{k-1}(S; A_2)) \cdot g(S) + (f(S) - g(S)) \cdot (p_k(S; A_2) - p_k(S; A_1)) \\
=\ & \Delta ALG(k; A_2) - \Delta ALG(k; A_1).
\end{aligned}
$$

This concludes the proof of Proposition 4. $\qquad \square$

For a purchase path $(S_1, S_2, \ldots, S_l)$, if the purchasing process follows the $AC(\sigma; \alpha)$ strategy, then any super item that lies along the incremental optimal offline cost trajectory satisfies the three properties stated in Remark 2. *Therefore, to establish Proposition 1, it suffices to show that the $AC(\sigma; \alpha)$ strategy includes the optimal strategy for purchasing a single super item*, provided that the incremental optimal offline cost associated with this super item satisfies the conditions specified in Remark 2.

**Remark 2.** *For a purchase path $(S_1, S_2, \ldots, S_l)$, if the purchasing strategy adheres to $AC(\sigma; \alpha)$, then any super item that lies along the incremental optimal offline cost function $\Delta OPT(T)$ satisfies the following three properties:*

(1) ***Non-negativity**: $\Delta OPT(T) \geq 0$, for all $T \in \mathbb{N}^+$.*

(2) ***Monotonicity**: $\Delta OPT(T)$ is non-increasing for $T > 1$.*

(3) ***Stability**: There exists a time $T_{OPT} \in \mathbb{N}^+$ such that $\Delta OPT(T) > 0$ for $T = T_{OPT}$, and $\Delta OPT(T) = 0$ for all $T \geq T_{OPT} + 1$.*

To facilitate the analysis, we assume that the purchase of a super item $S_i$ begins at time $n = 1$. Since part of the cost at $n = 1$ may be attributed to the purchase of a previous super item, the incremental optimal offline cost $\Delta\text{OPT}(T)$ is monotonically non-increasing for $n > 1$. To prove Proposition 1, it suffices to establish the following proposition (Proposition 5).

**Proposition 5.** *Consider a super item $S$ with a purchase price $f(S)$ and a rental rate $g(S)$. Suppose the optimal offline function for this super item satisfies the conditions outlined in Remark 2. Let the competitive ratio of algorithm $A_1$ be $\text{CR}(A_1)$. Then, there exists an algorithm $A_2$ that satisfies the $\text{AC}(\sigma; \alpha)$ strategy and whose competitive ratio satisfies $\text{CR}(A_2) \leq \text{CR}(A_1)$, i.e., algorithm $A_1$ is not superior to algorithm $A_2$.*

*Proof.* If algorithm $A_1$ satisfies the $\text{AC}(\sigma; \alpha)$ strategy, the proof is straightforward by taking $A_2 = A_1$.

However, if algorithm $A_1$ does *not* satisfy the $\text{AC}(\sigma; \alpha)$ strategy, then there must exist a day $k < T_{A_1}$ such that

$$\Delta\text{ALG}(k; A_1) < \text{CR}(A_1) \cdot \Delta\text{OPT}(k). \tag{5}$$

Without loss of generality, assume that $k$ is the *first* day for which the above inequality holds. To make the equality in Eq. (5) hold, we can consider *shifting* the purchase probability: increase it on day $k$ and decrease it after day $k$. There are two possible cases when making this adjustment.

**Case 1:** Reducing the purchase probability after day $k$ is *not sufficient* to achieve the equality. This means that even if we increase the purchase probability on day $k$ to the maximum extent (i.e., make the cumulative purchase probability reach 1 by day $k$), the cost incurred by the algorithm on day $k$ still does not reach the required budget. Mathematically, this implies the following inequality holds

$$\frac{\text{CR}(A_1) \cdot \Delta\text{OPT}(k) - (1 - F_{k-1}(S; A_1)) \cdot g(S)}{f(S) - g(S)} \geq 1 - F_{k-1}(S; A_1). \tag{6}$$

Eq. (6) can be simplified to $(1 - F_{k-1}(S; A_1)) \cdot f(S) < \text{CR}(A_1) \cdot \Delta\text{OPT}(k)$.

Based on this, we can construct a new algorithm $A_2$ with modified purchase probabilities as follows

$$p_n(S; A_2) = \begin{cases} p_n(S; A_1), & \text{if } n < k, \\ 1 - F_{n-1}(S; A_1), & \text{if } n = k, \\ 0, & \text{if } n > k. \end{cases}$$

With this construction, algorithm $A_2$ satisfies the $\text{AC}(\sigma; \alpha)$ strategy, and we also ensure that $\text{CR}(A_2) \leq \text{CR}(A_1)$.

**Case 2:** Suppose it is possible to make the equality in Eq. (5) hold on day $k$ by reducing the purchase probabilities after day $k$. This implies that

$$(1 - F_{k-1}(S; A_1)) \cdot f(S) \geq \text{CR}(A_1) \cdot \Delta\text{OPT}(k).$$

We then construct a modified algorithm $A^{(1)}$ with the following purchase probability allocation

(1) $p_n(S; A^{(1)}) = p_n(S; A_1)$, for $n < k$ or $n > k + \tau$.

(2) $p_n(S; A^{(1)}) = 0$, for $k + 1 \leq n \leq k + \tau - 1$.

(3) $p_n(S; A^{(1)}) = \frac{\text{CR}(A_1) \cdot \Delta\text{OPT}(k) - (1 - F_{k-1}(S; A_1)) \cdot g(S)}{f(S) - g(S)}$, for $n = k$.

(4) $p_n(S; A^{(1)}) = \sum_{j=k+1}^{k+\tau} p_j(S; A_1) - \left(p_k(S; A^{(1)}) - p_k(S; A_1)\right)$, for $n = k + \tau$.

The integer $\tau$ is chosen to ensure the feasibility of the probability adjustment, satisfying

$$\sum_{j=k+1}^{k+\tau-1} p_j(S; A_1) < p_k(S; A^{(1)}) - p_k(S; A_1) \text{ and } \sum_{j=k+1}^{k+\tau} p_j(S; A_1) \geq p_k(S; A^{(1)}) - p_k(S; A_1).$$

Algorithm $A^{(1)}$ is valid since it satisfies the following conditions

$$\sum_{n=1}^{\infty} p_n(S; A^{(1)}) = 1, \quad \text{and} \quad 0 \le p_n(S; A^{(1)}) \le 1.$$

We now proceed to prove that $\text{CR}(A^{(1)}) \le \text{CR}(A_1)$ case by case.

*Subcase* (1): for $n < k$.

According to the construction of algorithm $A^{(1)}$, we have

$$\Delta\text{ALG}(n; A^{(1)}) = \Delta\text{ALG}(n; A_1) = \text{CR}(A_1) \cdot \Delta\text{OPT}(n), \quad \text{for } n < k.$$

Therefore, it follows that

$$\text{ALG}(n; A^{(1)}) = \text{CR}(A_1) \cdot \text{OPT}(n), \quad \text{for } n < k.$$

*Subcase* (2): for $n = k$.

$$\Delta\text{ALG}(k; A^{(1)}) = \left(1 - F_{k-1}(S; A^{(1)})\right) \cdot g(S) + (f(S) - g(S)) \cdot p_k(S; A^{(1)}) = \text{CR}(A_1) \cdot \Delta\text{OPT}(k).$$

Thus, we obtain $\text{ALG}(k; A^{(1)}) \le \text{CR}(A_1) \cdot \text{OPT}(k)$.

*Subcase* (3): for $n = k + \tau$.

According to Proposition 4, reallocating costs from later stages to earlier ones results in lower total costs. Therefore, we derive

$$\text{ALG}(k + \tau; A^{(1)}) < \text{ALG}(k + \tau; A_1) \le \text{CR}(A_1) \cdot \text{OPT}(k + \tau).$$

*Subcase* (4): for $n > k + \tau$.

In this case, we have

$$\text{ALG}(n; A^{(1)}) = \sum_{j=1}^{n} \Delta\text{ALG}(j; A^{(1)}) = \text{ALG}(k + \tau; A^{(1)}) + \sum_{j=k+\tau+1}^{n} \Delta\text{ALG}(j; A^{(1)})$$

$$\le \text{ALG}(k + \tau; A_1) + \sum_{j=k+\tau+1}^{n} \Delta\text{ALG}(j; A_1)$$

$$\le \text{CR}(A_1) \cdot \text{OPT}(n), \quad \text{for } n > k + \tau.$$

*Subcase* (5): for $k < n < k + \tau$.

We prove that $\text{ALG}(n; A^{(1)}) \le \text{CR}(A_1) \cdot \text{OPT}(n)$ by contradiction.

Assume there exists some $n$ with $k < n < k + \tau$ such that $\text{ALG}(n; A^{(1)}) > \text{CR}(A_1) \cdot \text{OPT}(n)$.

Without loss of generality, let $n$ be the smallest integer greater than $k$ for which the inequality fails. Therefore, we have

$$\text{ALG}(n - 1; A^{(1)}) \le \text{CR}(A_1) \cdot \text{OPT}(n - 1),$$
$$\Delta\text{ALG}(n; A^{(1)}) > \text{CR}(A_1) \cdot \Delta\text{OPT}(n).$$

According to the construction of $A^{(1)}$, we know

$$\Delta\text{ALG}(n; A^{(1)}) = \Delta\text{ALG}(n + 1; A^{(1)}) = \cdots = \Delta\text{ALG}(k + \tau - 1; A^{(1)}) \le \Delta\text{ALG}(k + \tau; A^{(1)}).$$

By the monotonicity of $\Delta\text{OPT}$, it holds that

$$\Delta\text{OPT}(n) \ge \Delta\text{OPT}(n + 1) \ge \cdots \ge \Delta\text{OPT}(k + \tau).$$

Hence, for all $i$ such that $n \le i \le k + \tau$, we have $\Delta\text{ALG}(i; A^{(1)}) > \text{CR}(A_1) \cdot \Delta\text{OPT}(i)$, which leads to $\text{ALG}(k + \tau; A^{(1)}) > \text{CR}(A_1) \cdot \text{OPT}(k + \tau)$. This contradicts the conclusion $\text{ALG}(k + \tau; A^{(1)}) \le \text{CR}(A_1) \cdot \text{OPT}(k + \tau)$. Therefore, it hold that $\text{ALG}(n; A^{(1)}) \le \text{CR}(A_1) \cdot \text{OPT}(n)$, for $k < n < k + \tau$.

In summary, we have shown that the competitive ratio of algorithm $A^{(1)}$ does not exceed that of $A_1$, i.e.,

$$\mathtt{CR}(A^{(1)}) \leq \mathtt{CR}(A_1).$$

By iteratively applying the above procedure, we can construct a sequence of algorithms $A^{(1)}, A^{(2)}, \ldots, A^{(d)}$, where the final algorithm $A^{(d)}$ satisfies the $\mathtt{AC}\,(\sigma; \alpha)$ strategy. These algorithms satisfy the following conditions

$$\mathtt{CR}(A_1) \geq \mathtt{CR}(A^{(1)}) \geq \mathtt{CR}(A^{(2)}) \geq \cdots \geq \mathtt{CR}(A^{(d)}).$$

Let $A_2 = A^{(d)}$. Then, $A_2$ satisfies $\mathtt{AC}\,(\sigma; \alpha)$ strategy and $\mathtt{CR}(A_2) \leq \mathtt{CR}(A_1)$. Therefore, Proposition 5 is proved.

$\square$

Given the result in Proposition 1, we now proceed to establish Proposition 2 and 3. Proposition 2 shows that achieving a smaller target competitive ratio requires a longer duration to complete the purchasing process. Proposition 3 demonstrate that an $\mathtt{AC}(\sigma; \alpha)$ algorithm with a completion time exceeding $T_{\mathtt{OPT}}$ does not exist.

## B.2 Proof of Proposition 2.

*Proof.* If algorithm $A$ buys only $S_i$ on day $n$ following purchase path $\sigma$, then on that day, only the super item $S_i$ has a non-zero purchase probability, while all other super items have a purchase probability of $0$. Consequently, the cost incurred by algorithm $A$ on day $n$ is given by

$$\Delta\mathtt{ALG}(n; A) = f(S_i) \cdot p_n(S_i; A) + (1 - F_n(S_i; A)) \cdot g(S_i) + \sum\nolimits_{j=i+1}^{l} g(S_j). \tag{7}$$

If algorithm $A$ is able to completely purchase $S_i$ on day $n$ and proceed to the next super item, then set $p_n(S_i; A) = 1 - F_{n-1}(S_i; A)$ and let

$$\Delta\mathtt{ALG}(n; A) = \Delta\mathtt{ALG}(n; A) - f(S_i) \cdot p_n(S_i; A). \tag{8}$$

Since the purchase path is fixed and identical for both algorithms $A_1$ and $A_2$, it is sufficient to demonstrate the conclusion for $|\mathcal{M}| = 1$ based on Eq. (7) and Eq. (8). Consider the set $S = \{1\}$. The purchase probability on day $n$ can be expressed as

$$p_n(S) = \frac{\Delta\mathtt{ALG}(T) - (1 - F_{n-1}(S)) \cdot g(S)}{f(S) - g(S)}.$$

Since $\alpha_1 < \alpha_2$, it follows that $\Delta\mathtt{ALG}(T; A_1) < \Delta\mathtt{ALG}(T; A_2)$. Consequently, we have $p_n(S; A_1) < p_n(S; A_2)$ for $n \leq T_{A_2}$. Algorithms $A_1$ and $A_2$ satisfy the following conditions

$$\sum\nolimits_{i=1}^{T_{A_1}} p_i(S; A_1) = 1; \quad \sum\nolimits_{i=1}^{T_{A_2}} p_i(S; A_2) = 1.$$

Thus, $T_{A_2} \leq T_{A_1}$. $\square$

## B.3 Proof of Proposition 3.

*Proof.* Consider the purchase path $\sigma = (S_1, S_2, \ldots, S_l)$. We use a proof by contradiction. Suppose that the completion time of algorithm $A$ exceeds $T_{\mathtt{OPT}}$. In this case, we have $F_{T_{\mathtt{OPT}}}(S_l; A) < 1$. Let $n = T_{\mathtt{OPT}} + 1$,

$$\begin{aligned}
\Delta\mathtt{ALG}(n; A) = {} & \sum_{i \in [l-1]} \left[ f(S_i) \cdot p_n(S_i; A) + (1 - F_n(S_i; A)) \cdot g(S_i) \right] \\
& + f(S_l) \cdot p_n(S_l; A) + (1 - F_n(S_l; A)) \cdot g(S_l) \\
= {} & \underbrace{\sum_{i \in [l-1]} \left[ f(S_i) \cdot p_n(S_i; A) + (1 - F_n(S_i; A)) \cdot g(S_i) \right]}_{\text{Term 1}} \\
& + \underbrace{(f(S_l) - g(S_l)) \cdot p_n(S_l; A)}_{\text{Term 2}} + \underbrace{(1 - F_{T_{\mathtt{OPT}}}(S_l; A)) \cdot g(S_l)}_{\text{Term 3}}.
\end{aligned}$$

Since Term 1 $\geq 0$, Term 2 $\geq 0$ and Term 3 $> 0$, we have $\Delta\mathtt{ALG}(n; A) > 0$, while $\alpha\Delta\mathtt{OPT}(n) = 0$. This implies $\Delta\mathtt{ALG}(n; A) > \alpha\Delta\mathtt{OPT}(n)$, which contradicts the $\mathtt{AC}(\sigma; \alpha)$ strategy. Thus, Proposition 3 holds. $\qquad\square$

## C Proof for Theorem 1

The proof of Theorem 1 is established through Lemma 2, Lemma 3, 4 and 5. First, Lemma 2 establishes that for a given purchase path $\sigma$, the strategy $\mathtt{OAC}(\sigma)$ achieves optimality. For a given purchase set, Lemma 3 guarantees the optimality of the purchase order. Building on this result, Lemma 4 further demonstrates that the optimal strategy for a $\mathtt{CSR}$ with a given purchase probability $q$ is equivalent to a $\mathtt{CSR}$ characterized by an augmented purchase path $\gamma^{\mathtt{BR}}(q)$. This augmented path $\gamma^{\mathtt{BR}}(q)$ is constructed based on the purchase probability vector $q$ and the purchase path set $\Sigma$. Consequently, the strategy $\mathtt{OAC}(\gamma^{\mathtt{BR}}(q))$ is guaranteed to be optimal for a $\mathtt{CSR}$ with purchase probability $q$. Moreover, Lemma 5 establishes that the competitive ratio $\alpha(\gamma^{\mathtt{BR}}(q))$ is a convex function of the purchase probability $q$. This convexity property allows the efficient computation of the optimal purchase probability $q^*$ using numerical optimization methods.

### C.1 Proof of Lemma 3.

To evaluate the performance of the algorithm under disjoint sets $\gamma$, including randomized paths, we consider strategies satisfying

$$
\begin{aligned}
\Delta\mathtt{ALG}(n; A) &= \alpha \cdot \Delta\mathtt{OPT}(n), \quad \text{when } n \leq T_{\mathtt{OPT}}, \\
\Delta\mathtt{ALG}(n; A) &= 0, \quad \text{when } n > T_{\mathtt{OPT}},
\end{aligned}
\tag{9}
$$

which are not necessarily constrained to a fixed purchase order. Define a strategy $A_1$ such that $\sum_{n=1}^{T_{\mathtt{OPT}}} p_n(S_v; A_1) = 1$ and $\Delta\mathtt{ALG}(n; A_1) = \alpha_1 \cdot \Delta\mathtt{OPT}(n)$, for any day $n \in \mathbb{N}^+$ and $v \in [l]$. Suppose strategy $A$ does not follow an ascending order of the buy-to-rent ratio. That is, there exist times $k$, $u$, and super items $S_i$, $S_j$ such that $k < u$, $\frac{f(S_i)}{g(S_i)} > \frac{f(S_j)}{g(S_j)}$, and $p_k(S_i; A_1) > 0, p_u(S_j; A_2) > 0$. Let $0 < \delta_1 \leq p_k(S_i; A_1)$ and $\delta_2 = \frac{f(S_i)-g(S_i)}{f(S_j)-g(S_j)}\delta_1 > 0$. Define a strategy $A_2$ such that

$$
p_n(S_v; A_2) = \begin{cases}
p_n(S_v; A_1), & \text{for } n \neq k, u \text{ and } v \in [l], \\
p_n(S_v; A_1), & \text{for } n = k, u \text{ and } v \in [l], \ v \neq i, j, \\
p_k(S_i; A_1) - \delta_1, \ p_k(S_j; A_1) + \delta_2, & \text{for } n = k, \\
p_u(S_i; A_1) + \delta_1, \ p_u(S_j; A_1) - \delta_2, & \text{for } n = u.
\end{cases}
$$

Then the strategy $A_2$ is well-defined, i.e., $\sum_{n=1}^{T_{\mathtt{OPT}}} p_n(S_i; A_2) = 1$, and $\Delta\mathtt{ALG}(k; A_2) = \alpha_1 \cdot \Delta\mathtt{OPT}(k)$. we will prove that strategy $A_2$ outperforms strategy $A_1$, i.e., $\Delta\mathtt{ALG}(n; A_2) \leq \alpha_1 \cdot \Delta\mathtt{OPT}(n)$ for any day $n \in \mathbb{N}^+$.

(1) For $n \leq k$, it is evident that $\Delta\mathtt{ALG}(n; A_2) = \alpha_1 \cdot \Delta\mathtt{OPT}(n)$.

(2) For $k < n < u$,

$$
\begin{aligned}
&\Delta\mathtt{ALG}(n; A_1) - \Delta\mathtt{ALG}(n; A_2) \\
&= (1 - F_n(S_i; A_1))\, g(S_i) + f(S_i)p_n(S_i; A_1) + (1 - F_n(S_j; A_1))\, g(S_j) + f(S_j)p_n(S_j; A_1) \\
&\quad - (1 - F_n(S_i; A_2))\, g(S_i) - f(S_i)p_n(S_i; A_2) - (1 - F_n(S_j; A_2))\, g(S_j) - f(S_j)p_n(S_j; A_2) \\
&= -\delta_1 g(S_i) + \delta_2 g(S_j) \\
&= \frac{(f(S_i) - g(S_i))\, g(S_j) - (f(S_j) - g(S_j))\, g(S_i)}{f(S_j) - g(S_j)}\delta_1.
\end{aligned}
$$

Since $\frac{f(S_i)}{g(S_i)} > \frac{f(S_j)}{g(S_j)}$, it follows that $\frac{f(S_i)-g(S_i)}{f(S_j)-g(S_j)} > \frac{g(S_i)}{g(S_j)}$. Thus, $\Delta\mathtt{ALG}(n; A_1) - \Delta\mathtt{ALG}(n; A_2) > 0$, implying $\Delta\mathtt{ALG}(n; A_2) < \Delta\mathtt{ALG}(n; A_1) = \alpha_1 \cdot \Delta\mathtt{OPT}(n)$.

(3) For $n = u$, similar to (2), we derive the following:

$$
\Delta\mathtt{ALG}(n; A_1) - \Delta\mathtt{ALG}(n; A_2) = \frac{(f(S_i) - g(S_i)) \cdot f(S_j) - (f(S_j) - g(S_j)) \cdot f(S_i)}{f(S_j) - g(S_j)} \cdot \delta_1.
$$

Given that $\frac{f(S_i)}{g(S_i)} > \frac{f(S_j)}{g(S_j)}$, it follows that $\frac{\left(\frac{g(S_i)}{f(S_i)}-1\right)}{\left(\frac{g(S_j)}{f(S_j)}-1\right)} > 1$. By multiplying both numerator and denominator by $f(S_i) \cdot f(S_j)$, we obtain

$$\frac{(f(S_i) - g(S_i)) \cdot f(S_j)}{(f(S_j) - g(S_j)) \cdot f(S_i)} > 1.$$

Thus, $\Delta\text{ALG}(n; A_1) - \Delta\text{ALG}(n; A_2) > 0$, which implies

$$\Delta\text{ALG}(n; A_2) < \Delta\text{ALG}(n; A_1) = \alpha_1 \cdot \Delta\text{OPT}(n).$$

(4) For $n > u$, it is evident that $\Delta\text{ALG}(n; A_2) = \alpha_1 \cdot \Delta\text{OPT}(n)$.

In summary, strategy $A_2$ outperforms strategy $A_1$, i.e., $\Delta\text{ALG}(n; A_2) \leq \alpha_1 \cdot \Delta\text{OPT}(n)$, for any $n \in \mathbb{N}^+$. Subsequently, by adjusting strategy $A_2$ to move the purchase sequence forward according to the order of probabilities until $\Delta\text{ALG}(n; A_2) = \alpha_2 \cdot \Delta\text{OPT}(n)$, it follows from Proposition 5 that $\alpha_2 \leq \alpha_1$. By iteratively applying this method, the optimal strategy is obtained by arranging items in ascending order of the values $f(S_i)/g(S_i)$. This completes the proof of Lemma 3.

### C.2   Proof of Lemma 4.

Given $\boldsymbol{q}$, the CSR problem of designing the optimal randomized algorithm to minimize the competitive ratio can be formulated according to Eq. (1) as follows

$$\min_{\alpha \geq 1, \boldsymbol{p}} \quad \alpha$$
$$\text{s.t.} \sum_{n=1}^{T}\sum_{i=1}^{|\Gamma|}\sum_{S \in \gamma_i}[q(\sigma_i)f(S) \cdot p_n(S \mid \sigma_i) + (1 - F_n(S \mid \sigma_i)) \cdot q(\sigma_i)g(S)] \leq \alpha \cdot \text{OPT}(T), \ \forall T \in \mathbb{N}^+$$

$$(10)$$

Since $\cup_{i=1}^{|\Gamma|}\{q(\sigma_i)\gamma_i\} = \gamma(\boldsymbol{q})$, where $q(\sigma_i)\gamma_i = \{q(\sigma_i)S\}_{S \in \gamma_i}$, Eq. (10) can be reformulated as

$$\min_{\alpha \geq 1, \boldsymbol{p}} \quad \alpha, \quad \text{s.t.} \sum_{n=1}^{T}\sum_{S \in \gamma(\boldsymbol{q})}[f(S) \cdot p_n(S) + (1 - F_n(S)) \cdot g(S)] \leq \alpha \cdot \text{OPT}(T), \ \forall T \in \mathbb{N}^+, \quad (11)$$

where $p_n(S)$ represents the probability of purchasing super item $S$ at time $n$, and $F_n(S) = \sum_{i=1}^{n}p_n(S)$.

Given a purchase probability $\boldsymbol{q} = \big(q(\sigma_1), q(\sigma_2), \ldots, q(\sigma_{|\Gamma|})\big)$, Eq. (11) indicates that the problem reduces to determining the optimal purchase probability $\boldsymbol{p}$ for the set $\gamma(\boldsymbol{q})$. According to Lemma 3, the optimal purchase path is the augmented purchase path $\gamma^{\text{BR}}(\boldsymbol{q})$. Thus, $\text{OAC}(\gamma^{\text{BR}}(\boldsymbol{q}))$ is optimal for a given purchase path probability $\boldsymbol{q}$, proving Lemma 4.

### C.3   Proof of Lemma 5.

For all purchase path probability $\boldsymbol{x}, \boldsymbol{y} \in [0,1]^{|\Gamma|}$, where $\boldsymbol{x} = \big(x(\sigma_1), x(\sigma_2), \ldots, x(\sigma_{|\Gamma|})\big)$ and $\boldsymbol{y} = \big(y(\sigma_1), y(\sigma_2), \ldots, y(\sigma_{|\Gamma|})\big)$ satisfy $\sum_{i=1}^{|\Gamma|} x(\sigma_i) = 1$ and $\sum_{i=1}^{|\Gamma|} y(\sigma_i) = 1$. Let algorithm $A = \text{OAC}(\gamma^{\text{BR}}(\boldsymbol{x}))$ and algorithm $B = \text{OAC}(\gamma^{\text{BR}}(\boldsymbol{y}))$, where $\gamma^{\text{BR}}(\boldsymbol{x}) = (I(\hat{x}_{[1]}, S_{[1]}), I(\hat{x}_{[2]}, S_{[2]}), \ldots, I(\hat{x}_{[D]}, S_{[D]}))$ and $\gamma^{\text{BR}}(\boldsymbol{y}) = (I(\hat{y}_{[1]}, S_{[1]}), I(\hat{y}_{[2]}, S_{[2]}), \ldots, I(\hat{y}_{[D]}, S_{[D]}))$. $\hat{x}_{[i]} = \sum_{\sigma \in \Sigma: S_{[i]} \in \sigma} x(\sigma)$ and $\hat{y}_{[i]} = \sum_{\sigma \in \Sigma: S_{[i]} \in \sigma} y(\sigma)$ is the sum of selection probabilities from paths that contain super item $S_{[i]}$. The competitive ratios are $\alpha_1 = \alpha(\gamma^{\text{BR}}(\boldsymbol{x}))$ and $\alpha_2 = \alpha(\gamma^{\text{BR}}(\boldsymbol{y}))$.

$\forall \ t \in [0,1]$, let $\boldsymbol{z} = t\boldsymbol{x} + (1-t)\boldsymbol{y} := \big(z(\sigma_1), z(\sigma_2), \ldots, z(\sigma_{|\Gamma|})\big)$. Define algorithm $C = \text{OAC}(\gamma^{\text{BR}}(\boldsymbol{z}))$, where $\gamma^{\text{BR}}(\boldsymbol{z}) = (I(\hat{z}_{[1]}, S_{[1]}), I(\hat{z}_{[2]}, S_{[2]}), \ldots, I(\hat{z}_{[D]}, S_{[D]}))$ and $\hat{z}_{[i]} = \sum_{\sigma \in \Sigma: S_{[i]} \in \sigma} z(\sigma)$ is the sum of selection probabilities from paths that contain super item $S_{[i]}$.. We will prove that $\alpha(\gamma^{\text{BR}}(\boldsymbol{z})) \leq \alpha_1 + (1-t)\alpha_2$.

Define algorithm $C_1$ purchase probability is $\boldsymbol{z} = t\boldsymbol{x} + (1-t)\boldsymbol{y}$ and the purchase probability of the super item $I(\hat{z}_{[i]}, S_{[i]})$ $(i = 1, 2 \ldots D)$ on day $n$ is, for any $i = 1, 2, \ldots, D$,

$$p_n\left(I(\hat{z}_{[i]}, S_{[i]}); C_1\right) = \frac{t\,\hat{x}_{[i]}}{t\,\hat{x}_{[i]} + (1-t)\,\hat{y}_{[i]}} \cdot p_n\left(I(\hat{x}_{[i]}, S_{[i]}); A\right)$$
$$+ \frac{(1-t)\,\hat{y}_{[i]}}{t\,\hat{x}_{[i]} + (1-t)\,\hat{y}_{[i]}} \cdot p_n\left(I(\hat{y}_{[i]}, S_{[i]}); B\right),$$

where $p_n\left(I(\hat{z}_{[i]}, S_{[i]}); C_1\right)$ denotes the probability that algorithm $A$ buys $I(\hat{z}_{[i]}, S_{[i]})$ on day $n$. Consequently, the cumulative distribution function of the super item $I(\hat{z}_{[i]}, S_{[i]})$ on $n$ days is, for any $i = 1, 2, \ldots, D$,

$$F_n\left(I(\hat{z}_{[i]}, S_{[i]}); C_1\right) = \frac{t\,\hat{x}_{[i]}}{t\,\hat{x}_{[i]} + (1-t)\,\hat{y}_{[i]}} F_n\left(I(\hat{x}_{[i]}, S_{[i]}); A\right)$$
$$+ \frac{(1-t)\,\hat{y}_{[i]}}{t\,\hat{x}_{[i]} + (1-t)\,\hat{y}_{[i]}} F_n\left(I(\hat{y}_{[i]}, S_{[i]}); B\right).$$

Since $p_n\left(I(\hat{x}_{[i]}, S_{[i]}); A\right), p_n\left(I(\hat{y}_{[i]}, S_{[i]}); B\right) \in [0, 1]$ and $F_{T_{\alpha_1}}\left(I(\hat{x}_{[i]}, S_{[i]}); A\right) = 1$, $F_{T_{\alpha_2}}\left(I(\hat{y}_{[i]}, S_{[i]}); B\right) = 1$, we have $p_n\left(I(\hat{z}_{[i]}, S_{[i]}); C_1\right) \in [0, 1]$ and $F_{\max\{T_{\alpha_1}, T_{\alpha_2}\}}\left(I(\hat{z}_{[i]}, S_{[i]}); C_1\right) = 1$. Therefore, this definition of algorithm $C_1$ is reasonable. The daily cost for algorithm $C_1$ is

$$\Delta\texttt{ALG}(n; C_1) = \sum_{i=1}^{D}\left((1 - F_n(I(\hat{z}_{[i]}, S_{[i]}); C_1))\hat{z}_{[i]} \cdot g(S_{[i]}) + \hat{z}_{[i]} \cdot f(S_{[i]}) \cdot p_n(I(\hat{z}_{[i]}, S_{[i]}); C_1)\right)$$

$$= \sum_{i=1}^{D} t\hat{x}_{[i]} \cdot \left((1 - F_n(I(\hat{x}_{[i]}, S_{[i]}); A)) \cdot g(S_{[i]}) + f(S_{[i]}) \cdot p_n(I(\hat{x}_{[i]}, S_{[i]}); A)\right)$$

$$+ \sum_{i=1}^{D}(1-t)\hat{y}_{[i]} \cdot \left((1 - F_n(I(\hat{y}_{[i]}, S_{[i]}); B)) \cdot g(S_{[i]}) + f(S_{[i]}) \cdot p_n(I(\hat{y}_{[i]}, S_{[i]}); B)\right).$$

Based on the definitions of algorithm $A$ and algorithm $B$, we have

$$\sum_{i=1}^{D}\left((1 - F_n(I(\hat{x}_{[i]}, S_{[i]}); A))\hat{x}_{[i]} \cdot g(S_{[i]}) + \hat{x}_{[i]} \cdot f(S_{[i]}) \cdot p_n(I(\hat{x}_{[i]}, S_{[i]}); A)\right) = \alpha_1 \cdot \Delta\texttt{OPT}(n) \tag{12}$$

$$\sum_{i=1}^{D}\left((1 - F_n(I(\hat{y}_{[i]}, S_{[i]}); B))\hat{y}_{[i]} \cdot g(S_{[i]}) + \hat{y}_{[i]} \cdot f(S_{[i]}) \cdot p_n(I(\hat{y}_{[i]}, S_{[i]}); B)\right) = \alpha_2 \cdot \Delta\texttt{OPT}(n) \tag{13}$$

By multiplying both sides of Eq. (12) by $t$ and both sides of Eq. (13) by $1 - t$, and then adding the resulting equations, we obtain

$$\Delta\texttt{ALG}(n; C_1) = (t\alpha_1 + (1-t)\alpha_2) \cdot \Delta\texttt{OPT}(n). \tag{14}$$

Thus, algorithm $C_1$ achieves a competitive ratio of $t\alpha_1 + (1-t)\alpha_2$ with a purchase probability $\boldsymbol{z} = t\boldsymbol{x} + (1-t)\boldsymbol{y}$. According to Lemma 4, $\texttt{OAC}(\gamma^{\texttt{BR}}(\boldsymbol{z}))$ is the optimal algorithm for the purchase path probability $\boldsymbol{z}$, resulting in $\alpha(\gamma^{\texttt{BR}}(\boldsymbol{z})) \leq t\alpha_1 + (1-t)\alpha_2$. Consequently, the competitive ratio $\alpha(\gamma^{\texttt{BR}}(\boldsymbol{q}))$ is a convex function with respect to $\boldsymbol{q}$.

## D  Optimal algorithm for CSR with upgrading

In this section, we present results related to CSR with upgrading. Specifically, we demonstrate that the optimal purchase strategy corresponds to $\texttt{OAC}(\sigma^*)$, where $\sigma^* = (S_1^*, S_2^*, \ldots, S_K^*)$ represents the purchase path of the optimal offline algorithm. Furthermore, we establish that the upper bound on the algorithm's competitive ratio is $e/(e-1)$.

## D.1 Problem setting

In CSR with upgrading, a purchase path is a sequence of super items $(S_1, S_2, \ldots, S_l)$ with non-decreasing purchase cost, i.e., $f(S_i) \leq f(S_j)$ for all $i, j \in \{1, 2, \ldots, l\}$ and $i \leq j$. Once the super item $S_{i+1}$ is purchased, the algorithm replaces the previously purchased super item $S_i$ with $S_{i+1}$. The expected cost of an online algorithm $A$ is

$$
\mathtt{ALG}(T; A) = \underbrace{\sum_{n \in [T]} \sum_{\sigma \in \Sigma} \sum_{i \in [l]} q(\sigma) \cdot (f(S_i) - f(S_{i-1})) \cdot p_n(S_i \mid \sigma)}_{\text{expected purchase cost}}
$$
$$
+ \underbrace{\sum_{n \in [T]} \sum_{\sigma \in \Sigma} \sum_{i \in [l]} q(\sigma) \cdot (1 - F_n(S_i \mid \sigma)) \cdot (g(S_i) - g(S_{i-1}))}_{\text{expected rental cost}}, \tag{15}
$$

where $F_n(S \mid \sigma) = \sum_{i=1}^{n} p_i(S \mid \sigma)$ is the cumulative distribution function of buying super item $S$ by day $n$ when purchase path $\sigma$ is chosen. The first term represents the expected purchase cost, where $q(\sigma) \cdot (f(S_i) - f(S_{i-1})) \cdot p_n(S_i \mid \sigma)$ denotes the expected cost incurred by purchasing $S_i$ on day $n$ when the purchase path $\sigma$ is chosen. The cost of purchasing $S_i$ is $f(S_i) - f(S_{i-1})$ due to upgrading. The second term represents the expected rental cost, where $1 - F_n(S_i \mid \sigma)$ denotes the probability that $S_i$ has not yet been purchased by day $n$. Additionally, under the purchase path $\sigma$, $p_n(S_{i+1} \mid \sigma) = 0$ when $F_n(S_i \mid \sigma) < 1$. The differences in $\mathtt{ALG}(T; A)$ between CSR with upgrading and CSR lie in the purchase path $\Sigma$ and the structure of the purchase cost.

In CSR with upgrading, the player can buy super items with incremental purchase cost over time and still benefit from combined purchase discounts. For example, if a player switches from a Microsoft 365 basic plan to a personal plan, they just need to equivalently pay the difference in cost between the two plans. However, in CSR, a player must pay the full price for each additional item purchased. If the player initially buys a subset of items and later decides to buy additional unpurchased items, they must pay the full cost without any discount regardless of previous purchase.

**Comparison between CSR and CSR with upgrading** We use a simple example to compare CSR and CSR with upgrading. Consider a scenario involving two items. The purchasing decision depends not only on the cost and potential discounts associated with bundles but also on the constraints imposed by the online setting. Figure 7(a) shows the purchase paths for buying items as one combination (Path 1), buying them separately (Path 2), and the optimal offline path (Path 3) as the skiing season progresses. In the CSR with upgrading setting, players can adopt an adaptive strategy by first purchasing one item and later upgrading to the bundle, allowing them to exploit discounts when advantageous. However, in CSR, once an item is purchased, upgrading to the bundle is not feasible without incurring the full bundle price. Consequently, players must commit to either Path 1 or Path 2 at the outset, making it challenging to optimize decisions without prior knowledge of future needs. The CSR problem can result in competitive ratios exceeding $e/(e-1)$ due to the irreversibility of the purchase paths, especially under moderate discount factors, as decision-making becomes more complex in this case. In contrast, CSR with upgrading consistently maintains competitive ratios below this threshold and aligns with CSR only in scenarios involving extreme discount rates [1, 19, 31].

## D.2 Overview the main results for CSR with upgrading

For the upgrading version, the expected purchase cost of acquiring all base items is fixed, regardless of the order of purchase. As a result, the only cost difference between strategies arises from the expected rental cost. At any given moment, the optimal online algorithm will choose to buy the super item that minimizes the current rental cost through upgrading. Since this super item is fixed, the selection process must follow a fixed purchase path rather than being randomized. Based on Lemma 2, the optimal purchase strategy must adhere to the OAC $(\sigma)$ strategy. Furthermore, we show that the optimal purchase path is the same as the purchase sequence by the offline algorithm $\sigma^* = (S_1^*, S_2^*, \ldots, S_K^*)$, which solves the Eq. (2) as the skiing season expands.

**Theorem 3.** *Given information setup $\mathcal{I}$ for the CSR with upgrading problem, OAC $(\sigma^*)$ is the optimal online algorithm and achieves the optimal competitive ratio $\alpha(\sigma^*)$.*

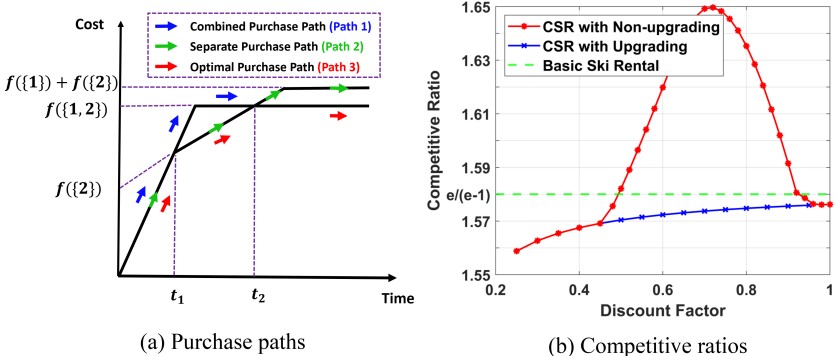

(a) Purchase paths                    (b) Competitive ratios

Figure 7: (a) The figure illustrates three offline purchase paths: *Path 1* (blue arrow) represents the combined purchase, *Path 2* (green arrow) corresponds to purchasing the items separately, and *Path 3* (red arrow) denotes the optimal offline path, where item 2 is purchased first, followed by the combination of items 1 and 2. In the CSR with upgrading scenario, all three paths are feasible. However, in the CSR scenario, the player is restricted to Path 1 or Path 2, as upgrading is not allowed. (b) The upper bound of the competitive ratio is computed for two items with prices in the range $[20, 80]$, where the combined purchase price is represented as the product of a discount factor and the sum of the individual prices.

The optimal offline algorithm determines which items to buy or rent at the beginning of the ski season by knowing the number of skiing days in advance. The elements in the set $\sigma^*$ represent the purchasing strategy that incurs the least cost as the skiing time increases. For CSR with upgrading, even though the number of skiing days is not known in advance, the algorithm can reverse previous decisions through upgrades. The optimal online algorithm will continuously upgrade its purchases to ensure that the current cost grows at the slowest possible rate. Therefore, the optimal purchase path should be $\sigma^*$. We formally prove this in Appendix D.3.

The optimal competitive ratio $\alpha(\sigma^*)$ of OAC $(\sigma^*)$ depends on the information setup in a complicated manner, and it is challenging to obtain a closed-form solution. However, the following lemma shows that $\alpha(\sigma^*)$ is upper bounded by $e/(e-1)$, which is the optimal competitive ratio of the classic ski rental problem.

**Lemma 6.** *For any information setup $\mathcal{I}$,* OAC $(\sigma^*)$ *for* CSR *with upgrading* *achieves a competitive ratio upper bounded by $e/(e-1)$.*

Compared to classic ski rental that buys or rents all items in a combination, CSR with upgrading achieves a better competitive ratio. This arises because, although both offline and online algorithms gain flexibility in fine-grained purchase options to reduce overall cost, the online algorithm experiences a more significant improvement.

### D.3   The optimal purchase path

In this section, we demonstrate that for CSR with upgrading, the optimal algorithm follows a fixed purchase path $\sigma^*$ (Theorem 3). We prove Proposition 1, Proposition 2, and Proposition 3 for CSR with upgrading based on the same principle, with necessary adjustments to the rental rate and purchase price for CSR with upgrading.

For CSR with upgrading, the cost of purchasing all base items is fixed at $f(\mathcal{M})$, regardless of the purchase order. The difference in cost among different strategies arises from the rental cost. Intuitively, when the expected purchase cost up to time $n$ is $C_n$, the optimal purchase strategy should select the super item that minimizes the rental cost while spending $C_n$ on purchases. Consequently, the optimal purchase strategy follows a fixed purchase path and adheres to the OAC $(\sigma)$ strategy, as demonstrated in Lemma 2. The next step is to determine the optimal purchase path. An intuitive idea is to follow the order $\sigma^* = (S_0^*, S_1^*, S_2^*, \ldots, S_K^*)$ in OPT, as OPT will always select the super item that minimizes the current cost. We prove this conclusion as stated in Theorem 3.

**Proof of Theorem 3.** We define the augmented set of super items is defined as $\gamma(q) := \{I(q(\sigma), S^\sigma)\}_{\sigma \in \Sigma, S^\sigma \in \sigma}$, where $I(q(\sigma), S^\sigma)$ represents a new super item containing the same base items as $S^\sigma$ but with a purchase cost of $q(\sigma) \cdot f(S^\sigma)$ and a rental rate of $q(\sigma) \cdot g(S^\sigma)$. For simplicity, we refer to $S^\sigma$ as $S$. Thus, for CSR with upgrading, all randomized strategies can be expressed as a fixed purchase path $(I(q(\sigma_0), S_0), I(q(\sigma_1), S_1), I(q(\sigma_2), S_2), \ldots, I(q(\sigma_l), S_l))$, where $S_0 = \varnothing$, and for all $i, j \in \{0, 1, 2, \ldots, l\}$ with $i \leq j$, it holds that $q(\sigma_i) \cdot f(S_i) \leq q(\sigma_j) \cdot f(S_j)$. Additionally, $q(\sigma_l) = 1$ and $S_l = \{\mathcal{M}\}$.

At this point, the problem transforms into an CSR problem with $l$ super items to be sequentially purchased, where the purchase cost and rental rate of the $i$-th super item are given by $q(\sigma_i) \cdot f(S_i) - q(\sigma_{i-1}) \cdot f(S_{i-1})$ and $q(\sigma_i) \cdot g(S_i) - q(\sigma_{i-1}) \cdot g(S_{i-1})$, respectively, for $i \in [l]$. The purchase order follows the sequence $i = 1, \ldots, l$. In other words, the problem reduces to minimizing the CSR problem with the purchase order $(I(q(\sigma_1), S_1)/I(q(\sigma_0), S_0), I(q(\sigma_2), S_2)/I(q(\sigma_1), S_1), \ldots, I(q(\sigma_l), S_l)/I(q(\sigma_{l-1}), S_{l-1}))$, where $q(\sigma_l) = 1$, $S_l = \{\mathcal{M}\}$, and $I(q(\sigma_i), S_i)/I(q(\sigma_{i-1}), S_{i-1})$ denotes a super item with a purchase cost of $q(\sigma_i) \cdot f(S_i) - q(\sigma_{i-1}) \cdot f(S_{i-1})$ and a rental cost of $q(\sigma_i) \cdot g(S_i) - q(\sigma_{i-1}) \cdot g(S_{i-1})$.

By Lemma 3, the optimal purchase path can be recursively characterized as

$$\hat{S}_i = \arg\min_{S_i} \left\{ \frac{f(S_i) - f(\hat{S}_{i-1})}{g(S_i) - g(\hat{S}_{i-1})} \,\middle|\, f(S_i) > f(\hat{S}_{i-1}) \right\}, \quad i = 1, 2, \ldots \tag{16}$$

This result can be rigorously established via mathematical induction.

When $i = 1$,

$$\hat{S}_1 = \arg\min_{S_1} \left\{ \frac{q(\sigma_1) \cdot f(S_1) - q(\sigma_0) \cdot f(\hat{S}_0)}{q(\sigma_1) \cdot g(S_1) - q(\sigma_0) \cdot g(\hat{S}_0)} \,\middle|\, q(\sigma_1) \cdot f(S_1) > q(\sigma_0) \cdot f(\hat{S}_0) \right\}$$

$$= \arg\min_{S_1} \left\{ \frac{f(S_1) - f(\hat{S}_0)}{g(S_1) - g(\hat{S}_0)} \,\middle|\, f(S_1) > f(\hat{S}_0) \right\},$$

where we set $q(\sigma_1) = 1$ because $\hat{S}_1$ is the optimal purchase set.

Assume the characterization holds for $i - 1 \geq 1$. For $i$, we have

$$\hat{S}_i = \arg\min_{S_i} \left\{ \frac{q(\sigma_i) \cdot f(S_i) - q(\sigma_{i-1}) \cdot f(\hat{S}_{i-1})}{q(\sigma_i) \cdot g(S_i) - q(\sigma_{i-1}) \cdot g(\hat{S}_{i-1})} \,\middle|\, q(\sigma_i) \cdot f(S_i) > q(\sigma_{i-1}) \cdot f(\hat{S}_{i-1}) \right\}$$

$$= \arg\min_{S_i} \left\{ \frac{q(\sigma_i) \cdot f(S_i) - f(\hat{S}_{i-1})}{q(\sigma_i) \cdot g(S_i) - g(\hat{S}_{i-1})} \,\middle|\, q(\sigma_i) \cdot f(S_i) > f(\hat{S}_{i-1}) \right\}$$

$$= \arg\min_{S_i} \left\{ \frac{f(S_i) - f(\hat{S}_{i-1})}{g(S_i) - g(\hat{S}_{i-1})} \,\middle|\, f(S_i) > f(\hat{S}_{i-1}) \right\},$$

The last equation holds because if both $q(\sigma_i) \cdot f(S_i^{(1)}) > f(\hat{S}_{i-1})$ and $q(\sigma_i) \cdot f(S_i^{(2)}) > f(\hat{S}_{i-1})$ hold, and if

$$\frac{f(S_i^{(1)})}{g(S_i^{(1)})} > \frac{f(S_i^{(2)})}{g(S_i^{(2)})},$$

then $S_i^{(2)}$ is strictly better than $S_i^{(1)}$. In other words, the selection of $\hat{S}_i$ is independent of the value of $q(\sigma_i)$. Therefore, we set $q(\sigma_i) = 1$, as $\hat{S}_i$ represents the optimal choice at this stage. This completes the proof of the claim.

Hence, by induction, the recurrence relation in Eq. (16) holds for all $i \geq 1$. This relation recursively determines the sequence $(S_0^*, S_1^*, S_2^*, \ldots, S_K^*)$.

## D.4 An $e/(e-1)$ upper bound for CSR with upgrading

To prove the upper bound of $e/(e-1)$ on the competitive ratio for CSR with upgrading (Lemma 6), we show there exists an algorithm for CSR with upgrading that is $e/(e-1)$ competitive. OAC$(\sigma^*)$ is the optimal online algorithm for CSR with upgrading and thus $\alpha(\sigma) \leq e/(e-1)$.

Given a setup $\mathcal{I} = (\mathcal{M}, f, g)$ of CSR with upgrading, we define $K$ classic ski rental problems using the purchase sequence $\sigma^* = (S_0^*, S_1^*, S_2^*, \ldots, S_K^*)$ derived from the optimal offline cost. Each information setup is defined as $\mathcal{I}_i = (\mathcal{M}_i, f_i, g_i)$ for $i \in \{1, 2, \ldots, K\}$, where $|\mathcal{M}_i| = 1$. The purchase cost is $f_i(\mathcal{M}_i) = f(S_i^*) - f(S_{i-1}^*)$, and the rental cost is $g_i(\mathcal{M}_i) = g(S_i^*) - g(S_{i-1}^*)$. The optimal offline cost for setup $\mathcal{I}_i$ is given by

$$\text{OPT}(T; \mathcal{I}_i) = \min \left\{ \left( g(S_i^*) - g(S_{i-1}^*) \right) \cdot T, f(S_i^*) - f(S_{i-1}^*) \right\}. \tag{17}$$

When the algorithm buys the $i$-th super item in the OPT purchase sequence, it is referred to as state $i$. Unlike CSR, which has $K + 1$ states, the classic ski rental problem has only two states: state 0, when the item is rented for the entire duration, and state 1, when the item is purchased at the beginning of the ski season. We define $p_n^{(i)}(A, \mathcal{I})$ as the probability of being in state $i$ at time $n$ for algorithm $A$.

Let $A_i^*$ denote the optimal randomized algorithm for information setup $\mathcal{I}_i$, $i \in \{1, 2, \ldots, K\}$. The competitive ratio of this strategy is known to be $e/(e-1)$ for each instance $i$. For CSR with upgrading, define algorithm $A^*$ to satisfy the following conditions: among multiple combinations of purchases, only the sets $S_0^*, S_1^*, S_2^*, \ldots, S_K^*$ are likely to be purchased, with the probability of purchasing any other combination being zero. The probability of being in each state is given by

$$p_n^{(0)}(A^*, \mathcal{I}) = p_n^{(0)}(A_1^*, \mathcal{I}_1),$$
$$p_n^{(i)}(A^*, \mathcal{I}) = p_n^{(1)}(A_i^*, \mathcal{I}_i) - p_n^{(1)}(A_{i+1}^*, \mathcal{I}_{i+1}), i = 1, 2, \ldots, K - 1,$$
$$p_n^{(K)}(A^*, \mathcal{I}) = p_n^{(1)}(A_K^*, \mathcal{I}_K).$$

Since upgrading is allowed, algorithm $A^*$ at time $n$ must be in one of these $K + 1$ states. The cost of algorithm $A^*$ at time $n$ is

$$\text{ALG}(n; A^*, \mathcal{I}) = \sum_{i=0}^{K} p_n^{(i)}(A^*, \mathcal{I}) \cdot f(S_i^*) + \sum_{j=1}^{n} \sum_{i=0}^{K} p_j^{(i)}(A^*, \mathcal{I}) \cdot g(\mathcal{M} \setminus S_i^*), \tag{18}$$

where the first term represents the expected purchase cost until day $n$, and the second term represents the expected rental cost until day $n$.

We then show that the algorithm $A^*$ defined above achieves a competitive ratio of $e/(e-1)$ (Lemma 6). To establish this result, we prove Proposition 6, 7, and 8. Proposition 6 demonstrates the soundness of algorithm $A^*$; Proposition 7 states that the sum of the costs of $K$ online algorithms $A_i^*$ is equal to the cost of the online algorithm $A^*$; Proposition 8 clarifies that the sum of the optimal offline costs for these $K$ instances equals the optimal offline cost for the CSR with upgrading problem.

**Proposition 6.** *For the randomized algorithm $A^*$ defined above, the following conditions hold*

*(1) $p_n^{(i)}(A^*, \mathcal{I}) \geq 0$, $i \in \{0, 1, 2, \ldots, K\}$.*

*(2) $\sum_{i=0}^{K} p_n^{(i)}(A^*, \mathcal{I}) = 1$.*

*Proof.* To support the claim (1), we need to show that $p_n^{(1)}(A_i^*, \mathcal{I}_i) \geq p_n^{(1)}(A_{i+1}^*, \mathcal{I}_{i+1})$ for $i = 1, 2, \ldots, K - 1$. Given that $A_i^*$ is the optimal randomized strategy for the classic ski rental problem, Proposition 9 proves $p_n^{(1)}(A_i^*, \mathcal{I}_i) = \frac{(1-1/s_i)^{-n} - 1}{(1-1/s_i)^{-s_i} - 1}$ and indicates that $p_n^{(1)}(A_i^*, \mathcal{I}_i)$ is monotonically decreasing with respect to $t_i$. Since $t_i \leq t_{i+1}$ for $i = 1, 2, \ldots, K - 1$, it follows that $p_n^{(1)}(A_i^*, \mathcal{I}_i) \geq p_n^{(1)}(A_{i+1}^*, \mathcal{I}_{i+1})$. For claim (2), we derive the following

$$\sum_{i=0}^{K} p_n^{(i)}(A^*, \mathcal{I}) = p_n^{(0)}(A_1^*, \mathcal{I}_1) + \sum_{i=1}^{K-1} (p_n^{(1)}(A_i^*, \mathcal{I}_i) - p_n^{(1)}(A_{i+1}^*, \mathcal{I}_{i+1})) + p_n^{(1)}(A_K^*, \mathcal{I}_K)$$
$$= p_n^{(0)}(A_1^*, \mathcal{I}_1) + p_n^{(1)}(A_1^*, \mathcal{I}_1) = 1.$$

$\square$

**Proposition 7.** $\text{ALG}(n; A^*, \mathcal{I}) = \sum_{i=1}^{K} \text{ALG}(n; A_i^*, \mathcal{I}_i), \forall n \in \mathbb{N}^+$.

*Proof.* The cost of algorithm $A_i^*$ at time $n$ is

$$\text{ALG}(n; A_i^*, \mathcal{I}_i) = (f(S_i^*) - f(S_{i-1}^*) \cdot p_n^{(1)}(A_i^*, \mathcal{I}_i) + \sum_{j=1}^{n}(g(S_i^*) - g(S_{i-1}^*)) \cdot p_j^{(0)}(A_i^*, \mathcal{I}_i).$$

Then, we have

$$\sum_{i=1}^{K} \text{ALG}(n; A_i^*, \mathcal{I}_i) = \underbrace{\sum_{i=1}^{K}(f(S_i^*) - f(S_{i-1}^*)) \cdot p_n^{(1)}(A_i^*, \mathcal{I}_i)}_{\text{Term (1)}} + \underbrace{\sum_{j=1}^{n}\sum_{i=1}^{K}(g(S_i^*) - g(S_{i-1}^*)) \cdot p_j^{(0)}(A_i^*, \mathcal{I}_i)}_{\text{Term (2)}}.$$

For the first term of Eq. (18), we have

$$\sum_{i=0}^{K} p_n^{(i)}(A^*, \mathcal{I}) \cdot f(S_i^*)$$

$$=p_n^{(0)}(A_1^*, \mathcal{I}_1) \cdot f(S_0^*) + \sum_{i=1}^{K-1}\left(p_n^{1}(A_i^*, \mathcal{I}_i) - p_n^{(1)}(A_{i+1}^*, \mathcal{I}_{i+1})\right) \cdot f(S_i^*) + p_n^{(1)}(A_K^*, \mathcal{I}_K) \cdot f(S_K^*)$$

$$=p_n^{(1)}(A_1^*, \mathcal{I}_1) \cdot f(S_1^*) + \sum_{i=2}^{K} p_n^{(1)}(A_i^*, \mathcal{I}_i) \cdot \left(f(S_i^*) - f(S_{i-1}^*)\right) = \sum_{i=1}^{K} p_n^{(1)}(A_i^*, \mathcal{I}_i) \cdot \left(f(S_i^*) - f(S_{i-1}^*)\right)$$

$$=\text{Term (1)}.$$

For the second term of Eq. (18), by utilizing the relationship $p_n^{(1)}(A_i^*, \mathcal{I}_i) = 1 - p_n^{(0)}(A_i^*, \mathcal{I}_i)$ for $i \in \{1, 2, \ldots, K\}$, we obtain

$$\sum_{j=1}^{n}\sum_{i=0}^{K} p_j^{(i)}(A^*, \mathcal{I}) \cdot g(\mathcal{M} \setminus S_i^*)$$

$$=\sum_{j=1}^{n}\left(p_j^{(0)}(A_1^*, \mathcal{I}_1) \cdot g(\mathcal{M} \setminus S_0^*) + \sum_{i=1}^{K-1}(p_j^{(1)}(A_i^*, \mathcal{I}_i) - p_j^{(1)}(A_{i+1}^*, \mathcal{I}_{i+1})) \cdot g(\mathcal{M} \setminus S_i^*) + p_j^{(1)}(A_K^*, \mathcal{I}_K) \cdot g(\mathcal{M} \setminus S_K^*)\right)$$

$$=\sum_{j=1}^{n}\left(p_j^{(0)}(A_1^*, \mathcal{I}_1) \cdot g(S_K^*) + \sum_{i=1}^{K-1}(p_j^{(0)}(A_{i+1}^*, \mathcal{I}_{i+1}) - p_j^{(0)}(A_i^*, \mathcal{I}_i)) \cdot g(\mathcal{M} \setminus S_i^*)\right)$$

$$=\sum_{j=1}^{n}\sum_{i=1}^{K} p_j^{(0)}(A_i^*, \mathcal{I}_i) \cdot (g(S_i^*) - g(S_{i-1}^*)) = \text{Term (2)}.$$

Therefore, it can be concluded that the cost of algorithm $A^*$ is equal to the sum of the costs of the $K$ algorithms for $i \in \{1, 2, \ldots, K\}$. $\qquad\square$

**Proposition 8.** $\text{OPT}(n; \mathcal{I}) = \sum_{i=1}^{K} \text{OPT}(n; \mathcal{I}_i), \forall n \in \mathbb{N}^+.$

*Proof.* Let $i(n)$ be the state that optimal is in at time $n$, where $n \in \mathbb{N}^+$. Then, when $i \leq i(n)$, $\text{OPT}(n; \mathcal{I}_i) = f(S_i) - f(S_{i-1})$; when $i > i(n)$, $\text{OPT}(n; \mathcal{I}_i) = (g(S_i) - g(S_{i-1})) \cdot n$. We can get that

$$\sum_{i=1}^{K} \text{OPT}(n; \mathcal{I}_i) = \sum_{i=1}^{i(n)}(f(S_i) - f(S_{i-1})) + \sum_{i=i(n)+1}^{K}(g(S_i) - g(S_{i-1})) \cdot n$$

$$= f(S_{i(n)}) - f(S_0) + (g(S_K) - g(S_{i(n)})) \cdot n$$

$$= f(S_{i(n)}) + g(\mathcal{M} \setminus S_{i(n)}) \cdot n$$

$$= \text{OPT}(n; \mathcal{I}).$$

$\qquad\square$

**Proof of Lemma 6.** Since each algorithm $A_i^*$ has an upper bound on its competitive ratio of $e/(e-1)$, we can derive the following relationship

$$\mathtt{ALG}(n; A^*, \mathcal{I}) = \sum_{i=1}^{K} \mathtt{ALG}(n; A_i^*, \mathcal{I}_i) \leq \frac{e}{e-1} \sum_{i=1}^{K} \mathtt{OPT}(n; \mathcal{I}_i) = \frac{e}{e-1}\mathtt{OPT}(n; \mathcal{I}),$$

which means that algorithm $A^*$ has $e/(e-1)$ competitive ratio.

### D.5 Proof of Proposition 9

To prove Proposition 6, we establish Proposition 9, which outlines some properties of the optimal randomized strategy for the classic ski rental problem.

**Proposition 9.** *For a classic ski rental problem $\mathcal{I} = (\mathcal{M}, f, g)$, where $\mathcal{M} = \{1\}$. The rental rate is given by $g(\{1\}) = r$ and the purchase price is $f(\{1\}) = b$, with $b/r \in \mathbb{N}^+$ and $b/r > 1$. The cumulative distribution function of the optimal randomized algorithm $A^*$ on day $i$ ($i \leq b/r$) is given by*

$$F_i(\{1\}; A^*) = \frac{(1 - r/b)^{-i} - 1}{(1 - r/b)^{-b/r} - 1}.$$

*Moreover, $F_i(\{1\}; A^*)$ for $i \leq b/r$ is monotonically decreasing with respect to $b/r$.*

*Proof.* The probability that the optimal randomized strategy [12] for the classic ski rental problem buys on day $i$ is given by

$$p_i(\{1\}; A^*) = \begin{cases} \left(\frac{b-r}{b}\right)^{b/r-i} \frac{1}{b/r\left(1-(1-(r/b))^{b/r}\right)}, & i \leq b/r \\ 0, & n > b/r \end{cases}$$

When $i \leq b/r$, this represents a geometric series with a common ratio of $b/(b-r)$. Therefore, we have

$$F_i(\{1\}; A^*) = \frac{p_1(\{1\}; A^*) \cdot (1 - (b/(b-r))^i)}{1 - (b/(b-r))} = \frac{(1 - r/b)^{-i} - 1}{(1 - r/b)^{-b/r} - 1}.$$

To demonstrate that $F_i(\{1\}; A^*)$ for $i \leq b/r$ is monotonically decreasing with respect to $b/r$, define the function $F(n)$ as follows

$$F(n) = \frac{(1 - 1/n)^{-i} - 1}{(1 - 1/n)^{-n} - 1} = \frac{(n/(n-1))^i - 1}{(n/(n-1))^n - 1}, \quad 1 \leq i \leq n, i \in \mathbb{N}^+.$$

We just need to demonstrate that the function $F(n)$ is monotonically decreasing with respect to $n$ ($n \geq 2$), establishing the desired property of $F_i(\{1\}; A^*)$. Define two auxiliary functions: $f_1(n) = (n/(n-1))^i$, $f_2(n) = (n/(n-1))^n$. Then, we can express $F(n)$ as

$$F(n) = \frac{f_1(n) - 1}{f_2(n) - 1}.$$

Then, we compute the derivatives of $f_1(n)$ and $f_2(n)$

$$\frac{df_1(n)}{dn} = -\frac{i f_1(n)}{n(n-1)}, \quad \frac{df_2(n)}{dn} = f_2(n)\left(\ln\frac{n}{n-1} - \frac{1}{n-1}\right).$$

Using these derivatives, we can express the derivative of $F(n)$ as follows

$$\frac{dF(n)}{dn} = -\frac{i f_1(n)}{n(n-1)}(f_2(n) - 1) - f_2(n)(f_1(n) - 1)\left(\ln(1 + \frac{1}{n-1}) - \frac{1}{n-1}\right).$$

By applying the inequality $\ln(1 + \frac{1}{n-1}) \geq \frac{1}{n-1} - \frac{1}{2(n-1)^2}$, we obtain

$$\frac{dF(n)}{dn} = \frac{(1/2)n f_2(n)(f_1(n) - 1) - i(n-1)f_1(n)(f_2(n) - 1)}{n(n-1)^2}.$$

Then, we derive the expression for $n(n-1)^2 F'(n)$,

$$n(n-1)^2 F'(n) = \frac{1}{2} n f_2(n)(f_1(n)-1) - i(n-1)f_1(n)(f_2(n)-1)$$

$$= \left(\frac{n}{2} - i(n-1)\right)\left(\frac{n}{n-1}\right)^{n+i} + i(n-1)\left(\frac{n}{n-1}\right)^i - \frac{n}{2}\left(\frac{n}{n-1}\right)^n.$$

We simplify further

$$\frac{n(n-1)^2 F'(n)}{(n/(n-1))^i} = \left(\frac{n}{2} - i(n-1)\right)\left(\frac{n}{n-1}\right)^i + i(n-1) - \frac{n}{2}\left(\frac{n}{n-1}\right)^{n-i}$$

$$\leq \left(\frac{n}{2} - i(n-1)\right)\left(\frac{n}{n-1}\right)^i + i(n-1)\left(\frac{n}{n-1}\right)^{n-i} - \frac{n}{2}\left(\frac{n}{n-1}\right)^{n-i}$$

$$= \left(\frac{n}{2} - i(n-1)\right)\left(\frac{n}{n-1}\right)^{n-i}\left(\left(\frac{n}{n-1}\right)^i - 1\right).$$

Since $\frac{n}{2} - i(n-1) \leq 0$ for $1 \leq i \leq n$, $i \in \mathbb{N}^+$ and $n \geq 2$, it follows that

$$\frac{n(n-1)^2 F'(n)}{(n/(n-1))^i} \leq 0,$$

which implies that $F'(n) \leq 0$. Therefore, we can conclude that $F(n)$ is monotonically decreasing with respect to $n$ for $n \geq 2$.

$\square$

# E    Proof for Lemma 1

The proof of Lemma 1 can be decomposed into three lemmas, each demonstrating how the CSR problem is reduced to one of the following problems: *multi-shop ski rental* [1], *multi-slope ski rental* [19], and *multi-commodity ski rental* [29]. For details on CSR with upgrading, see Appendix D, Theorem 3.

**Proposition 10.** *The CSR problem can be reduced to the multi-shop ski rental problem by restricting the purchase path set $\Sigma$ to single-element paths corresponding to individual shops. Consequently, the SOAC algorithm achieves the optimal competitive ratio for the multi-shop ski rental problem.*

**Proposition 11.** *The CSR problem can be reduced to the multi-slope ski rental problem by restricting the purchase path $\sigma$ to a single path. Consequently, the SOAC algorithm achieves the optimal competitive ratio for the multi-slope ski rental problem.*

**Proposition 12.** *The CSR problem can be reduced to the multi-commodity ski rental problem by assuming that no combination purchases provide a discount. Consequently, the SOAC algorithm achieves the optimal competitive ratio for the multi-commodity ski rental problem.*

The online strategy of the CSR is characterized by two fundamental decision dimensions:

- *Purchase Path* $(\sigma_1, \sigma_2, \ldots, \sigma_{|\Sigma|})$: This represents the process of purchasing all required items along multiple distinct paths.

- *Purchase Time* $(\boldsymbol{t}(\sigma) := \{t(S \mid \sigma)\}_{S \in \sigma})$: This defines the specific time $t(S \mid \sigma)$ at which a super item $S$ is purchased along the path $\sigma$.

Compared to the classical ski rental problem, the CSR introduces two additional layers of complexity:

- **Path Dimension**: The decision-making framework expands from a single purchase path to multiple paths.

- **Time Dimension**: The decision process extends from single-stage to multi-stage decision-making, which requires determining when different super items should be purchased.

### E.1 Proof of Proposition 10

*Proof.* In the multi-shop ski rental problem, a skier must decide how to ski for an unknown duration $T$. The skier chooses a shop $i$ from a set of shops $\{1, 2, \ldots, m\}$ and can only rent or purchase ski equipment from that selected shop. Each shop $i$ offers a daily rental price $r_i$ and a purchase price $b_i$, where $r_i, b_i > 0$. It is assumed that $r_1 < r_2 < \cdots < r_m$ and $b_1 > b_2 > \cdots > b_m$, ensuring that shops with lower rental prices have higher purchase prices to avoid trivial decisions. The skier's goal is to select a shop and determine the optimal purchase time to minimize total costs, without changing shops midway.

In the CSR problem, consider a scenario where there is only one item and this item is combined with itself, resulting in different rental and purchase costs. Under these conditions:

- The purchase path set $\Sigma$ is restricted to single-element paths, where each path corresponds to selecting one "super item" $S_i = \{i\}$.

- Each super item $S_i$ has a rental cost $r_i$ and a purchase cost $b_i$, directly corresponding to the rental and purchase prices of shop $i$.

Thus, the purchase path set becomes $\Sigma = \{\{1\}, \{2\}, \ldots, \{m\}\}$, similar to choosing a shop in the multi-shop ski rental problem. This demonstrates that the multi-shop ski rental problem is a special case of the CSR problem. $\square$

**Remark 3** (Analysis of Path Irreversibility). *In the multi-shop ski rental problem, changing shops midway during the rental period is prohibited. If such changes were allowed, the problem would reduce to the classical ski rental problem, where the user would simply rent from the shop with the lowest rental price and purchase from the shop with the lowest purchase price. Similarly, the primary distinction between the* CSR *and* CSR with upgrading *problems is that the CSR problem disallows changing purchase paths midway. Allowing such changes would transform the* CSR *problem into the multi-slope ski rental problem, as shown in Theorem 3.*

### E.2 Proof of Proposition 11

*Proof.* In the multi-slope ski rental problem, the user needs to utilize a resource for an unknown duration. The cost of resource usage is determined by multiple *states* (or *slopes*) $\mathcal{S} = \{0, 1, \ldots, k\}$. Each state $i$ has a purchase cost $b_i$ and a rental rate $r_i$, typically satisfying $b_0 < b_1 < \cdots < b_k$ and $r_0 > r_1 > \cdots > r_k$, where $b_0 = 0$. For simplicity, we assume that $r_k = 0$. If $r_k \neq 0$, an additional state can be introduced, defined as $b_{k+1} = \infty$ and $r_{k+1} = 0$, to accommodate this case. The user's objective is to determine a set of transition times $(t_1, t_2, \ldots, t_k)$, where $t_i$ denotes the time to transition from state $i - 1$ to state $i$. Transitions are only allowed in the forward direction $i \to i + 1$, and skipping states is not permitted.

To reduce this problem to the CSR problem, we restrict the purchase path in the CSR problem to a single path $\sigma = (S_0, S_1, S_2, \ldots, S_k)$. This restriction is feasible under the assumption that each $S_i$ corresponds to a base item containing only one element. In this case, there is no discount for any combination of elements, so the only valid path to purchase is $\sigma$. The cost parameters are defined as follows

- The purchase cost of super item $S_i$ is $f(S_i) = b_i - b_{i-1}$ for $i \in \{1, \ldots, k\}$ and $f(S_0) = 0$;

- The rental cost of super item $S_i$ is $g(S_i) = r_{i-1} - r_i$ for $i \in \{1, \ldots, k\}$ and $g(S_0) = 0$.

In the CSR problem, state $i$ corresponds to the state where super items $S_0, S_1, \ldots, S_i$ have been purchased. In this state, the purchase cost is $\sum_{j=0}^{i} f(S_j)$, and the rental cost is $\sum_{j=i+1}^{k} g(S_j)$. By the definition of the purchase and rental costs for the super items, we have

$$\sum_{j=0}^{i} f(S_j) = b_i, \quad \sum_{j=i+1}^{k} g(S_j) = r_i.$$

Thus, under this special case, the decision path dimension of the CSR problem reduces to a single path, making it equivalent to the multi-slope ski rental problem. $\square$

**Algorithm 4** $\overline{\mathtt{OAC}}$: Extended Amortized Cost

---

1: **Input**: Information setup $\mathcal{I} = (\mathcal{M}, f, g)$; purchase path $\sigma$; stopping criterion $\varepsilon$; critical time $T_{\mathtt{ML}}$
2: **Initialization**: $\alpha = 1$, $\alpha_{\max} = f(\mathcal{M})$, $\alpha_{\min} = 1$
3: Compute $\mathtt{OPT}(T)$ and $T_{\mathtt{OPT}}$ using Eq. (2)
4: Compute $\overline{\mathtt{OPT}}(T)$ using Eq. (19)
5: Construct algorithm $A = \overline{\mathtt{AC}}(\sigma; \alpha)$
6: **while** $\alpha_{\max} - \alpha_{\min} > \varepsilon$ **do**
7:     Calculate $T_A$ using $\alpha$ according to Eq. (1) and Eq. (20)
8:     **if** $T_A < T_{\mathtt{ML}}$ **then**
9:         $\alpha_{\max} \leftarrow \alpha$
10:     **else**
11:         $\alpha_{\min} \leftarrow \alpha$
12:     **end if**
13:     Update $\alpha \leftarrow (\alpha_{\max} + \alpha_{\min})/2$
14: **end while**
15: **return** $\alpha_{\max}$, $\overline{\mathtt{AC}}(\sigma; \alpha_{\max})$

---

### E.3 Proof of Proposition 12

*Proof.* The multi-commodity ski rental problem considers a set of items that must be utilized simultaneously, where each item has its own independent rental and purchase prices. By imposing the constraint in the $\mathtt{CSR}$ problem that no combinations of items are eligible for discounts, the $\mathtt{CSR}$ problem can be simplified and directly reduced to the multi-commodity ski rental problem. $\qquad\square$

**Remark 4.** *Existing research on the multi-commodity ski rental problem has achieved optimal solutions only in scenarios where all items share the same buy-to-rent ratio. However, determining the optimal randomized algorithm for arbitrary purchase and rental price configurations remains an open problem. The algorithm proposed in this paper provides the first optimal randomized solution for the general multi-commodity ski rental problem, thereby addressing this gap in the literature.*

## F Supplementary Material for $\mathtt{LA\text{-}SOAC}$

### F.1 Supplementary details of $\mathtt{LA\text{-}SOAC}$

This section details the implementation of the $\mathtt{LA\text{-}SOAC}$ algorithm, presented in Section 4. We begin by outlining the core design principles of the algorithm, and then formally describe the mechanics of the algorithm.

The $\mathtt{LA\text{-}SOAC}$ algorithm aims to optimize purchasing strategies based on machine-learned predictions. It takes a hyperparameter $\lambda \in (0, 1)$ as input and generates two different purchasing strategies based on the predicted time $y$ relative to the optimal completion time $T_{\mathtt{OPT}}$. The strategies are defined as follows

- If $y \geq T_{\mathtt{OPT}}$, the algorithm prioritizes early purchases, setting the completion time to $T_{\mathtt{ML}} = \lfloor \lambda T_{\mathtt{OPT}}^{(1)} \rfloor$. This is different from a baseline approach without machine learning, which completes the purchase at $T_{\mathtt{OPT}}$.

- If $y < T_{\mathtt{OPT}}$, the algorithm delays the purchase, setting the completion time to $T_{\mathtt{ML}} = \lceil T_{\mathtt{OPT}}^{(2)}/\lambda \rceil$.

An issue arises because the incremental cost $\Delta\mathtt{OPT}(n) = 0$ for $n > T_{\mathtt{OPT}}$, rendering it infeasible to track the optimal cost function $\mathtt{OPT}(n)$ beyond this point. To address this, we introduce an augmented cost function, $\overline{\mathtt{OPT}}(n)$, define its increment $\Delta\overline{\mathtt{OPT}}(n)$ as

$$\Delta\overline{\mathtt{OPT}}(n) = \begin{cases} \Delta\mathtt{OPT}(n), & n \leq t^*, \\ \frac{\mathtt{OPT}(T_{\mathtt{OPT}})}{T_{\mathtt{OPT}}}, & n > t^*. \end{cases} \tag{19}$$

As established in Lemma 2, $\overline{\mathtt{OPT}}(n)$ must be concave to ensure the validity of prior theoretical results. The definition of $\overline{\mathtt{OPT}}(n)$ satisfies this requirement, enabling feasible cost tracking. We propose an

**Algorithm 5** $\overline{\text{SOAC}}$: Sorted Optimal Amortized Cost Algorithm

---

1: **Input**: Information setup $\mathcal{I} = (\mathcal{M}, f, g)$; stopping criterion $\varepsilon$; Learning rate: $\eta$; critical time $T_{\text{ML}}$
2: Calculate all disjoint divisions: $\Gamma = \{\gamma_1, \gamma_2, \ldots, \gamma_{|\Gamma|}\}$
3: Sort divisions $\Gamma$ by BR: $\{\sigma_1, \sigma_2, \ldots, \sigma_{|\Gamma|}\}$
4: Construct $\gamma^{\text{BR}}(\boldsymbol{q})$ based on Eq. (4)
5: **Initialization**:
6: $\boldsymbol{q} = (q(\sigma_i))_{i=1}^{|\Gamma|} \leftarrow (\frac{1}{|\Gamma|}, \ldots, \frac{1}{|\Gamma|}), \alpha \leftarrow f(\mathcal{M})$
7: **for** $t = 1, \ldots$ **do**
8:     **for** $i = 1, \ldots, |\Gamma|$ **do**
9:         */\* Compute Gradient by the Finite Difference Method*
10:         $\text{grad}_i \leftarrow \frac{\partial}{\partial q(\sigma_i)} \text{CR}(\overline{\text{OAC}}(\gamma^{\text{BR}}(\boldsymbol{q})))$
11:     **end for**
12:     */\* Gradient Update*
13:     $\boldsymbol{q} \leftarrow \boldsymbol{q} - \eta \cdot \text{grad}$
14:     */\* Projection to Simplex*
15:     $\boldsymbol{q} \leftarrow \text{ProjectSimplex}(\boldsymbol{q})$
16:     $\alpha_{\text{new}} = \text{CR}(\overline{\text{OAC}}(\gamma^{\text{BR}}(\boldsymbol{q})))$
17:     **if** $|\alpha - \alpha_{\text{new}}| < \varepsilon$ **then**
18:         **Break**
19:     **end if**
20:     $\alpha \leftarrow \alpha_{\text{new}}$
21: **end for**

---

algorithm, denoted as $\overline{\text{OAC}}$, which modifies the AC strategy by substituting $\text{OPT}(n)$ with $\overline{\text{OPT}}(n)$ in Algorithm 1, while adhering to the specified conditions. Denote a strategy satisfying Eq. (20) as the $\overline{\text{AC}}$ strategy.

$$
\begin{aligned}
\Delta\text{ALG}(n; A_{y;\lambda}) &= \alpha \cdot \Delta\overline{\text{OPT}}(n), && \text{if } n < T_{A_{y;\lambda}}, \\
\Delta\text{ALG}(n; A_{y;\lambda}) &\leq \alpha \cdot \Delta\overline{\text{OPT}}(n), && \text{if } n = T_{A_{y;\lambda}}, \\
\Delta\text{ALG}(n; A_{y;\lambda}) &= 0, && \text{if } n > T_{A_{y;\lambda}}.
\end{aligned}
\tag{20}
$$

The implementation of the $\overline{\text{OAC}}$ algorithm is detailed in Algorithm 4. The algorithm employs a binary search to iteratively refine the parameter $\alpha$ within the interval $[\alpha_{\min}, \alpha_{\max}]$. Initially, $\alpha$ is set, and the $\overline{\text{AC}}$ strategy is constructed based on the input purchase path $\sigma$. The algorithm then calculates the optimal cost $\text{OPT}(T)$ and the augmented cost $\overline{\text{OPT}}(T)$ using Eq. (2) and (19), respectively. Convergence is achieved when the difference between $\alpha_{\max}$ and $\alpha_{\min}$ falls below the stopping criterion $\varepsilon$. Then, by replacing the subroutine SOAC in algorithm OAC with $\overline{\text{OAC}}$, we obtain the modified algorithm $\overline{\text{SOAC}}$.

### F.2   Proof of Theorem 2

*Proof.* We analyze the performance of the ML-based algorithm case by case.

**Case 1:** $y \geq T_{\text{OPT}}$    In this case, the threshold determined by the ML algorithm is given by $T_{\text{ML}} = \lfloor \lambda T_{\text{OPT}}^{(1)} \rfloor$. According to Algorithm 3, the algorithm ensures a uniform ratio across decision points up to $T_{\text{ML}}$:

$$
\frac{\Delta\text{ALG}(n; A_{y;\lambda}^{(1)})}{\Delta\overline{\text{OPT}}(n)} = c, \quad \forall n \leq \lfloor \lambda T_{\text{OPT}}^{(1)} \rfloor,
$$

for some constant $c$. Moreover, $\Delta\text{ALG}(n; A_{y;\lambda}^{(1)}) = 0$ for $n > \lfloor \lambda T_{\text{OPT}}^{(1)} \rfloor$.

**Consistency:** When $T = y \geq T_{\text{OPT}}$, the cost of the offline optimal algorithm beyond $T_{\text{OPT}}$ remains constant, and the online algorithm incurs no additional cost beyond $T_{\text{ML}}$. The worst-case competitive ratio, therefore, occurs at $x = T_{\text{OPT}}$, yielding $\frac{\text{ALG}(T_{\text{OPT}}; A_{y;\lambda}^{(1)})}{\text{OPT}(T_{\text{OPT}})}$, which upper bounds the consistency ratio.

**Robustness:** We consider arbitrary $T > 0$ and analyze the robustness under two subcases:

*Subcase 1.1:* $\lfloor \lambda T_{\text{OPT}}^{(1)} \rfloor \leq t^*$.

In this regime, $\Delta\overline{\text{OPT}}(n) = \Delta\text{OPT}(n)$ for all $n \leq \lfloor \lambda T_{\text{OPT}}^{(1)} \rfloor$, hence the cost ratio remains constant. The worst case again occurs at $T = \lfloor \lambda T_{\text{OPT}}^{(1)} \rfloor$, giving a robustness ratio of $\frac{\text{ALG}(\lfloor \lambda T_{\text{OPT}}^{(1)} \rfloor; A_{y;\lambda}^{(1)})}{\text{OPT}(\lfloor \lambda T_{\text{OPT}}^{(1)} \rfloor)}$.

*Subcase 1.2:* $\lfloor \lambda T_{\text{OPT}}^{(1)} \rfloor > t^*$.

In this case, the relationship $\Delta\overline{\text{OPT}}(n) = \Delta\text{OPT}(n)$ still holds for $n \leq t^*$. Beyond this point, $\Delta\overline{\text{OPT}}(n)$ is constant. Due to the construction of Algorithm 3, we have:

$$\Delta\text{ALG}(n; A_{y;\lambda}^{(1)}) = \Delta\text{ALG}(n+1; A_{y;\lambda}^{(1)}),$$

while the offline cost decreases, i.e., $\Delta\text{OPT}(n) \geq \Delta\text{OPT}(n+1)$. It follows that

$$\frac{\Delta\text{ALG}(n; A_{y;\lambda}^{(1)})}{\Delta\text{OPT}(n)} \leq \frac{\Delta\text{ALG}(n+1; A_{y;\lambda}^{(1)})}{\Delta\text{OPT}(n+1)},$$

and consequently,

$$\frac{\text{ALG}(n; A_{y;\lambda}^{(1)})}{\text{OPT}(n)} = \frac{\sum_{i=1}^{n} \Delta\text{ALG}(i; A_{y;\lambda}^{(1)})}{\sum_{i=1}^{n} \Delta\text{OPT}(i)} \leq \frac{\sum_{i=1}^{n+1} \Delta\text{ALG}(i; A_{y;\lambda}^{(1)})}{\sum_{i=1}^{n+1} \Delta\text{OPT}(i)} = \frac{\text{ALG}(n+1; A_{y;\lambda}^{(1)})}{\text{OPT}(n+1)}.$$

Hence, the worst-case ratio is again achieved at $x = \lfloor \lambda T_{\text{OPT}}^{(1)} \rfloor$.

**Case 2:** $y < T_{\text{OPT}}$    Let $T_{\text{ML}} = \lceil T_{\text{OPT}}^{(2)}/\lambda \rceil$. According to Algorithm 3, the following holds:

$$\frac{\Delta\text{ALG}(n; A_{y;\lambda}^{(2)})}{\overline{\text{OPT}}(n)} = c', \quad \forall n \leq T_{\text{ML}},$$

and $\Delta\text{ALG}(n; A_{y;\lambda}^{(2)}) = 0$ for $n > T_{\text{ML}}$.

**Consistency:** When $n \leq t^*$, the augmented and actual optimal costs match, i.e., $\Delta\overline{\text{OPT}}(n) = \Delta\text{OPT}(n)$, which implies

$$\frac{\Delta\text{ALG}(n; A_{y;\lambda}^{(2)})}{\Delta\text{OPT}(n)} = c', \quad \forall n \leq t^*.$$

For $t^* < n < T_{\text{ML}}$, $\Delta\overline{\text{OPT}}(n)$ is constant. We can get that

$$\frac{\Delta\text{ALG}(n; A_{y;\lambda}^{(2)})}{\Delta\text{OPT}(n)} \leq \frac{\Delta\text{ALG}(n+1; A_{y;\lambda}^{(2)})}{\Delta\text{OPT}(n+1)},$$

and consequently,

$$\frac{\text{ALG}(n; A_{y;\lambda}^{(2)})}{\text{OPT}(n)} = \frac{\sum_{i=1}^{n} \Delta\text{ALG}(i; A_{y;\lambda}^{(2)})}{\sum_{i=1}^{n} \Delta\text{OPT}(i)} \leq \frac{\sum_{i=1}^{n+1} \Delta\text{ALG}(i; A_{y;\lambda}^{(2)})}{\sum_{i=1}^{n+1} \Delta\text{OPT}(i)} = \frac{\text{ALG}(n+1; A_{y;\lambda}^{(2)})}{\text{OPT}(n+1)}.$$

The competitive ratio is non-decreasing, with the worst case again occurring at $T_{\text{OPT}}$.

**Robustness:** Since $\Delta\text{OPT}(n) = 0$ for $n > T_{\text{OPT}}$, the worst case occurs when $T = \lceil T_{\text{OPT}}^{(2)}/\lambda \rceil$, in which the cost ratio of the online algorithm to the optimal offline algorithm is

$$\frac{\text{ALG}(\lceil T_{\text{OPT}}^{(2)}/\lambda \rceil; A_{y;\lambda}^{(2)})}{\text{OPT}(\lceil T_{\text{OPT}}^{(2)}/\lambda \rceil)} = \frac{\text{ALG}(\lceil T_{\text{OPT}}^{(2)}/\lambda \rceil; A_{y;\lambda}^{(2)})}{\text{OPT}(T_{\text{OPT}})}.$$

In summary, Algorithm 3 exhibits $\max\left\{ \frac{\text{ALG}(T_{\text{OPT}}; A_{y;\lambda}^{(1)})}{\text{OPT}(T_{\text{OPT}})}, \frac{\text{ALG}(T_{\text{OPT}}; A_{y;\lambda}^{(2)})}{\text{OPT}(T_{\text{OPT}})} \right\}$-consistency and $\max\left\{ \frac{\text{ALG}(\lfloor \lambda T_{\text{OPT}}^{(1)} \rfloor; A_{y;\lambda}^{(1)})}{\text{OPT}(\lfloor \lambda T_{\text{OPT}}^{(1)} \rfloor)}, \frac{\text{ALG}(\lceil T_{\text{OPT}}^{(2)}/\lambda \rceil; A_{y;\lambda}^{(2)})}{\text{OPT}(T_{\text{OPT}})} \right\}$-robustness.

$\square$

### F.3 Proof of Corollary 1

*Proof.* In the classic ski rental problem, the optimal offline cost is given by $\texttt{OPT}(T) = \min\{b, T\}$, and the critical threshold at which buying becomes optimal is $T_{\texttt{OPT}} = b$. For a randomized online algorithm $A$, the expected cost can be written as:

$$\Delta\texttt{ALG}(n; A_{y;\lambda}) = \sum_{i \in [n]} (i - 1 + b) \cdot p_i(\{1\}) + n \cdot \left(1 - \sum_{i \in [n]} p_i(\{1\})\right),$$

where $p_i(\{1\})$ denotes the probability of purchasing the item on day $i$.

Note that $\frac{\texttt{OPT}(T_{\texttt{OPT}})}{T_{\texttt{OPT}}} = 1$. Therefore, $\overline{\texttt{OPT}}(n) = 1$ for all $n \in \mathbb{N}^+$ in this setting. Let $k = \lfloor \lambda b \rfloor$ and $l = \lceil b/\lambda \rceil$. According to Eq. (20), the probability distributions under the proposed algorithm are given as follows:

(1) When $y \geq b$ (corresponding to the algorithm $A_{y;\lambda}^{(1)}$),

$$p_n(\{1\}) = \begin{cases} \left(\frac{b-1}{b}\right)^{k-n} \cdot \frac{1}{b\left(1-\left(1-\frac{1}{b}\right)^k\right)} & \text{if } n \leq k, \\ 0 & \text{otherwise.} \end{cases}$$

The corresponding target competitive ratio is $\frac{1}{1-\left(1-\frac{1}{b}\right)^k}$.

(2) When $y < b$ (corresponding to the algorithm $A_{y;\lambda}^{(2)}$),

$$p_n(\{1\}) = \begin{cases} \left(\frac{b-1}{b}\right)^{l-n} \cdot \frac{1}{b\left(1-\left(1-\frac{1}{b}\right)^l\right)} & \text{if } n \leq l, \\ 0 & \text{otherwise.} \end{cases}$$

The corresponding target competitive ratio is $\frac{1}{1-\left(1-\frac{1}{b}\right)^l}$.

**Consistency.** We first bound the competitive ratio when the predictor is correct.

**Bound 1:** For the algorithm $A_{y;\lambda}^{(1)}$,

$$\frac{\texttt{ALG}(T_{\texttt{OPT}}; A_{y;\lambda}^{(1)})}{\texttt{OPT}(T_{\texttt{OPT}})} = \frac{\texttt{ALG}(b; A_{y;\lambda}^{(1)})}{b} \leq \frac{\lfloor \lambda b \rfloor}{1 - (1 - (1/b))^{\lfloor \lambda b \rfloor}} \cdot \frac{1}{b} \leq \frac{\lfloor \lambda b \rfloor / b}{1 - e^{-\frac{\lfloor \lambda b \rfloor}{b}}} \leq \frac{\lambda}{1 - e^{-\lambda}}.$$

**Bound 2:** For the algorithm $A_{y;\lambda}^{(2)}$,

$$\frac{\texttt{ALG}(T_{\texttt{OPT}}; A_{y;\lambda}^{(2)})}{\texttt{OPT}(T_{\texttt{OPT}})} = \frac{\texttt{ALG}(b; A_n)}{b} \leq \frac{b}{1 - (1 - (1/b))^{\lceil b/\lambda \rceil}} \cdot \frac{1}{b} \leq \frac{1}{1 - e^{-\frac{\lceil b/\lambda \rceil}{b}}} \leq \frac{1}{1 - e^{-\frac{1}{\lambda}}} \leq \frac{\lambda}{1 - e^{-\lambda}},$$

where the last inequality follows from Lemma 19 in [3].

Therefore, the consistency bound for both cases is upper bounded by $\frac{\lambda}{1-e^{-\lambda}}$.

**Robustness.** We now consider the case where the prediction is inaccurate, and we evaluate robustness to prediction errors.

**Bound 1:** According to Theorem 2,

$$\frac{\texttt{ALG}(\lfloor \lambda T_{\texttt{OPT}} \rfloor; A_{y;\lambda}^{(1)})}{\texttt{OPT}(\lfloor \lambda T_{\texttt{OPT}} \rfloor)} = \frac{\texttt{ALG}(\lfloor \lambda T_{\texttt{OPT}} \rfloor; A_{y;\lambda}^{(1)})}{\lfloor \lambda T_{\texttt{OPT}} \rfloor} \leq \frac{1}{1 - (1 - (1/b))^{\lfloor \lambda b \rfloor}} \leq \frac{1}{1 - e^{-\frac{\lfloor \lambda b \rfloor}{b}}} \leq \frac{1}{1 - e^{-(\lambda - 1/b)}}.$$

**Bound 2:**

$$\frac{\texttt{ALG}(\lceil T_{\texttt{OPT}}/\lambda \rceil; A_{y;\lambda}^{(2)})}{\texttt{OPT}(T_{\texttt{OPT}})} \leq \frac{\lceil b/\lambda \rceil}{1 - (1 - (1/b))^{\lceil b/\lambda \rceil}} \cdot \frac{1}{b} \leq \frac{1/\lambda + 1/b}{1 - e^{-\frac{\lceil b/\lambda \rceil}{b}}} \leq \frac{1/\lambda + 1/b}{1 - e^{-\frac{1}{\lambda}}} \leq \frac{1 + 1/b}{1 - e^{-\lambda}},$$

where we use $l = \lceil b/\lambda \rceil \leq b/\lambda + 1$.

Hence, the robustness bound is at most $\frac{1+1/b}{1-e^{-\lambda}}$. $\qquad\square$

### F.4 Proof of Corollary 2

*Proof.* In the multi-shop ski rental problem, we choose shop $n$ when $y \geq T_{\text{OPT}}$ and shop 1 when $y < T_{\text{OPT}}$, while keeping all other aspects of Algorithm 3 unchanged. According to Theorem 1, this shop selection strategy yields a higher target competitive ratio compared to the case where all shops are simultaneously considered. As a result, the resulting algorithm serves as an upper bound for both the consistency bound and the robustness bound of Algorithm 3. In the following, we employ this selection strategy to prove the consistency bound and the robustness bound.

Let the optimal offline cost is given by $\text{OPT}(T) = \min\{b_s, T\}$, and the critical time is $T_{\text{OPT}} = b_s$. Note that $\frac{\text{OPT}(T_{\text{OPT}})}{T_{\text{OPT}}} = 1$. Therefore, $\overline{\text{OPT}}(n) = 1$ for all $n \in \mathbb{N}^+$ in this setting. Let $k = \lfloor \lambda b_s \rfloor$ and $l = \lceil b_1/\lambda \rceil$.

When $y \geq b_s$, the corresponding target competitive ratio is $\frac{r_s}{1-\left(1-\frac{b_s-r_s}{b_s}\right)^k} \leq \frac{r_s}{1-e^{-\frac{r_s k}{b_s}}} \leq \frac{r_s}{1-e^{-r_s(\lambda-\frac{1}{b_s})}}$. When $y < b$, the corresponding target competitive ratio is $\frac{1}{1-\left(1-\frac{1}{b_1}\right)^l} \leq \frac{1}{1-e^{-\frac{1}{\lambda}}}$.

**Consistency.** We first bound the competitive ratio when the predictor is correct.

**Bound 1:** For the algorithm $A_{y;\lambda}^{(1)}$,

$$\frac{\text{ALG}(T_{\text{OPT}}; A_{y;\lambda}^{(1)})}{\text{OPT}(T_{\text{OPT}})} = \frac{\text{ALG}(b_s; A_{y;\lambda}^{(1)})}{b_s} \leq \frac{r_s \cdot \lfloor \lambda b_s \rfloor}{1-e^{-\frac{r_s k}{b_s}}} \cdot \frac{1}{b_s} \leq \frac{\lfloor \lambda b_s \rfloor \frac{r_s}{b_s}}{1-e^{-\frac{\lfloor \lambda b_s \rfloor r_s}{b_s}}} \leq \frac{r_s \lambda}{1-e^{-\lambda r_s}}.$$

**Bound 2:** For the algorithm $A_{y;\lambda}^{(2)}$,

$$\frac{\text{ALG}(T_{\text{OPT}}; A_{y;\lambda}^{(2)})}{\text{OPT}(T_{\text{OPT}})} = \frac{\text{ALG}(b_s; A_n)}{b_s} \leq \frac{1}{1-e^{-\frac{1}{\lambda}}} \leq \frac{\lambda}{1-e^{-\lambda}} \leq \frac{r_s \lambda}{1-e^{-\lambda r_s}},$$

where the last inequality follows from $\lambda \leq r_s \lambda$.

Therefore, the consistency bound for both cases is upper bounded by $\frac{r_s \lambda}{1-e^{-\lambda r_s}}$.

**Robustness.** We now consider the case where the prediction is inaccurate, and we evaluate robustness to prediction errors.

**Bound 1:** According to Theorem 2,

$$\frac{\text{ALG}(\lfloor \lambda T_{\text{OPT}}^{(1)} \rfloor; A_{y;\lambda}^{(1)})}{\text{OPT}(\lfloor \lambda T_{\text{OPT}} \rfloor)} \leq \frac{r_s}{1-e^{-r_s(\lambda-\frac{1}{b_s})}} \leq \frac{b_1}{b_s} \cdot \frac{r_s}{1-e^{-r_s(\lambda-\frac{1}{b_s})}}.$$

**Bound 2:**

$$\frac{\text{ALG}(\lceil T_{\text{OPT}}^{(2)}/\lambda \rceil; A_{y;\lambda}^{(2)})}{\text{OPT}(T_{\text{OPT}})} \leq \frac{\lceil b_1/\lambda \rceil}{1-e^{-\frac{1}{\lambda}}} \cdot \frac{1}{b_s} \leq \frac{b_1}{b_s} \cdot \frac{1/\lambda + 1/b_1}{1-e^{-\frac{1}{\lambda}}},$$

where we use $l = \lceil b_1/\lambda \rceil \leq b_1/\lambda + 1$.

Hence, the robustness bound is at most $\frac{b_1}{b_s} \max\left\{ \frac{r_s}{1-e^{-r_s(\lambda-1/b_s)}}, \frac{1/\lambda+1/b_1}{1-e^{-1/\lambda}} \right\}$. $\qquad\square$

## G  Supplementary Numerical Results

In this section, we conduct numerical experiments to evaluate the performance of the SOAC algorithm and LA-SOAC algorithm. Specifically, we investigate the relationship between path selection complexity and the competitive ratio. Additionally, we perform further experiments using the SOAC algorithm to solve multi-shop, multi-slope problems. The experimental platform is an AMAX TR40-X4 server, configured with dual Intel Xeon Gold 6448H processors. The system is equipped with 512 GB of DDR5-4800 ECC memory, two 16 TB hard disk drives, two 960 GB solid-state drives, and four graphics processing units.

## G.1 Supplementary numerical results in Section 5

Building upon the experimental setup detailed in Section 5, we conducted more comprehensive numerical experiments to investigate the impact of various parameters on the performance of the LA-SOAC algorithm.

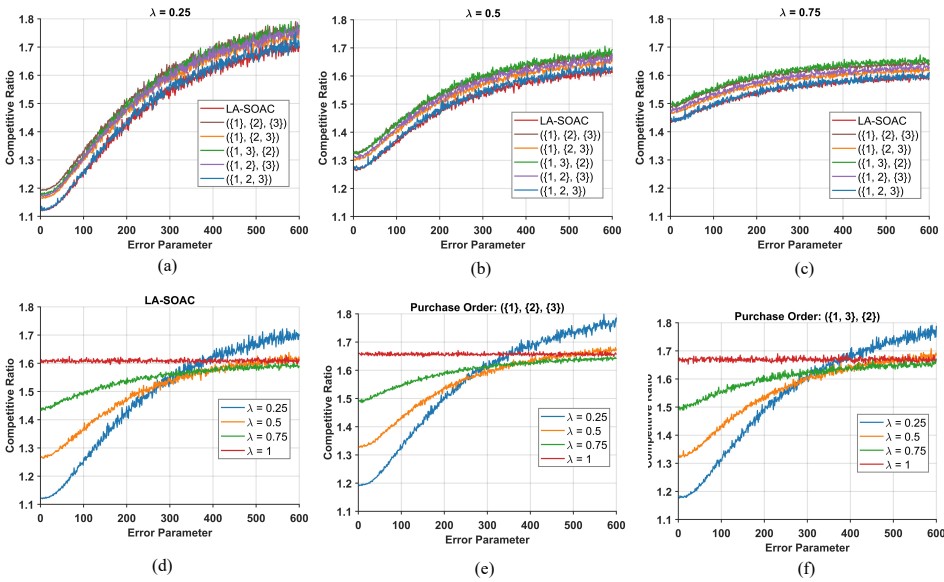

Figure 8: Average competitive ratio across different error parameter $\eta$ with $\lambda \in \{0.25, 0.5, 0.75, 1\}$ and $\delta = 0$.

**The impact of the hyperparameter $\lambda$.** Figures 8 illustrate the average competitive ratio of LA-SOAC under different values of $\lambda \in \{0.25, 0.5, 0.75, 1\}$ and varying error parameters $\eta$. The hyperparameter $\lambda$ captures the degree of trust placed in machine learning predictions: a smaller $\lambda$ corresponds to higher trust, while a larger $\lambda$ reflects greater caution. When predictions are highly accurate, a smaller $\lambda$ enables LA-SOAC to achieve performance close to the offline optimal. However, as the prediction error increases, over-reliance on inaccurate predictions (i.e., low $\lambda$) can lead to significant performance degradation. In contrast, with a larger $\lambda$, the algorithm becomes more conservative, and the competitive ratio rises more slowly as $\eta$ increases. These results show the trade-off between consistency and robustness controlled by $\lambda$, and demonstrate that LA-SOAC consistently performs at least as well as algorithms that follow a single deterministic path.

**The impact of biased error $\delta$.** Figure 9 illustrates the average competitive ratio of LA-SOAC under different bias levels $\delta \in \{50, 100, 150, 200\}$ and varying error parameters $\eta$, with fixed values of $\lambda = 0.25$ and $\lambda = 0.75$. The results reveal how the bias parameter $\delta$ influences the trade-off between exploiting accurate predictions and mitigating the risk of large errors. When the error level $\eta$ is small, a smaller bias $\delta$ leads to better competitive ratios, as it enables the algorithm to more aggressively utilize accurate predictions. Conversely, when $\eta$ is large, a larger $\delta$ becomes advantageous by hedging the impact of significant prediction errors, thereby improving its overall performance.

## G.2 Application of SOAC to the multi-shop ski rental problem

In the multi-shop ski rental problem, a skier must decide how to ski for an unknown duration $T$. The skier chooses a shop $i$ from a set of shops $\{1, 2, \ldots, m\}$ and can only rent or purchase ski equipment from that selected shop. Each shop $i$ offers a daily rental price $r_i$ and a purchase price $b_i$, where $r_i, b_i > 0$. It is assumed that $r_1 < r_2 < \cdots < r_m$ and $b_1 > b_2 > \cdots > b_m$, ensuring that shops with lower rental prices have higher purchase prices to avoid trivial decisions. The skier's goal is to select a shop and determine the optimal purchase time to minimize total costs, without changing shops midway.

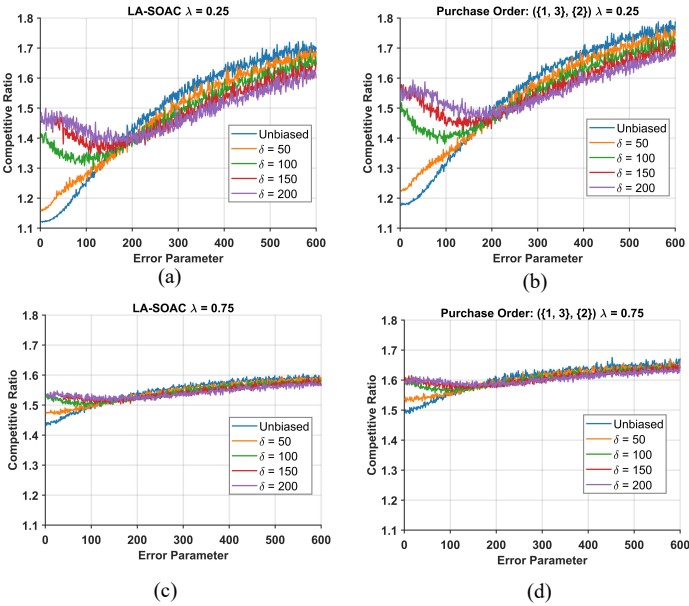

Figure 9: Average competitive ratio across different error parameter $\eta$ and bias $\delta \in \{50, 100, 150, 200\}$, with fixed values of $\lambda = 0.25$ and $\lambda = 0.75$.

We consider a scenario with three shops, each offering different rental and purchase prices. The rental and purchase prices at these shops are as follows: shop 1 with $r_1 = 1$ and $b_1 = 594$, shop 2 with $r_2 = 1.2$ and $b_2 = 576$, and shop 3 with $r_3 = 1.3$ and $b_3 = 560$. Based on the calculated probabilities, the likelihood of selecting each shop is determined: the probability of choosing shop 1 is $87.9\%$, the probability for shop 2 is $0\%$, and the probability for shop 3 is $12.1\%$. This indicates that the user is only likely to choose between shop 1 and shop 3. The daily purchase probabilities for these two shops are illustrated in Figure 10(a). We then compute the probability of choosing each shop for different values of $b_1$ by varying the purchase price $b_1$ of shop 1 while keeping the prices of the other shops constant, as shown in Figure 10(b). The results indicate that as $b_1$ increases, the probability of selecting shop 1 decreases, while the probability of selecting shop 2 increases. Additionally, the competitive ratio rises. This occurs as a higher purchase price for shop 1 increases the risk associated with selecting it, as the potential rental benefit diminishes.

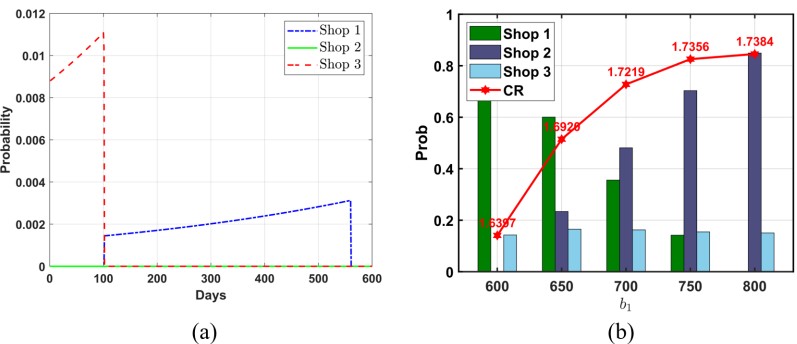

Figure 10: (a) Probability of purchase for each shop in the three-shop ski rental problem. Shop 1 has a purchase price $b_1 = 594$ and a rental price $r_1 = 1$, shop 2 has a purchase price $b_2 = 576$ and a rental price $r_2 = 1.2$, and shop 3 has a purchase price $b_3 = 560$ and a rental price $r_3 = 1.3$. (b) The impact of the competition ratio and the probability of selecting each shop after adjusting the purchase price of shop 1.

### G.3 Application of SOAC to the multi-slope ski rental problem

The multi-slope ski rental problem considers a scenario where a user needs to utilize a resource for an unknown duration. The cost of resource usage is determined by multiple *states* (or *slopes*), denoted as $\mathcal{S} = \{0, 1, \ldots, k\}$. Each state $i$ is associated with a purchase cost $b_i$ and a rental rate $r_i$, which typically satisfy the conditions $b_0 < b_1 < \cdots < b_k$ and $r_0 > r_1 > \cdots > r_k$, with $b_0 = 0$. State transitions are constrained to be forward-only, meaning that one cannot move backward or skip states. The objective is to determine the optimal state (or slope) to minimize the cost at a given time $t$.

A randomized profile is defined as the probability vector $p(t) = (p_0(t), \ldots, p_k(t))$, where $p_i(t)$ represents the probability of being in state $i$ at time $t$. To illustrate how SOAC can be applied to this problem, consider an example inspired by [19]. Assume there are three states with purchase and rental costs defined as $b = (b_0, b_1, b_2)$ and $r = (r_0, r_1, r_2)$, respectively. This corresponds to the CSR problem, where a fixed purchase path is given by $\sigma = (S_0, S_1, S_2, S_3)$. For each state $i$, the associated costs are defined as: $f(S_0) = g(S_0) = 0$, $f(S_1) = b_1 - b_0$, $g(S_1) = r_0 - r_1$, $f(S_2) = b_2 - b_1$, $g(S_2) = r_1 - r_2$, $f(S_3) = \infty$, $g(S_3) = 0$. For example, consider the following configurations of $b$ and $r$

- Example 1: $b = (0, 0.5, 0.9)$, $r = (2, 0.5, 0.1)$,
- Example 2: $b = (0, 0.5, 0.7)$, $r = (2, 0.5, 0.3)$,
- Example 3: $b = (0, 0.5, 0.55)$, $r = (2, 0.5, 0.45)$.

By applying the SOAC algorithm to these examples, the probabilities of being in each of the three states can be computed, as illustrated in Figure 11. These results are consistent with the experimental results in [19].

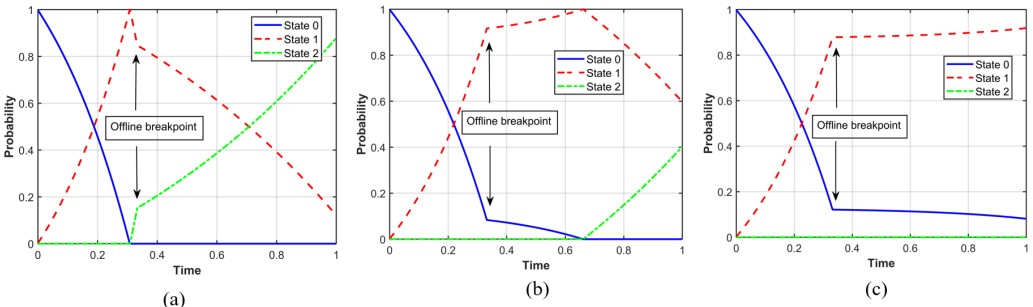

Figure 11: Illustrative examples demonstrating the application of the SOAC algorithm to solve the multi-slope problem. (a) Example 1: $b = (0, 0.5, 0.9)$, $r = (2, 0.5, 0.1)$; (b) Example 2: $b = (0, 0.5, 0.7)$, $r = (2, 0.5, 0.3)$; (c) Example 3: $b = (0, 0.5, 0.55)$, $r = (2, 0.5, 0.45)$.

### G.4 Supplementary numerical results of SOAC and LA-SOAC

Consider a scenario in which a company requires specific software to support both its daily operations and long-term growth. However, the duration for which these software applications will be used remains uncertain. The market offers a diverse range of software options, which can be acquired either through a perpetual purchase or on-demand leases. It is important to note that some software applications are interrelated, and bundling multiple products from the same vendor may result in additional discounts. The company's business needs may require anywhere from 6 to 16 software applications. For the sake of simplicity, we assume each application is priced at one unit for rental. Furthermore, discounts are available only for certain combinations of the first six software packages. A summary of the purchase prices for these 16 software packages, including the discounted combination prices, is provided in Table 1.

For applications that do not qualify for combination discounts, Lemma 3 establishes that the optimal strategy is to purchase them individually. As a result, there are eight distinct purchase paths. These paths are summarized in Table 2, which excludes applications 7 through 16. These applications are purchased individually and are incorporated into the table based on the buy-to-rent ratio.

Table 1: Purchase prices of 16 software applications and corresponding combination purchase prices.

| Items | {1} | {2} | {3} | {4} | {5} |
|---|---|---|---|---|---|
| Prices | 202 | 535 | 960 | 370 | 206 |
| **Items** | {6} | {7} | {8} | {9} | {10} |
| **Prices** | 171 | 800 | 120 | 714 | 221 |
| **Items** | {11} | {12} | {13} | {14} | {15} |
| **Prices** | 556 | 314 | 430 | 558 | 187 |
| **Items** | {16} | {1, 2} | {2, 3, 4} | {4, 5} | {5, 6} |
| **Prices** | 472 | 663.3 | 1715.8 | 524.16 | 327.99 |

We computed the probability of selecting the optimal purchase path for firms acquiring between 6 and 16 applications. The results indicate that the decision ultimately narrows down to either path 1 or path 2, as shown in Figure 12(a). As the number of applications increases, the probability of choosing path 1 decreases, while the probability of selecting path 2 rises. This shift leads to a corresponding reduction in the competitive ratio. The underlying reason for this change is that as the path selection process becomes more deterministic, the decisions in the CSR problem are simplified. This reduction in decision uncertainty results in a lower competitive ratio.

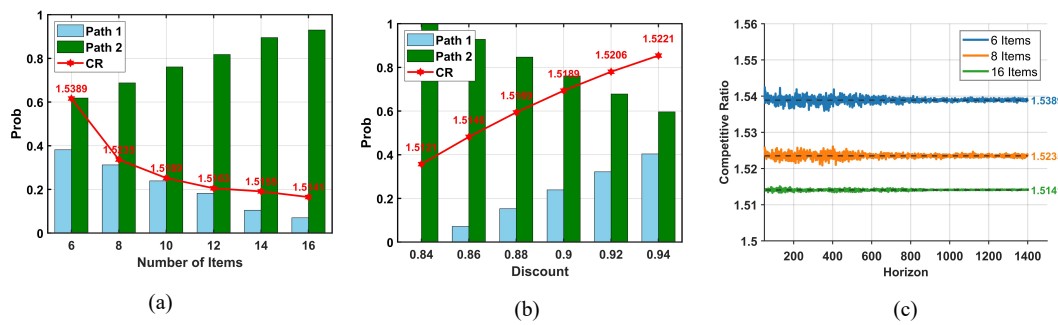

(a)  (b)  (c)

Figure 12: Analysis of path selection probabilities and competitive ratios. Path 1 is defined as $(\{5, 6\}, \{1\}, \{2, 3, 4\})$, and Path 2 as $(\{5, 6\}, \{1, 2\}, \{4\}, \{3\})$. The remaining items are purchased independently and assigned to paths based on the buy-to-rent ratio. (a) Illustrates the effect of increasing the number of undiscounted items on the path selection probability and the competitive ratio. (b) Illustrates the case where the number of items is fixed at 10, and different discounts are applied to the combination $\{1, 2\}$. The discount for this combination is defined as the sum of the purchase prices of items 1 and 2, multiplied by the discount factor. (c) Validation of theoretical vs. experimental competitive ratios.

Furthermore, we fixed the selection of the first 10 items, modified the price of the combination {1,2}, and kept the prices of the remaining items unchanged. We then computed the probabilities of selecting path 1 and path 2 for various discount factors for the combination {1,2}, along with the corresponding competitive ratios, as shown in Figure 12(b). Similar to the conclusions drawn in Figure 12(a), an increase in the discount leads to greater uncertainty in the path selection process. This increased uncertainty leads to a more complex decision-making process in the CSR problem, which in turn results in a higher competitive ratio.

Based on the same pricing settings as in Figure 12(a), we further evaluate the theoretical performance of the SOAC algorithm. Specifically, we consider scenarios with 6, 8, and 16 items, where purchase decisions are made according to the randomized probabilities computed by Algorithm 2. We then evaluate the empirical performance of the algorithm by computing the empirical average competitive ratio for varying end times of the game. As illustrated in Figure 12(c), the experimental competitive ratios align well with their theoretical counterparts, validating the accuracy of the algorithm. Moreover, we observe that the fluctuation in the experimental competitive ratio is more pronounced when the end time is shorter. This is because randomized experiments conducted over shorter durations tend to exhibit greater variance. Additionally, the fluctuation decreases as the number of items increases.

Table 2: Purchase paths for multiple software applications, with software 7 through 16 omitted for convenience.

| Path 1-4 | Path 5-8 |
|---|---|
| $(\{5,6\}, \{1\}, \{2,3,4\})$ | $(\{6\}, \{1\}, \{5\}, \{4\}, \{2\}, \{3\})$ |
| $(\{5,6\}, \{1,2\}, \{4\}, \{3\})$ | $(\{5,6\}, \{1\}, \{4\}, \{2\}, \{3\})$ |
| $(\{6\}, \{1\}, \{4,5\}, \{2\}, \{3\})$ | $(\{6\}, \{1\}, \{5\}, \{2,3,4\})$ |
| $(\{6\}, \{5\}, \{1,2\}, \{4\}, \{3\})$ | $(\{6\}, \{4,5\}, \{1,2\}, \{3\})$ |

This is attributed to the fact that with more items, the decision paths become more deterministic, thereby reducing the variance, as also reflected in Figure 12(a).

For the learning-augmented algorithm, we let the actual number of days, $T$, be uniformly distributed within the region $[1, 4T_{\text{off}}]$. The predicted number of days, $y$, is set to $y = T + \epsilon$, where the simulated error $\epsilon$ follows a normal distribution with a mean of $\delta$ and a standard deviation $\eta$. For each standard deviation $\eta$, 10,000 samples are randomly sampled in this study, and their average competitive ratios are calculated. As illustrated in Figure 13, we evaluated the average competitive ratio of the `LA-SOAC` algorithm for 10 items across varying error parameters. Our analysis focused on how this ratio changes with the parameter $\lambda$, and we also compared it to the average competitive ratio achieved by selecting a single path. The results indicate that `LA-SOAC` effectively balances consistency and robustness. Specifically, when the algorithm increasingly relies on the predicted values and the prediction error is minimal, its average competitive ratio approaches that of the offline optimal solution. Furthermore, `LA-SOAC` demonstrates superior performance compared to strategies that commit to a single path.

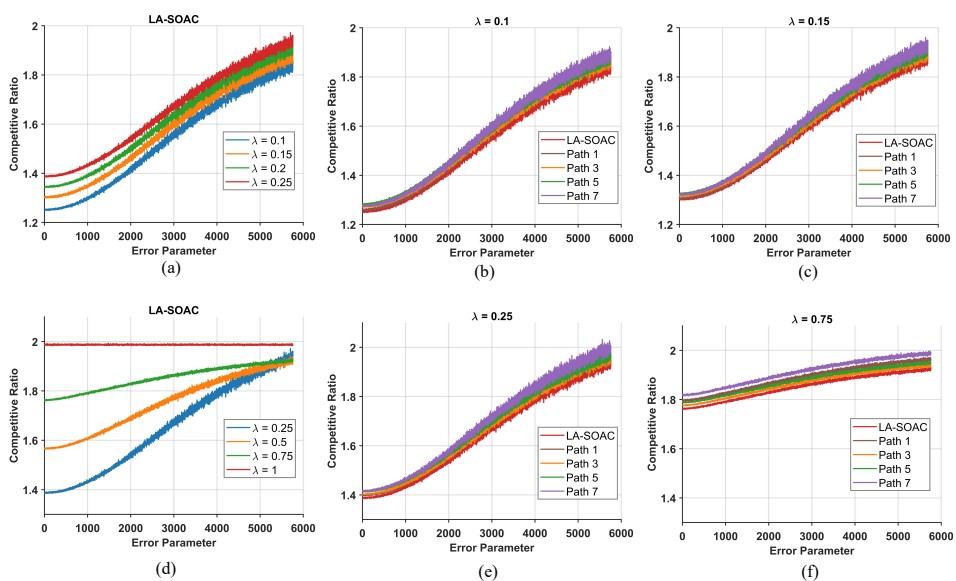

Figure 13: Average competitive ratio across different error parameters $\eta$ with 10 items.

### G.5 Runtime Analysis

In this subsection, we provide a detailed runtime evaluation to complement the scalability experiments in Appendix G.4 and to demonstrate the practical feasibility of `SOAC`. All experiments were conducted on an AMAX TR40-X4 server (detailed specifications in Appendix G) using CPU-only computation without CUDA acceleration.

We study runtime as the number of discount combinations increases. All experiments were run with high precision ($10^{-6}$) and practical precision ($10^{-4}$). When reducing the precision from $10^{-6}$ to $10^{-4}$, the number of iterations decreases to approximately $7\%$ of the high-precision runs.

Table 3: Runtime scaling with purchase paths. Entries marked with * are extrapolated estimates.

| Combos | Purchase Paths | Time/Iter | Iterations → Runtime | |
|---|---|---|---|---|
| | | | High Precision ($10^{-6}$) | Practical Precision ($10^{-4}$) |
| 4 | 10 | 0.285s | 436 → 2.1 min | 10 → 2.85 s |
| 7 | 30 | 2.97s | 492 → 24.3 min | 36 → 2.4 min |
| 8 | 60 | 11.25s | 560 → 1.75 h | 35 → 11.1 min |
| 10 | 120 | 69.29s | 525 → 10.1 h | 40 → 48 min |
| 12 | 240 | 451.67s | 500* → 2.6 d | 35* → 4.39 h |
| 14 | 405 | 1972.02s | 500* → 11.4 d | 35* → 19.2 h |

The per-iteration time complexity is polynomial in the number of purchase paths. A cubic fit of the form

$$y = 0.0000253x^3 + 0.00180x^2 - 0.00746x + 0.334$$

achieves a good fit quality (residual sum of squares (RSS = 0.74) ), confirming theoretical polynomial behavior.

The total runtime increases rapidly as the number of discount combinations grows. However, the above experiments are based on adversarially constructed discount combinations designed to maximize overlap and diversity among purchase options, which tend to produce a large number of purchase paths. In practical scenarios, the structure of discount bundles is often more regular and aggregated. For example, practical combinations may include larger bundle sets, which significantly reduce the number of distinct purchase paths and, consequently, the total runtime.

To illustrate this difference, we further compare synthetic adversarial (many small, disjoint combinations) and practical realistic (fewer, larger combinations) scenarios. An additional 8-item case study is shown in Table 4. It can be seen that realistic structures lead to substantially fewer purchase paths and much shorter computation times. Moreover, the computation of purchase probabilities is performed offline during the planning phase, and the resulting online buy-or-rent decisions can be executed instantaneously without any runtime overhead.

Table 4: Runtime comparison under adversarial vs. realistic combination structures.

| Setting | Discount Combos | Purchase Paths | Total Time |
|---|---|---|---|
| Adversarial | 10 | 61 | 7.18 min |
| | 20 | 187 | 2.2 h |
| | 25 | 352 | – |
| | 30 | 524 | – |
| Realistic | 10 | 11 | 17.6 s |
| | 20 | 21 | 42.2 s |
| | 25 | 26 | 1.05 min |
| | 30 | 32 | 1.61 min |

