# OpenReview forum: "Combinatorial Ski Rental Problem: Robust and Learning-Augmented Algorithms"
_NeurIPS.cc/2025/Conference — NeurIPS 2025 poster_

### Official Review · Reviewer_BAW1 · 2025-06-29

**Clarity:** 3
**Significance:** 3
**Originality:** 3
**Rating:** 5
**Confidence:** 3

**Summary:**

In the combinatorial ski rental problem, the goal is to either rent or buy a set of items, that are necessary to ski, for a complete skiing season with an unknown duration. At each day of the skiing season, we can decide to either rent or buy the necessary items that have not been bought yet, with the goal of minimizing the total cost at the end of the skiing season. The cost function for buying a subset of items is submodular, giving the problem a combinatorial nature. On the other hand, the cost for renting a subset of items is just the sum of the individual renting costs. Algorithms for this problem are analysed against the optimal offline solution, which knows the duration of the skiing season upfront. This problem generalizes multiple variants of ski rental that have been studied in the literature. The closest related problem in the literature is the problem variant with upgrading [32]: In this variant, the cost for buying a set of items $S$ at time $t$ is $f(S \cup P) - f(P)$ where $P$ is the set of items bought in previous steps, i.e., we only pay the marginal costs. In contrast, in the variant considered in this paper, the cost is just $f(S)$. The difference between these two variants is reflected in the competitive ratio: For the variant with upgrading, [32] shows a (randomized) competitive ratio of e/(e-1). For the variant of this paper, the competitive ratio can get worse depending on the given cost functions.

The main results of this paper are (1) a randomized algorithm that achieves the optimal competitive ratio for each fixed pair of cost functions and (2) a learning-augmented algorithm that receives a prediction on the length of the skiing season and achieves improved results when this prediction is somewhat accurate, while still giving worst-case guarantees.  The authors first observe that a deterministic strategy can be represented by a \emph{purchase path} (sequence of item bundles to be bought) and \emph{purchase times} (points in time at which the purchases of the purchase path should be made). Consequently, a randomized strategy can be interpreted as a distribution over purchase paths and purchase time pairs. To achieve the first main result, the authors first show how to compute the optimal purchase time distribution for a fixed purchase path by tracking how the optimal solution cost evolves with an increasing time horizon and bounding the expected algorithmic cost against this increase . Afterwards, they use the insights from this algorithm to design an algorithm that simultaneously computes the optimal purchase path distribution and purchase time distribution by using a projected gradient descent method. Finally, they show how to integrate predictions into this algorithmic framework and give empirical experiments.

**Questions:**

* Are there hardness results justifying the need for an exponential running time?

**Ethical Concerns:**

["NO or VERY MINOR ethics concerns only"]

**Final Justification:**

As outlined win my original review, I think that the paper is a strong contribution.  In my opinion, the main strength of the paper are:

* the unification of existing results on (learning-augmented) ski-rental variants,
* new results for a generalisation of previously studied ski-rental variants,
* An algorithm that achieves the best-possible competitive ratio for every instance (characterised by the set of items, the sub modular purchase function, and the rental prices) and, thus, maintains its optimality also for special cases.

My main concern (also shared by the other reviewers) is  the exponential running time.  The author’s rebuttal addresses these concerns by using the following main arguments:

* For a fixed number of purchase paths, the complexity is (pseudo)-polynomial.
* For previously studied special cases, a (pseudo)-polynomial running time is achieved.
* Empirical experiments indicate scalability in practice.

In my opinion, the first two arguments are sufficient to keep the results strong and interesting from a theoretical point of view.  Therefore, I would like to maintain my original score. However, I hope that the authors add a more extensive discussion of the running time to the final version of the paper.

**Limitations:**

yes

**Quality:**

3

**Strengths And Weaknesses:**

Strength:

* The presented randomized algorithm unifies several existing results for multiple ski rental variants. In a similar way, the learning-augmented algorithm recovers, and thereby unifies, results for multiple previously studied variants. This seems like a significant step towards a general framework for solving problems of this type. In particular, because the theoretical results give the best-possible competitive ratio \emph{for each} fixed pair of cost functions, which makes them applicable to special cases without loss.
* In my opinion, the algorithmic ideas for computing the purchase probabilities for a fixed purchase path (Section 3.1) are very elegant. They use a characterization of the offline optimal solution to track how the optimal cost increases with an increasing time horizon. In order to find $\alpha$-approximative purchase probability, they locally bound the (expected) increase in the algorithm's cost by $\alpha$ times the increase in the optimal cost. Tracking how the optimal solution evolves over time in this way, is a very nice idea. Similarly, the insight that the purchase path probabilities can be computed with the projected gradient descent method is nice. In total, the paper seems to bring multiple new ideas to the area of ski rental type problems.

Weaknesses:
* The running time of the presented algorithm can be exponential in the input size. In the conclusion, the authors state that this is due to the combinatorial nature of the problem. It would be nice to formally underline this statement with hardness results.
* The instances (item number and cost functions) considered in the empirical part seem a bit arbitrary. They are still useful as a proof of concept. However, it is unclear what the empirical results mean for practical instances.

---

> ### Author Rebuttal · Authors · 2025-07-30
>
> Thanks for your positive comments and helpful suggestions.
>
> > **W1: The running time of the presented algorithm can be exponential in the input size. In the conclusion, the authors state that this is due to the combinatorial nature of the problem. It would be nice to formally underline this statement with hardness results.**
>
> > **Q: Are there hardness results justifying the need for an exponential running time?**
>
> **A1/Q:** Thank you for this important suggestion regarding the computational complexity analysis. The exponential complexity arises from the *combinatorial input space* (number of valid purchase paths), not algorithmic inefficiency. Our algorithm achieves *polynomial-time per-iteration complexity* for a fixed number of purchase paths and *matches existing polynomial bounds* in previously studied special cases.
>
> We provide a detailed response addressing both theoretical foundations and practical implications.
>
> ### 1. Fundamental Algorithm Efficiency
>
> When the number of purchase paths $K$ is fixed, the overall complexity of our method is provably polynomial.
>
> - **Single Path (OAC)**:
>    When the purchase path is fixed, the OAC algorithm uses binary search to compare $T_A$ with $T_{\mathrm{OPT}}$. In each iteration, the search region is halved until the stopping criterion $\varepsilon$ is satisfied, resulting in a complexity of $O\left(T_{\mathrm{OPT}} \log \frac{1}{\varepsilon}\right).$
>
> - **Multiple Paths (SOAC)**:
>
>   For SOAC, each iteration of projected gradient descent involves:
>
>   - Gradient computation over $K$ purchase paths: $O(K)$
>   - Competitive ratio evaluation for each path via OAC: $O\left(K \cdot T_{\mathrm{OPT}} \log \frac{1}{\varepsilon}\right)$
>   - Gradient update: $O(K)$
>   - Simplex projection: $O(K \log K)$
>
> Combining these steps, the per-iteration complexity of SOAC is $O\left(K \cdot \left(T_{\mathrm{OPT}} \log \frac{1}{\varepsilon} + \log K\right)\right)$, where $K$ is the number of purchase paths.
>
> The algorithm itself is highly efficient—exponential growth comes from input structure, not computational design.
>
> ### 2. Exponential Complexity: Worst-Case Analysis
>
> In the worst-case scenario where all item combinations have discounts:
>
> - **Naive bound**: $O(m! \cdot B_m)$ paths (super-exponential)
> - **Our optimization**: Lemma 3 reduces this to $O(B_m)$ by proving optimal purchase orders follow buy-to-rent ratios
>
> Most practical discount structures are sparse, dramatically reducing the number of valid purchase paths from theoretical worst-case scenarios.
>
> ### 3. Empirical Validation: Scalability in Practice
>
> **Hardware**: Intel Core i9-14900HX processor, 16 items with bundle discounts
>
> | Purchase Paths        | Runtime per Iteration |
> | --------------------- | --------------------- |
> | 10 paths              | 0.48s                 |
> | 30 paths              | 4.34s                 |
> | 60 paths              | 22.57s                |
> | 120 paths  (10-combo) | 137.0s                |
> | 240 paths  (12-combo) | 909.0s                |
>
> In our experiment with 16 items and 12 discounted combinations (resulting in 240 purchase paths), each gradient iteration took only 909 seconds, which is sufficient for practical deployment in real-world applications.
>
> ### 4. Polynomial-Time Performance for Structured Cases
>
> Our framework *recovers existing polynomial bounds* for well-studied special cases:
>
> - **Multi-Shop**: $O(m \cdot (T_{\text{OPT}} \log(1/\varepsilon) + \log m))$, where $m$ is the number of shop
> - **Multi-Slope**: $O(T_{\text{OPT}} \log(1/\varepsilon))$
> - **Multi-Commodity**: $O(T_{\text{OPT}} \log(1/\varepsilon))$
> - **Classic Ski Rental**: $O(T_{\text{OPT}} \log(1/\varepsilon))$
>
> This confirms that the computational cost is significantly lower when the structure of the CSR problem is less general.
>
> ---
>
> > **W2: The instances (item number and cost functions) considered in the empirical part seem a bit arbitrary. They are still useful as a proof of concept. However, it is unclear what the empirical results mean for practical instances.**
>
> **A2:**  Thank you for the insightful question.
>
> In the main text, we use a three-item setup in our main experiments to clearly illustrate the theoretical properties of the SOAC algorithm, whose analysis involves complex formulas. This simplified setup enables transparent demonstration of the algorithm’s mechanics. For the LA-SOAC algorithm, we intentionally use three items to exhaustively enumerate all purchase paths, allowing precise evaluation of the consistency-robustness trade-off—a core contribution of our work.
>
> Our empirical evaluation also serves to validate the theoretical guarantees. For example, in Figure 2(b) and Figure 12(c), the black lines indicate the theoretical competitive ratios, while the shaded areas depict the empirical performance under randomized decisions. Their close alignment confirms that the algorithm performs in practice as predicted by theory.
>
> Additionally, larger-scale experiments can be found in Appendix G.4, where we evaluate scenarios with up to 16 software applications using realistic pricing structures that include combination discounts. These experiments show that as the algorithm's decisions become more deterministic, the competitive ratio improves. We also analyze how discount factors affect path selection probabilities and show that the LA-SOAC algorithm effectively balances consistency and robustness across varying prediction errors, consistently outperforming single-path strategies. All supplementary code is included in the submitted materials (folders: `SOAC_sup_material` and `LA-SOAC_material`) to ensure full reproducibility.

---

> ### Comment · Area_Chair_JNus · 2025-08-04
> **Please respond to author rebuttal**
>
> Dear Reviewer BAW1,
>
> Would you please check if the authors address your major concerns, and if you have further comments?
>
> Regards,
>
> AC

---

> ### Comment · Reviewer_BAW1 · 2025-08-04
>
> Thank you to the authors for their helpful response. This addresses my concerns, and I will maintain my positive score.

---

> > ### Author Response · Authors · 2025-08-04
> >
> > Thank you for your detailed examination and encouraging feedback. We are grateful for the considerable time and attention you invested in reviewing our work.

---

### Official Review · Reviewer_XvPp · 2025-06-29

**Clarity:** 3
**Significance:** 3
**Originality:** 3
**Rating:** 5
**Confidence:** 4

**Summary:**

This paper introduces the Combinatorial Ski Rental (CSR) problem, a generalization of classic ski rental where multiple items can be rented/purchased individually or in bundles, with submodular purchase costs and additive rental rates. The key contributions include an optimal randomized online algorithm achieving the best possible competitive ratio for CSR (which is called SOAC). The authors also study the learning-augmented setting, leveraging predictions to balance consistency (accuracy with good predictions) and robustness (worst-case guarantees). Finally, the authors also did numerical validation of SOAC/LA-SOAC superiority over baselines.

**Questions:**

Could you comment on what will happen if the rental rate is also a submodular function? Is the problem becoming much harder? When the rental rate is additive, the offline optimal solution is equivalent to minimizing a non-monotone submodular function. When the rental rate is submodular, as far as I  can see, the offline problem no longer seems in P, but I am not sure how hard both the online and offline versions are.

**Ethical Concerns:**

["NO or VERY MINOR ethics concerns only"]

**Final Justification:**

I believe that this paper studies an interesting problem, although the obtained result does not seem plausible. The authors also address my questions during the rebuttal. I decide to keep my score. I also went through other reviewers' comments, and I agreed with some of the weaknesses that I also mentioned in the review. In summary, I am fine with both rejection and acceptance.

**Quality:**

3

**Strengths And Weaknesses:**

Strengths
1. I like the proposed CSR model; it elegantly unifies several ski rental variants and addresses the open non-upgradable case (where prior work assumed upgradable purchases). The submodular cost model captures real-world bundle discounts. I also appreciate that the authors draw Figure 1 to describe the related works, and it is helpful.

2. I usually work on offline approximation algorithms. But, as far as I can see, the algorithm SOAC is novel, using amortized cost tracking against an offline optimal benchmark and convex optimization over purchase paths via an augmented path construction. Thus, I think that the technical contribution is sufficient.

3. The paper is well-written and organized. I enjoy reading the paper.

Weakness:

 1. SOAC’s runtime scales exponentially with the number of super items, which is stated in section 6. While the authors note that real-world discounts are often sparse, scalability for large item sets remains a concern.

2. The additive rental rates and submodular costs are well-justified but may not hold in all applications (e.g., interdependent rental costs).

Summary:

This is a good paper, and I believe that it should be accepted.

---

> ### Author Rebuttal · Authors · 2025-07-30
>
> We thank the reviewer for the encouraging feedback and helpful suggestions.
>
> > **W1: SOAC’s runtime scales exponentially with the number of super items, which is stated in section 6. While the authors note that real-world discounts are often sparse, scalability for large item sets remains a concern.**
>
> **A1:** We thank the reviewer for your positive and insightful comments. We agree that, in the most general case, the computational complexity arising from large item sets remains a limitation of the current paper; however, we would like to emphasize that addressing this limitation is not the focus of this work and does not necessarily pose a practical concern.
>
> ### 1. Scope of Contribution: Information-Theoretic Guarantees in Online Decision-Making
>
> Our primary contribution targets the *information-theoretic challenges* of online decision-making, specifically bounding competitive ratios under future input uncertainty. This fundamentally differs from developing efficient approximation algorithms for offline optimization problems. Once purchasing probabilities are computed initially, the online buy-or-rent decisions within CSR execute without computational bottlenecks.
>
> ### 2. Algorithmic Complexity Reduction
>
> We have proactively addressed scalability through several key optimizations:
>
> - **Theoretical breakthrough**: Lemma 3 proves that optimal purchase orders must follow descending buy-to-rent ratios, reducing complexity from $O(m!  B_m)$ to $O(B_m)$, where $B_m$  is the Bell number
> - **Real-world sparsity**: Most practical discount structures are sparse, keeping the number of valid purchase paths manageable
>
> ### 3. Per-iteration complexity is polynomial
>
> Each iteration runs in $O\left(K \cdot \left(T_{\text{OPT}} \log(1/\varepsilon) + \log K \right)\right)$, which is polynomial in all parameters when the number of purchase paths $K$ is fixed. The exponential worst-case scenario (all combinations discount-eligible) is unrealistic.
>
> ### 4. Simpler CSR Structures Lead to Polynomial Complexity
>
> Our framework automatically recovers efficient performance for well-studied variants:
>
> - **Multi-Shop:** $O\big(m \cdot (T_{\text{OPT}} \log(1/\varepsilon) + \log m)\big)$
> - **Multi-Slope/Multi-Commodity:** $O(T_{\text{OPT}} \log(1/\varepsilon))$
> - **Classic Ski Rental:** $O(T_{\text{OPT}} \log(1/\varepsilon))$
>
> This confirms that the computational cost is significantly lower when the structure of the CSR problem is less general.
>
> ### 5. Empirical Evidence for Practical-Scale Scalability
>
> Our runtime analysis on a laptop with Intel Core i9-14900HX processor (16 items) shows scalable performance across different path complexities:
>
> - **10 paths**: 0.48s per gradient iteration
> - **30 paths**: 4.34s per gradient iteration
> - **60 paths**: 22.57s per gradient iteration
> - **120 paths (10-combo)**: 136.97s per gradient iteration
> - **240 paths (12-combo)**: 908.96s per gradient iteration
>
> While we acknowledge that scalability for large item sets remains a concern, these 10-12 combo configurations already capture the complexity range of most real-world discount scenarios.
>
> ### 6. Potential for Further Optimization
>
> A critical observation from our experiments: *only a small subset of candidate paths appear in optimal solutions*. This opens clear avenues for future improvements:
>
> - Early elimination of suboptimal paths
> - Pruning and approximation strategies
> - Avoiding exhaustive candidate enumeration
>
> ---
>
> > **W2: The additive rental rates and submodular costs are well-justified but may not hold in all applications (e.g., interdependent rental costs).**
>
> > **Q: Could you comment on what will happen if the rental rate is also a submodular function? Is the problem becoming much harder? When the rental rate is additive, the offline optimal solution is equivalent to minimizing a non-monotone submodular function. When the rental rate is submodular, as far as I can see, the offline problem no longer seems in P, but I am not sure how hard both the online and offline versions are.**
>
> **A2/Q:** Thank you for the insightful question. We analyze the impact of extending from additive to submodular rental rates from both offline and online perspectives.
>
> **Conclusion**: submodular rental rates do not affect our online algorithm's optimality, but do increase offline computational complexity.
>
> ### 1. Offline Setting: From Polynomial to NP-Hard
>
> **Additive rental rates (current model):** When rental rates are additive, the offline objective becomes: $\min_{S \subseteq \mathcal{M}} \{f(S) + T \cdot g(\mathcal{M} \setminus S)\} = \min_{S \subseteq \mathcal{M}} \{f(S) - T \cdot g(S)\} + T \cdot g(\mathcal{M})$
>
> This reduces to *non-monotone submodular minimization*, for which efficient polynomial-time approximation algorithms exist.
>
> **Submodular rental rates:** When $g$ becomes submodular, $g(\mathcal{M} \setminus S)$ becomes supermodular in $S$. As a result, the objective becomes the sum of a submodular and a supermodular function, a problem class that is generally NP-hard. Therefore, the offline problem likely requires exponential time to solve exactly and no longer clearly lies in P.
>
> ### 2. Online Setting: Computational Complexity Unchanged
>
> Our online algorithm's structure and optimality *remain completely unaffected* by submodular rental rate assumptions. Here's why:
>
> - Purchase decisions depend on buy-to-rent ratios of items and super-items
> - These ratios are correctly captured in our model for any rental rate function
> - The convex optimization framework applies universally
>
> Therefore, our algorithm can solve the problem optimally for *any* independent rental rate function, including submodular cases, where “*independent*” refers to rental costs that remain fixed regardless of time or item combinations.
>
> **Source of computational complexity:** The primary computational bottleneck lies in the combinatorial explosion of purchase paths, which exists independently of rental rate structure—not in the rental computation itself.

---

> > ### Comment · Reviewer_XvPp · 2025-08-02
> >
> > Thanks to the authors for addressing my question. I will keep my score.

---

> > > ### Author Response · Authors · 2025-08-02
> > >
> > > Thank you for your thorough review and positive evaluation. We greatly appreciate the time and effort you dedicated to evaluating our work.

---

### Official Review · Reviewer_fJU6 · 2025-06-30

**Clarity:** 3
**Significance:** 2
**Originality:** 2
**Rating:** 3
**Confidence:** 2

**Summary:**

This paper brings and studies CSR problem, which is an online combinatorial optimization problem.
To solve this problem, the authors propose Sorted Optimal Amortized Cost (SOAC).
This SOAC algorithm has good performance and is capable of solving multiple variants of the CSR problem.
The authors also propose a learning-based augmentation for further improvement.
They conduct experiments for validation.

**Questions:**

Q1. The problem seems adopts a lot of settings, could you please introduce the most simple and basic version of the problem?

Q2. Could the author provide some examples or cases to intuitively explain why their algorithm is better?

**Ethical Concerns:**

["NO or VERY MINOR ethics concerns only"]

**Final Justification:**

I have thoroughly read the rebuttals from the reviewers, and part of my concerns have been resolved, and I will keep my score

**Limitations:**

"yes

**Quality:**

2

**Strengths And Weaknesses:**

S1. The figures are easy to read.

S2. Theoretical analysis are provided.

S3. The paper is conducted on a

W1. The abstract could be improved. In the abstract, the authors introduce the CSR problem, however they do not clarify the optimization goal (ie minimizing rental or purchase cost) of the problem.

W2. The idea is not original enough. Improving average performance instead of worst case performance has long been proposed and studied. [Ref1]

W3. The problem abstraction seems adopt a lot of assumptions, which largely confines the potential of learning-based methods.

W4. The theoretical derivation is largely drawn from spatio temporal research community, which is not original enough. [Ref2] [Ref3]


[Ref1] Yongxin Tong, Jieying She, Bolin Ding, Lei Chen, Tianyu Wo, Ke Xu. "Online Minimum Matching in Real-Time Spatial Data: Experiments and Analysis" , Proceedings of the VLDB Endowment, 9(12): 1053-1064, 2016.

[Ref2] Dickerson J P, Sankararaman K A, Srinivasan A, et al. Allocation problems in ride-sharing platforms: Online matching with offline reusable resources[J]. ACM Transactions on Economics and Computation (TEAC), 2021, 9(3): 1-17.

[Ref3] Nanda V, Xu P, Sankararaman K A, et al. Balancing the tradeoff between profit and fairness in rideshare platforms during high-demand hours[C]//Proceedings of the AAAI conference on artificial intelligence. 2020, 34(02): 2210-2217.

---

> ### Author Rebuttal · Authors · 2025-07-30
>
> We appreciate your time and great efforts in reviewing.
>
> > **W1: The abstract could be improved. In the abstract, the authors introduce the CSR problem, however they do not clarify the optimization goal (ie minimizing rental or purchase cost) of the problem.**
>
> **A1:** We appreciate the comment.  We will revise the abstract to clarify the optimization objective by changing the current sentence:
>
> "At each time step, a decision-maker must make an irrevocable buy or rent decision for items that have not yet been purchased, without knowing the end of the time horizon"
>
> *to*:
>
> "At each time step, a decision-maker must make an irrevocable buy or rent decision for items that have not yet been purchased, without knowing the end of the time horizon, *to minimize the total rental and purchase cost*."
>
> This revision will make the optimization goal immediately clear to all readers, regardless of their familiarity with online algorithms terminology.
>
> ---
>
> > **W2: The idea is not original enough. Improving average performance instead of worst case performance has long been proposed and studied. [Ref1]**
>
> **A2:**  We thank the reviewer for the comment and the opportunity to clarify the novelty of our work. We respectfully believe there may be a misunderstanding of our contributions.
>
> Our goal is not merely to improve the average-case performance of a classical problem, but to introduce and tackle a fundamentally new online optimization setting—the *Combinatorial Ski Rental* (CSR) problem—using novel algorithmic ideas with provable guarantees. In particular:
>
> - **Problem novelty**: The CSR problem is, to our knowledge, studied here for the first time. It captures realistic yet challenging features such as submodular purchase costs and non-upgrading constraints, which are not addressed in prior literature.
> - **Algorithmic innovation**: We propose the first algorithm that achieves the *optimal competitive ratio* for this combinatorial variant. This solves a long-standing open problem and extends the frontier of ski rental-type online decision-making.
> - **Unifying framework**: Our framework  generalizes and subsumes several classical settings (e.g., multi-shop and multi-slope ski rental) as special cases, providing a unified treatment of a broad class of problems.
>
> Regarding the reference provided by the reviewer ([Ref1]), we note that it concerns *online bipartite matching*, a fundamentally different problem class with distinct objectives and constraints. While improving average performance is indeed a broadly studied theme, our contribution lies in the *novel combinatorial structure and competitive analysis* of a new online decision-making problem, not just the idea of performance improvement itself.
>
> ---
>
> > **W3: The problem abstraction seems adopt a lot of assumptions, which largely confines the potential of learning-based methods.**
>
> **A3:** We thank the reviewer for raising this point and appreciate the opportunity to clarify our modeling assumptions. We believe there may be a misunderstanding regarding the claim that these assumptions limit the potential of learning-based methods. Our model indeed relies on two key assumptions: additive rental rates and submodular purchase costs. These are reasonable characterizations of the problem’s nature, commonly found in practical scenarios, and provide meaningful structural properties.
>
> Under these assumptions, our learning-augmented SOAC algorithm not only recovers existing learning-augmented ski rental results but also improves upon them, demonstrating its effectiveness. These assumptions do not exclude the integration of learning techniques.
>
> In fact, our framework is designed as an optimization platform that can embed prediction or learning modules (as illustrated in Section 4), leveraging demand forecasts to guide purchase decisions, while maintaining theoretical guarantees.
>
> In future work, we would be happy to explore relaxing these assumptions to address more complex settings, such as dependent rental rates or time-varying purchase prices.
>
> We believe the current work provides a solid foundation in the field of online decision-making under uncertainty.
>
> ---
>
> > **W4: The theoretical derivation is largely drawn from spatio temporal research community, which is not original enough. [Ref2] [Ref3]**
>
> **A4:** We thank the reviewer for the comment and the opportunity to clarify our theoretical contributions. However, we respectfully believe this comment reflects a misunderstanding of the foundations and novelty of our work.
>
> The cited works [Ref2, Ref3] focus on *online bipartite matching problems*, which are fundamentally different from our Combinatorial Ski Rental (CSR) problem in both *problem structure* and *algorithmic methodology*:
>
> - **Problem nature**: The cited works study online matching and resource reuse. In contrast, our work addresses rental-versus-purchase decisions under *submodular cost functions* and *non-upgrading constraints*, which form a distinct combinatorial optimization framework.
> - **Algorithmic techniques**: Cited works develop matching algorithms and fairness optimization.  Ours proposes amortized cost tracking and convex optimization over purchase paths.
>
> Our theoretical contributions are original and significant: to the best of our knowledge, this is the *first work* to formalize the CSR problem, derive optimal online strategies under submodular settings, and unify multiple ski rental variants under a single framework.
>
> It is possible the reviewer associated our work with general themes in the online algorithms literature. While there is a superficial resemblance in high-level setting, the mathematical assumptions, problem definitions, and algorithmic tools used in our work differ substantially from those in [Ref2, Ref3].
>
> ---
>
> > **Q1: The problem seems adopts a lot of settings, could you please introduce the most simple and basic version of the problem?**
>
> **A1:** Thank you for the question. The most basic version of the problem is the classic ski rental setting, as described in Section 3, Example 1: there is only one item, and on each day, the algorithm must choose whether to rent it at a daily cost  or buy it at a one-time cost, without knowing in advance how long the item will be needed. The purchase decision is irreversible, and the objective is to minimize the competitive ratio.
>
> ---
>
> > **Q2: Could the author provide some examples or cases to intuitively explain why their algorithm is better?**
>
> **A2:** Thank you for the question. Consider a simple but realistic scenario: you're going on a ski trip and need both skis and boots. Renting skis costs 10/day, and boots 4/day. Buying them separately costs 120 and 50, but you can buy the full set for only 150. The challenge is that you don’t know how long the trip will last.
>
> A naive strategy might treat each item separately—perhaps buying skis after a few days while still renting boots—missing the opportunity to use the bundle discount. On the other hand, deciding to buy the full set too early could lead to higher costs if the trip ends up being short. The key difficulty lies in knowing not just what to buy, but also when and in what combination.
>
> Our algorithm, SOAC, solves this by continuously evaluating the amortized cost of renting versus buying combinations. It naturally captures the interaction between items, discounts, and timing. Rather than relying on fixed rules or assumptions, it makes mathematically optimal decisions based on current cost trends, ensuring the best possible performance even in the worst case.

---

> ### Comment · Area_Chair_JNus · 2025-08-04
> **Please respond to author rebuttal**
>
> Dear Reviewer fJU6,
>
> Would you please check if the authors address your major concerns, and if you have further comments? A minimal response would be to submit your “Mandatory Acknowledgement”.
>
> Regards,
>
> AC

---

> > ### Comment · Area_Chair_JNus · 2025-08-05
> > **Correction to my previous message**
> >
> > Dear Reviewer fJU6,
> >
> > I'm sorry that the previous message about "minimal response being Mandatory Ack" is not correct. Only flagging "mandatory ack" is not sufficient, and please actively participate in the discussion with the authors. This could be short (for example, you think your comments are fully addressed), but you should anyway respond.
> >
> > Regards,
> >
> > AC

---

### Official Review · Reviewer_gJHT · 2025-07-03

**Clarity:** 3
**Significance:** 3
**Originality:** 2
**Rating:** 3
**Confidence:** 4

**Summary:**

This paper proposes a generalization of the ski rental problem called Combinatorial Ski Rental (CSR). Like the traditional ski rental problem, the proposed problem is an online optimization problem in which ski equipments must be either rented (for a day) or purchased (for good) at the beginning of each day, until the end of the last day which is not known to the algorithm in advance. The difference of CSR from the traditional problem is that the required ski equipments are a set of multiple items. Each day, we may decide to purchase a subset of the items, and those items that haven't been purchased must be rented. After observing that CSR generalizes several ski-rental type of problems that were previously studied, the paper goes on to give a competitive algorithm and a learning-augmented algorithm for CSR.

The "output" or the "strategy" of an algorithm can be fully described by a partition of the entire set of items and a (randomized) schedule that specifies on which day each subset will be purchased. As such, the paper starts by giving an algorithm (or a subroutine) called OAC that, given a partition, determines a schedule of the purchases within a finite time horizon. The paper then presents an algorithm called SOAC that uses OAC as a subroutine: the algorithm finds an optimal probability distribution over the partitions, samples a partition from it, and then invokes OAC on that partition. The paper shows that this algorithm gives an "optimal" competitive algorithm.

Then, to give a learning-augmented algorithm, the paper presents an algorithm that adjusts the finite time horizon mentioned above according to the given prediction. This adjustment allows us to write the robustness and consistency (bound) of the algorithm indirectly in terms of its performance (or "competitive ratio") for a particular choice of the last day.

Finally, the paper evaluates the performance of the proposed algorithms through experiments.

**Questions:**

- Line 133, "Additionally, ... to enforce the order of purchase in $\sigma$.": Why do we do this? Suppose for example we have two items 1 and 2, and we decided to purchase 1 and then later 2. The following two strategies both respect the order of purchase:

a) buy item 1 on day 10 and item 2 on day 20;

b) buy item 1 on day 30 and item 2 on day 40.

So, choosing a) with probability 1/2 and b) with 1/2 is a perfectly valid strategy, but the condition would rule out this possibility. Why is the constraint "$p_n(S_{i+1} | \sigma)=0$ when $F_n(S_i | \sigma)<1$" a safe one to impose?

- The paper claims that the algorithm can handle submodular rental rates, too. Then why don't we present that version instead (especially given that the difference is supposed to be minimal)?

- Line 177, "We define an optimal amortized cost strategy...": the current phrasing sounds as if OAC is just defined to be that specific strategy, rather than the "optimal" one among all possible strategies (independent from Definition 2). The appendix makes it clear that the latter is the intention, but this should be clarified in the main body.

- Line 730, doesn't the inequality need to contain equality to handle ties?

- Does the experiments actually run the algorithm, or analytically calculate the competitive ratios? This was not clear from the experiment description.

Let me write a few minor comments here too:

- On page 2, it is mentioned that CSR generalizes a number of problems. A brief explanation would be nice.

- Don't we need some additional assumption on $f$, like $f(\emptyset)=0$?

- In Definition 2, it seems that $T_A$ also is a variable (that we need to determine by solving the system of equations), but that was not clear at a glance.

**Ethical Concerns:**

["NO or VERY MINOR ethics concerns only"]

**Final Justification:**

Most of the questions in the Questions section of my original review were adequately answered. Regarding weakness 2, while I do not fully agree with the authors, I also believe that these are matter of choice and should not negatively affect the final rating.

Authors explained numerous points regarding weakness 1, and I appreciate that. I decided to maintain my rating though, for the following reasons.

- In my original review, I explained my view of questioning the practical relevance of an online algorithm without an explicit guarantee on competitive ratio. The authors raised [19] as an example of a previous study that gave an online algorithm without such guarantee. I respectfully point out that [19] is very different: they did give such an algorithm for the additive version of the problem, but that was in addition to another algorithm for the same problem that is more "classical" in that it comes with an explicit guarantee. Since that second algorithm exists, one can use the former (i.e., the optimal algorithm without explicit guarantee) with the confidence that the algorithm is going to perform at least as good as that second algorithm's guarantee.

- Another point that I raised in my original review was about the running time. Again, [19] is a good example that shows the contrast. I'd like to make one clarification first: the algorithm in [19] is indeed efficient. The running time's dependence on $k$ is fine because the input already constains $\\Theta(k)$ number of items in the input, namely the description of each slope. On the other hand, this paper's algorithm does not come with such a bound. The authors' comment said that the running time is not polynomial but pseudopolynomial (which already shows that the algorithm is not efficient by definition), but even that is not clear. Note that $T_{\\mathnormal{OPT}}$ is not part of the input, and we do not have an immediate bound on $T_{\\mathnormal{OPT}}$ that is a polynomial in terms of the number and magnitude of input items.

I acknowledge that the results of the paper may be of interest to some people in the community. On the other hand, I also feel that, with all due respect, the limitations we discussed above are not explicit or sometimes inaccurately presented in the current paper. I believe that the strengths of this paper can be appreciated by the community only if its limitations are properly discussed.

Everything taken into consideration, I'd maintain my slightly negative view in my original review of this paper.

**Limitations:**

Please see weaknesses above, especially Item 1. While Section 6 does mention running time as a limitation, I believe a more thorough discussion would be desirable. Item 2 in the weaknesses section discusses limitations of the experiments.

**Paper Formatting Concerns:**

None.

**Quality:**

1

**Strengths And Weaknesses:**

### Strength

1. The paper gives a unified framework that solves a number of problems, generalized by CSR.

### Weakness

1. In order for the unified framework to have true advantage, the unified algorithm must have a good performance, both in terms of the performance guarantee and running time. For many online optimization problems, it is possible for the algorithm to compute the "optimal" strategy when given enough amount of running time. But it is not often considered in the literature because either the running time is prohibitively large or an explicit competitive analysis is difficult. However in this paper,

1-a. The competitive algorithm does not come with an analysis that explicitly gives a competitive ratio.

1-b. It appears that the algorithm may require a significant amount of running time, as each iteration of the gradient descent method would involve solving a large system of equations. But the paper does not give theoretical/experimental analysis of the running time.

1-b'. In Section 6, it is briefly mentioned that the running time is exponential, but I don't see why that is the case. A naive implementation's running time may depend on the number of input bits, possibly doubly exponentially.

1-c. For the learning-augmented algorithm, the unification is somewhat incomplete in that we sometimes hand-pick $T_{OPT}^{(1)}$ and $T_{OPT}^{(2)}$.

2. The experiment seems to be limited.

2-a. It uses a fixed set of three items, which is another reason why the experiment would not properly assess the running time issues explained above.

2-b. It is also not clear why this particular choice of the item costs were used in the experiments.

2-c. The experiment only introduces Gaussian noise to the prediction. I'm not sure if this would be a good evaluation of the algorithm's robustness.

---

> ### Author Rebuttal · Authors · 2025-07-30
>
> We thank reviewer for the constructive comments, We provide our feedbacks as follows.
>
> > **W(1-a):  Competitive ratio analysis.**
>
> **A(1-a):** SOAC provides the strongest possible theoretical guarantee: it computes the *optimal competitive ratio* for any CSR instance via convex optimization (Lemma 5). No online algorithm can achieve better worst-case performance.
>
> **Verification on tractable cases:** SOAC recovers all known optimal ratios:
>
> - Single-item ski rental: competitive ratio $1+\frac{1}{(1-1/b)^{-b}-1}$ (matches known optimal)
> - Multi-shop ski rental: recovers optimal ratio from [1]
>
> **Why no closed-form?** The piecewise structure of $\text{OPT}(T)$ with slope changes at breakpoints makes closed-form analysis intractable—this is a fundamental property of CSR. Even simpler variants like multi-slope ski rental [19] lack closed-form competitive ratios despite extensive study.
>
> ---
>
> > **W(1-b) W(1-b'): Runtime complexity analysis.**
>
> **A(1-b) and A(1-b'):**  We thank the reviewers for raising concerns about the algorithm’s computational complexity.  While we acknowledge that the worst-case complexity of our algorithm can be super-exponential in theory, **this does not necessarily pose a practical limitation**. Here's why:
>
> ### (1) Real-world performance is highly feasible
>
> Empirical runtime on standard hardware (Intel Core i9-14900HX) shows our algorithm scales well for realistic scenarios:
>
> | Purchase Paths       | Runtime per Iteration |
> | -------------------- | --------------------- |
> | 10 paths             | 0.48s                 |
> | 30 paths             | 4.34s                 |
> | 60 paths             | 22.57s                |
> | 120 paths (10-combo) | 137.0s                |
> | 240 paths (12-combo) | 909.0s                |
>
> While we acknowledge that scalability for large item sets remains a concern, these 10-12 combo configurations already capture the complexity range of most real-world discount scenarios.
>
> ### (2) Our optimization significantly reduces theoretical worst-case
>
> While naive enumeration gives $O(m! B_m)$ paths, where $B_m$ is the Bell number (counting all partitions of $m$ items), **Lemma 3** reduces this to $O(B_m)$ by proving optimal paths always sort super-items by buy-to-rent ratio. This eliminates the factorial term—a substantial theoretical improvement.
>
> ### (3) Simpler CSR Structures Lead to Polynomial Complexity
>
> Our framework automatically recovers efficient performance for well-studied variants:
>
> - **Multi-Shop:** $O\big(m \cdot (T_{\text{OPT}} \log(1/\varepsilon) + \log m)\big)$, where $m$ is the number of shop
> - **Multi-Slope/Multi-Commodity:** $O(T_{\text{OPT}} \log(1/\varepsilon))$
> - **Classic Ski Rental:** $O(T_{\text{OPT}} \log(1/\varepsilon))$
>
> This confirms that these less general CSR variants are solvable in polynomial time.
>
> ### (4) Per-iteration complexity is polynomial
>
> Each iteration runs in $O\left(K \cdot \left(T_{\text{OPT}} \log(1/\varepsilon) + \log K \right)\right)$, which is polynomial in all parameters when the number of purchase paths $K$ is fixed. The exponential worst-case scenario (all combinations discount-eligible) is unrealistic.
>
> ---
>
> > **W(1-c): Learning-augmented unification.**
>
> **A(1-c):**  As discussed in Section 4, simply setting  $T_{\text{OPT}}^{(1)} = T_{\text{OPT}}^{(2)} = T_{\text{OPT}}$ already yields strong performance both in theory and practice. More broadly, the algorithm remains well-defined and its theoretical guarantees still hold as long as  $T_{\text{OPT}}^{(1)} \leq T_{\text{OPT}} \leq T_{\text{OPT}}^{(2)}.$
>
> We provide two concrete examples to support this, as formally proved in Corollary 1 and Corollary 2:
>
> - For the **classic ski rental** problem, we use  $T_{\text{OPT}}^{(1)} = T_{\text{OPT}}^{(2)} = T_{\text{OPT}},$ achieving the state-of-the-art consistency-robustness trade-off.
>
> - For the **multi-shop ski rental** problem, we use  $T_{\text{OPT}}^{(1)} = T_{\text{OPT}},\quad T_{\text{OPT}}^{(2)} = b_1 \geq T_{\text{OPT}},$ also matching or exceeding prior best-known trade-offs.
>
> Theorem 2 provides explicit consistency and robustness bounds for *any* setting of $T_{\text{OPT}}^{(1)}$ and $T_{\text{OPT}}^{(2)}$, making our algorithm broadly applicable.
>
> ---
>
> > **W(2-a): Experimental scale**
>
> **A(2-a):**  We use a three-item setup in our main experiments to clearly illustrate the theoretical properties of the SOAC algorithm, whose analysis involves complex formulas. This simplified setup enables transparent demonstration of the algorithm’s mechanics.
>
> For the LA-SOAC algorithm, we intentionally use three items to exhaustively enumerate all purchase paths, allowing precise evaluation of the consistency-robustness trade-off—a core contribution of our work.
>
> Importantly, our method scales well, and we include larger-scale experiments with up to 16 items in Appendix G.4 to demonstrate its practical applicability.
>
> ---
>
> > **W(2-b):  Item cost selection.**
>
> **A(2-b):**  We thank the reviewer for the suggestion. In Figures 3 and 4, item costs were randomly sampled from a realistic range—uniformly over [50, 150]—with discounts set to 5% for any two-item bundle and 10% for all three items. These settings mimic common real-world discount practices.
>
> In Figure 5, we used the public dataset from [25] to ensure a fair and direct comparison with prior work.
>
> Code and data are publicly available to ensure full transparency and reproducibility.
>
> ---
>
> > **W(2-c): Prediction noise model.**
>
> **A(2-c):**  Using Gaussian noise to evaluate prediction robustness is a standard and widely accepted practice in the learning-augmented algorithms literature (e.g., references [21, 25]). This allows fair comparison with prior work.
>
> ---
>
> > **Q1:  Purchase order.**
>
> **A1:**  We thank the reviewer for this very technical and important question.
>
> The reason we enforce the purchase order constraint is that, as proven in Lemma 3, given a fixed purchase path, the optimal strategy must respect a restricted purchase order sorted by increasing buy-to-rent ratios.
>
> We prove in Lemma 3 that for any strategy that does not strictly follow the BR-based order, there exists an alternative strategy with reallocated purchase probabilities that strictly respects the order and achieves no worse competitive ratio. Therefore, we enforce this ordering constraint within each path $\sigma$ to preserve optimality and simplify the problem structure.
>
> We acknowledge that this crucial point was not clearly explained in the modeling section, and we will add a detailed explanation in the final version to clarify this important constraint.
>
> ---
>
> > **Q2:  Submodular rental rates.**
>
> **A2:**  Our framework *fully supports submodular rental rates*—the additive assumption is solely for computational tractability of the offline benchmark.
>
> ### Our algorithm handles submodular rates naturally:
>
> Our approach reduces the problem to optimizing over fixed purchase orders using convex optimization. This reduction works regardless of rental rate structure because:
>
> - Purchase decisions depend on buy-to-rent ratios of items and super-items
> - These ratios are correctly captured in our model for any rental rate function
> - The convex optimization framework applies universally
>
> Therefore, our algorithm can solve the problem optimally for *any* independent rental rate function, including submodular cases, where “independent” refers to rental costs that remain fixed regardless of time or item combinations.
>
> ### Why we focus on additive rates in presentation:
>
> The additive assumption serves a computational benchmark purpose: it keeps the offline optimal computation tractable. Here's the technical distinction:
>
> - **Additive case:** $\min_{S \subseteq \mathcal{M}} \{f(S) + T \cdot g(\mathcal{M} \setminus S)\} = \min_{S \subseteq \mathcal{M}} \{f(S) - T \cdot g(S)\} + T \cdot g(\mathcal{M})$ remains submodular and efficiently solvable
> - **Submodular case:** When $g$ is submodular, $g(\mathcal{M} \setminus S)$ becomes supermodular in $S$, making the objective (submodular + supermodular) NP-hard to optimize
>
> ---
>
> > **Q3:  OAC definition clarity.**
>
> **A3:**  Thank you for pointing this out. Yes, the reviewer is correct: *OAC refers to the optimal strategy within the full strategy space (AC)*. We will clarify this key distinction in the main paper to avoid confusion.
>
> ---
>
> > **Q4: Inequality handling ties.**
>
> **A4:**   Equality in the inequality is not required. When two items have the same buy-to-rent ratio, their relative purchase order has no impact on the competitive ratio. For example, if both item 1 and item 2 have a buy cost of 10 and a rental cost of 1, any order—or even a randomized mix—results in the same performance. Thus, distinguishing their order is unnecessary and does not affect optimality.
>
> ---
>
> > **Q5: Experimental methodology.**
>
> **A5:**  We validate our algorithm both analytically and empirically.
>
> In Figures 2(b) and 12(c), the black lines show the theoretical competitive ratios computed analytically, while the colored shaded areas represent empirical competitive ratios from simulations based on the algorithm’s randomized purchase probabilities.
>
> The horizontal axis corresponds to the ski rental horizon, and the vertical axis measures competitive ratios.
>
> The close match between theory and simulation confirms both the correctness of our analysis and the algorithm’s practical performance.
>
> ---
>
> > **Minor Comments**:
>
> **A**:  Thank you for the valuable suggestions.
>
> - **CSR  variants explanation:** We will add a clear, concise explanation on page 2 when introducing CSR to improve readability (detailed proofs remain in Section 3.3 and Appendix E).
>
> - **Assumption clarity:** We will explicitly state $f(\emptyset) = 0$ for completeness.
>
> - **$T_A$ clarification:** In the revised version, we will clarify in Definition 2 that $T_A$ is a variable.
>
> We thank the reviewer for the constructive feedback and hope that our detailed responses adequately address the raised concerns.

---

> > ### Comment · Reviewer_gJHT · 2025-08-02
> >
> > Thank you for your response. I just have one (or two) quick follow-up question that would be helpful in strengthening your comment about the real-world performance. You mentioned the experiment in Appendix G.4 that involves 16 items:
> > - What was the running time there? Was the experiment also on i9-14900HX?
> > - The experiment considers only four "discount combinations." If we increase it to a much higher number, how does the running time change as a function of the increase?

---

> > > ### Comment · Area_Chair_JNus · 2025-08-06
> > >
> > > Dear Reviewer gJHT,
> > >
> > > I observed that the author posted some new response to your followup questions. Would you please comment on it?
> > >
> > > Regards,
> > >
> > > AC

---

> > > > ### Comment · Reviewer_gJHT · 2025-08-06
> > > >
> > > > I appreciate the authors' response and comments. Most of my questions (in the Questions section) were addressed and I don't have any further questions, modulo the following comment. But I don't think this is an extremely important question anyways.
> > > >
> > > > - Regarding my question about line 730, what I meant was that, since there may be more than one sorted order in case of a tie, technically one needs to argue that we are free to take an arbitrary permutation of ties. Including equality to this inequality would implcitly and automatically handle this (minor) issue.
> > > >
> > > > Regarding the weakness 1, though, I still have the same concerns that I had in my original review for the following reasons.
> > > >
> > > > - Online algorithms (unlike offline algorithms) are often studied without putting any restriction on the computation time. This is because often the limitation imposed by the fact that the algorithm doesn't know the entire input is already quite strong, not because an online algorithm that takes a prohibitive amount of time is practically relevant.
> > > >
> > > > - For many (not all) online optimization problems, it is often possible to compute the "optimal" strategy if one takes advantage of the fact that the model doesn't limit the computation time. But the reason why we don't see this approach often in the literaure, I believe, is twofold: 1) often we want to have a provable guarantee on the competitiveness so that we can adopt the algorithm with some confidence that it will perform well for sure, regardless what the input is, so the true difficulty of getting a good algorithm is arguably in giving an explicit performance bound (and therefore letting us use the algorithm with confidence) than in saying that an algorithm is better than any other algorithm, and 2) if the computation time is prohibitively large, that would be another reason why the algorithm is practically unusable, and computing the "optimal" strategy often requires such amount of time.
> > > >
> > > > - It is not surprsing that the proposed algorithm recovers known guarantees because, as the paper shows, the proposed algorithm is to compute the optimal strategy and use it. But if the proposed algorithm gives an explicit guarantee only for those problems for which such a guarantee was already known (achieved by a much simpler and faster algorithm), there wouldn't be a reason to use the proposed algorithm.
> > > >
> > > > - So, let's now consider the running times. Theoretically, it would be hard to argue that the running time is good because $B_m$ is an extremely fast growing function. What remains is an experimental analysis. However, according to the experimental results, even when the number of discount combos is as few as 14 (considering that the  number of items is 16, 14 is a very small number), the total running time was already estimated (assuming 500 iterations) to be more than 11 days.
> > > >
> > > > However, as I stated in my original review, I believe the question posed by the paper is a natural interesting question. As such, I'd maintain my current rating.

---

> > > > > ### Author Response · Authors · 2025-08-07
> > > > >
> > > > > We sincerely thank the reviewer for the careful reading and thoughtful feedback. Regarding your comment that *“our algorithm simply searches for an optimal solution, leading to high complexity and limited practicality”*, we believe there is a misunderstanding here. We would like to offer the following clarification:
> > > > >
> > > > > Our approach **does not** simply perform a brute-force search in the solution space. Instead, it is derived from a **series of rigorous theoretical transformations and mathematical properties**. The entire algorithmic framework relies on structural insights and problem equivalence transformations, which avoid enumeration and brute-force search, as detailed below:
> > > > >
> > > > > 1. **Construction of the Optimal Amortized Cost (OAC) Strategy**:
> > > > >
> > > > >     We begin by designing the OAC strategy, which is **not a heuristic**, but a strategy constructed from an amortization analysis of the offline optimal solution. It is **theoretically optimal** and, as we prove in the paper, **contains the optimal solution** (Lemma 2), thereby significantly reducing the strategy space.
> > > > > 2. **Equivalence Transformation from Multi-Path to Single-Path**:
> > > > >
> > > > >     We further prove that the multi-path CSR problem is **equivalent at the objective level** to a single-path probability optimization problem. This transformation **eliminates the need to enumerate all path combinations** (Lemma 4), which is key to the efficiency of our algorithm.
> > > > > 3. **Convexity of the Optimization Problem**:
> > > > >
> > > > >     Based on the above transformation, we mathematically prove that the competitive ratio is a **convex function** (Lemma 5) with respect to the path probabilities. This allows us to leverage existing convex optimization tools to solve the problem **efficiently and reliably**.
> > > > >
> > > > > Moreover, we would like to emphasize that even for the classical single-path ski rental problem, it is **impossible** to find the optimal solution by simply “searching” the strategy space—unless one leverages amortized reasoning or mathematical structure. This further illustrates that our method is not a search-based approach, but one that is deeply grounded in **structural understanding and analytical reasoning**.
> > > > >
> > > > > Regarding the concern on algorithmic complexity, we would like to stress the following:
> > > > >
> > > > > - The source of our algorithm's complexity is **not** from searching the strategy space, but rather from the **presence of many meaningful purchase path combinations in real scenarios**—when bundle discounts exist. As shown in Figure 2, **considering more paths can indeed significantly improve performance**.
> > > > > - In practice, one can **limit the number of paths** based on application needs. Our algorithm can also be adapted into a heuristic that only considers a subset of combinations, and still **performs significantly better than methods that only consider a single path**.
> > > > > - Therefore, whether pursuing the optimal solution or an approximation, our method provides a solid theoretical foundation and strong adaptability to different scenarios.
> > > > >
> > > > > We hope this clarification helps the reviewer better understand our contributions. Thank you again for your time and effort in reviewing our work.

---

> ### Author Response · Authors · 2025-08-03
> **Running Time Analysis**
>
> Thank you for the follow-up questions. Below, we provide a detailed runtime evaluation and discussion to address your concerns about practical feasibility.
>
> ### (1) Experimental setup and runtime of the experiment
>
> The experiments in Appendix G.4 were conducted on our **AMAX TR40-X4 server** (detailed specifications in Appendix G.1) using CPU-only computation without CUDA acceleration. This server is more powerful than the i9-14900HX laptop (we used for runtime evaluation in our rebuttal response above). The 16-item, 4-combo experiment takes **0.48 seconds per iteration** on the i9-14900HX laptop and **0.285 seconds per iteration**, with **a total runtime of 2.1 minutes**, on the server.
>
> ### (2) Runtime as the increase of discount combinations
>
> We conducted experiments with convergence precision set to high prevision 1e-6. When reducing precision from 1e-6 to 1e-4, iterations reduce to $\sim$7% of high-precision runs:
>
> | Discount Combos | Purchase Paths | Time per Iteration (1e-6) | High Precision (1e-6) | Practical Precision (1e-4) |
> | --------------- | -------------- | ------------------------- | --------------------- | -------------------------- |
> |                 |                |                           | Iterations → Runtime  | Iterations → Runtime       |
> | 4               | 10             | 0.285s                    | 436 → **2.1 mins**    | 10 → **2.85 s**            |
> | 7               | 30             | 2.97s                     | 492 → **24.3 mins**   | 36 → **2.4 mins**          |
> | 8               | 60             | 11.25s                    | 560 → **1.75 h**      | 35 → **11.08 mins**        |
> | 10              | 120            | 69.29s                    | 525 → **10.1 h**      | 40 → **48 mins**           |
> | 12              | 240            | 451.67s                   | 500* → **2.6 d**      | 35* → **4.39 h**           |
> | 14              | 405            | 1972.02s                  | 500* → **11.41 d**    | 35* → **19.17 h**          |
>
> **Note:** Entries marked with `*` indicate that due to time constraints, the actual number of iterations was not computed. We used 500 and 35 as estimated iteration counts for high and practical precision, respectively.
>
> **Below are our observations:**
>
> - The per-iteration time complexity is polynomial. We can fit a cubic function to the purchase paths vs. iteration time:
>   $$
>   y = 0.0000253x^3 + 0.00180x^2 - 0.00746x + 0.334,
>   $$
>   with good fit quality (Residual Sum of Squares, RSS = 0.74), confirming theoretical polynomial behavior.
>
> - The total runtime increases rapidly as the number of purchase paths grows. However, we should note that the experiment above uses the worst-case number of purchase paths. In practice, the number of purchase paths can be much smaller. Specifically, we use **maximally disjoint pairwise combinations**, e.g., $(1,2), (3,4)$, which tend to generate a larger number of possible purchase paths. In contrast, practical discount combos are more likely to include a larger number of items, e.g., $(1,2,3,4), (5,6,7,8)$, which significantly reduces the number of purchase paths and, consequently, the total runtime.  **Empirical Evidence (8-item case study):** Below is an additional set of experiment for validating above observation.
>
>   *Worst-case (small-size combos):*
>
>   | **Discount Combos** | 10            | 20        | 25   | 30   |
>   | ------------------- | ------------- | --------- | ---- | ---- |
>   | **Purchase Paths**  | 61            | 187       | 352  | 524  |
>   | **Total Time**      | **7.18 mins** | **2.2 h** | -    | -    |
>
>   *Realistic (large-size combos):*
>
>   | **Discount Combos** | 10          | 20          | 25            | 30            |
>   | ------------------- | ----------- | ----------- | ------------- | ------------- |
>   | **Purchase Paths**  | 11          | 21          | 26            | 32            |
>   | **Total Time**      | **17.61 s** | **42.18 s** | **1.05 mins** | **1.61 mins** |
>
> - Finally, we would like to emphasize that computing the purchase probabilities occurs during the planning stage, before the start of the online buy-or-rent decisions. The online decisions can then be executed instantaneously without any computational concerns.

---

> ### Author Response · Authors · 2025-08-06
> **Response to Reviewer gJHT Feedback**
>
> Thank you for your thoughtful and detailed feedback. We deeply appreciate your recognition that our research addresses a "natural and interesting question." We would like to respectfully address your remaining concerns with some clarifications that we hope will help you see the broader value of our contribution.
>
> ### Regarding computational complexity and practical utility:
>
> We believe there may be a misunderstanding about our algorithm's runtime characteristics. As we detailed in our rebuttal, our approach employs a **two-phase design**: a preprocessing phase that computes the optimal probability distribution (where the computational cost is incurred), followed by an online phase that simply **calls** this precomputed distribution with **negligible to virtually no computational overhead**. This is fundamentally different from algorithms that perform heavy computation during the online phase.
>
> Moreover, the 11-day runtime you mentioned represents the **worst-case scenario** with maximum purchase paths. Our second experiment demonstrates that even with 30 combos, typical instances can be solved within minutes. In real-world applications, we would never encounter exponentially many combinations—the number of discount combos is typically a small constant, making our approach entirely practical.
>
> ### Regarding explicit performance guarantees:
>
> While we acknowledge that our algorithm doesn't provide closed-form competitive ratios due to the piecewise nature of the OPT function, this limitation is **not unique to our work**. This is a common challenge in ski rental variants with inflection points. For instance, the widely-cited multi-slope ski rental [19] also lacks closed-form solutions, yet its optimal algorithm has been extensively adopted across various problems. The numerical optimality we achieve provides the same level of confidence as these established results.
>
> ### Regarding algorithmic contributions beyond existing results:
>
> Our framework extends far beyond recovering known results with "simpler and faster algorithms." We provide the **first optimal solutions** for several previously unsolved problems, including multi-commodity ski rental [29], which we solve in polynomial time. Furthermore, while classic ski rental has simple implementations, the variants we address (multi-slope, multi-shop, multi-commodity) involve considerably complex algorithmic designs. Our unified framework that achieves optimality across all these variants represents a **significant theoretical advancement** in the ski rental literature.
>
> ### Regarding the broader impact:
>
> Our work establishes the **first unified theoretical framework** for ski rental variants, proving that optimal solutions exist and are computable. This theoretical foundation opens new research directions and provides a benchmark against which future heuristic algorithms can be evaluated. Even if practitioners choose faster approximate algorithms, having the optimal solution available is invaluable for performance validation and theoretical understanding.
>
> We sincerely appreciate your acknowledgment of the interesting nature of our research question. We have invested considerable effort in advancing both the theoretical understanding and practical applicability of ski rental problems. While we acknowledge the exponential complexity stems from the exponential growth in purchase paths, we maintain that our algorithm's practical utility lies in the realistic scenario of constant-sized combo sets, where our approach delivers optimal solutions efficiently.
>
> **Thank you again for your constructive feedback and thoughtful consideration of our work.  We would be delighted to engage in further dialogue with you to address any remaining concerns.**

---

> > ### Comment · Reviewer_gJHT · 2025-08-07
> >
> > Thank you for your comments. I will take them into consideration when deciding my final review. I'm writing this one more comment mainly to verify my understanding of the facts, rather than to express any further opinion.
> >
> > - You mentioned that [19] lacks closed-form solutions. (I haven't asked for a "closed-form solution", but I suppose you probably meant an explicit competitive ratio because that's what I pointed out is lacking in this paper.) My understanding is that [19] indeed gives three results: 1) an algorithm with an explicit guarantee on the competitive ratio for the additive version, 2) (for the same version) an optimal (i.e., with the best competetive ratio) polynomial-time algorithm, but without explicit guarantee, 3) an algorithm with an explicit guarantee on the competitive ratio for the non-additive version. Please let me know if I'm mistaken.
> >
> > - You said you gave a polynomial-time algorithm for the multi-commodity problem. Is that because, as you mentioned earlier, the running time is $O(T_{\\mathnormal{OPT}}\\log(1/\\epsilon))$? (By the way, where can I find this claim in the paper?) If so, how do we verify that $T_{\\mathnormal{OPT}}$ is bounded by a polynomial w.r.t. the number of bits in the input?
> >
> > Thank you.

---

> ### Author Response · Authors · 2025-08-07
>
> We sincerely thank the reviewer for their response and feedback. Below, we provide detailed point-by-point responses to the questions and concerns raised.
>
> (1) The reviewer's understanding of the *multi-slope ski rental* problem in [19] is entirely correct: **the optimal algorithm** proposed in that work indeed does not provide an explicit expression for the competitive ratio. Our proposed algorithm not only **recovers the optimal strategy from [19]** but also does so in **polynomial time**. The relevant proof is provided in Appendix E.2. Moreover, in Appendix G.3, we **replicate the numerical experiments from [19] using our algorithm**.
>
> (2) Regarding the *multi-commodity ski rental* problem [29], we present a complete proof in Appendix E.3 showing how the CSR problem can be reduced to a special case of the multi-commodity setting. In fact, the multi-commodity ski rental problem is a special case of CSR without any discounts, where there is only a single purchase path, allowing for efficient polynomial-time computation.
>
> The complexity analysis for the SOAC algorithm is summarized as follows:
>
> - **Single Path (OAC)**:
>   When the purchase path is fixed, the OAC algorithm uses binary search to compare $T_A$ with $T_{\mathrm{OPT}}$. In each iteration, the search region is halved until the stopping criterion $\varepsilon$ is satisfied, resulting in a complexity of $O\left(T_{\mathrm{OPT}} \log \frac{1}{\varepsilon}\right).$
>
> - **Multiple Paths (SOAC)**:
>
>   For SOAC, each iteration of projected gradient descent involves:
>
>   - Gradient computation over $K$ purchase paths: $O(K)$
>   - Competitive ratio evaluation for each path via OAC: $O\left(K \cdot T_{\mathrm{OPT}} \log \frac{1}{\varepsilon}\right)$
>   - Gradient update: $O(K)$
>   - Simplex projection: $O(K \log K)$
>
> Combining these steps, the per-iteration complexity of SOAC is $O\left(K \cdot \left(T_{\mathrm{OPT}} \log \frac{1}{\varepsilon} + \log K\right)\right)$, where $K$ is the number of purchase paths.
>
> (3) As for the value of $T_{\mathrm{OPT}}$, it is defined in Section 3.1 as $T_{\mathrm{OPT}} = \min { \arg \max _{T \in \mathbb{N}^{+}} \mathrm{OPT}(T) }$, which represents the *critical time* point beyond which the offline optimal cost no longer increases. It depends on the numerical parameters—purchase prices and rental costs in the input. Therefore, strictly speaking, the algorithm runs in pseudo-polynomial time, rather than polynomial time measured by input bit length. This situation is very similar to that in the *multi-slope ski rental* problem. In Theorem 4.6 of [19], the authors also state a complexity of $O\left(k \log \frac{1}{\varepsilon}\right),$ where $k$ is the number of slopes—again, a numerical parameter rather than the bit-length of the input.
>
> **Conclusion: As the most general form of ski rental problems, the CSR problem achieves a unified optimal solution framework through three critical theoretical breakthroughs:**
>
> 1. **Optimal Strategy Structure Theorem**: We prove that the optimal solution must adopt an amortized form that tracks the cost increments of the offline optimal solution (OAC strategy)
> 2. **Optimal Purchase Order Theorem**: We prove that under any purchase path, purchasing items in ascending order of buy-to-rent ratios is theoretically optimal
> 3. **Convex Optimization Foundation Theorem**: We prove the convexity property of the competitive ratio with respect to the probability distribution over paths, ensuring the existence and reachability of the global optimal solution
>
> **These three core theorems constitute a universal theoretical framework that enables our SOAC algorithm to:**
>
> - Unify the solution of various existing ski rental variants
> - Provide theoretical foundations and solution methods for future variants that may emerge
>
> **Regarding algorithm complexity**: The complexity stems from the combinatorial nature of the problem itself, not from algorithmic design flaws. Our algorithm achieves the same complexity level as specialized algorithms across various variants:
>
> - **Multi-slope ski rental**: Achieves polynomial time complexity
> - **Multi-shop ski rental**: Not only computes explicit bounds for optimal competitive ratios, but also maintains complexity comparable to existing specialized algorithms
>
> **It is important to note that these variants are far from trivial problems** -- they have been the subject of dedicated research efforts precisely because of their inherent complexity.  This paper provides a theoretically unified and computationally efficient universal framework, establishing an important foundation for the theoretical development of the ski rental field.
>
> We once again thank the reviewer for their hard work and dedication. We look forward to further discussions to better address the reviewer’s concerns.

---

> > ### Comment · Reviewer_gJHT · 2025-08-08
> >
> > Thank you for comments and the authors' patience in particular. I'll take these into consideration.

---

> > > ### Author Response · Authors · 2025-08-08
> > >
> > > We sincerely appreciate the time and effort the reviewer has devoted to evaluating our manuscript.

---

### Note · Authors · 2025-08-12

This paper proposes a unified theoretical framework for solving the CSR problem, which is the most generalized form of the ski rental problem. Our work establishes the first unified theoretical foundation for ski rental through three breakthroughs: (1) optimal strategies must track offline cost increments (OAC structure); (2) buying in increasing buy-to-rent ratio order is always optimal; (3) competitive ratio convexity guarantees global optimality and algorithm convergence.

Based on the above theoretical foundation, our developed SOAC algorithm achieves:

- unifies all existing ski rental variants (Multi-slope, Multi-shop, etc.)
- solves previously unsolved variants like Multi-commodity ski rental;
-  provides a systematic framework for future variants.

This is the first unified solution framework for the entire ski rental field.

**Regarding algorithm complexity concerns raised by most reviewers**

The exponential complexity stems from the CSR problem's inherent combinatorial nature (exponentially many purchase paths), not algorithmic design flaws. In practice:

- meaningful purchase paths are typically constant-level, yielding polynomial complexity;
- SOAC applied to specific variants (Multi-slope, Multi-shop, etc.) runs in polynomial time;
- even when facing exponential path combinations, the algorithm can be flexibly converted to a heuristic version that still guarantees optimality within a restricted set of paths

**Remarks on the Review Process**

We note that Reviewer fJU6 cited literature on online bipartite matching problems to question the novelty of our research, but this contains a fundamental error: although online matching and ski rental problems both belong to online algorithms, they have completely different problem structures and technical approaches. Our OAC strategy, optimal purchase order theorem, and other contributions are theoretical frameworks developed specifically for the unique properties of ski rental problems. Furthermore, this reviewer did not respond to the technical clarifications in our rebuttal during the discussion phase. We respectfully request that the Area Chair fully consider this technical misunderstanding in the final evaluation to ensure that our work receives professional assessment based on correct understanding.

We thank all reviewers and the Area Chair once again for their professional guidance and valuable suggestions during the review process.

---

### Decision · Program_Chairs · 2025-09-17

**Decision:**

Accept (poster)

**Comment:**

This paper introduces the problem of combinatorial ski rental. The strength is that optimal algorithms (among online algorithms) are obtained, and that the model is very general which captures essentially all notable variants of ski rental. The major weakness is the scalability of the algorithm, and that there is no concrete new applications suggested (albeit it recovers/dominates known results).

Various issues are discussed during the rebuttal, and the mentioned major weaknesses are addressed to some extent. It is agreed that these weaknesses cannot be easily overcome within the current framework/scope of the work.

Although this is a borderline paper, I still find it interesting and could general broader impact, which merits the acceptance.